# dcHiC detects differential compartments across multiple Hi-C datasets

Abhijit Chakraborty [1,6] ✉, Jeffrey G. Wang [1,2,5,6] & Ferhat Ay [1,3,4] ✉

The compartmental organization of mammalian genomes and its changes play important roles in distinct biological processes. Here, we introduce dcHiC, which utilizes a multivariate distance measure to identify significant changes in compartmentalization among multiple contact maps. Evaluating dcHiC on four collections of bulk and single-cell contact maps from in vitro mouse neural differentiation ($n = 3$), mouse hematopoiesis ($n = 10$), human LCLs ($n = 20$) and post-natal mouse brain development ($n = 3$ stages), we show its effectiveness and sensitivity in detecting biologically relevant changes, including those orthogonally validated. dcHiC reported regions with dynamically regulated genes associated with cell identity, along with correlated changes in chromatin states, subcompartments, replication timing and lamin association. With its efficient implementation, dcHiC enables high-resolution compartment analysis as well as standalone browser visualization, differential interaction identification and time-series clustering. dcHiC is an essential addition to the Hi-C analysis toolbox for the ever-growing number of bulk and single-cell contact maps. Available at: https://github.com/ay-lab/dcHiC.

The three-dimensional organization of chromatin in the nucleus has been of interest to scientists for more than a century now. The observation that different chromosomes occupy a defined space in the nucleus dates back to Carl Rabl's work in animal cells in 1885[1]. Since then, many experimental techniques have been developed to image and map chromatin, allowing us to look at chromatin organization at an ever-increasing resolution. The greatest strides in this area have been made in the past decade following the advent of genome-wide conformation capture techniques. We now know that interphase chromosomes are folded into multiple layers of hierarchical structures. Each layer contributes to the establishment and maintenance of the epigenetic landscape that controls cellular state and function.

Among these, the megabase-scale compartmental organization of eukaryotic genomes has been shown to play a critical role in transcription, DNA replication, accumulation of mutations, and DNA methylation[2–12]. In broad terms, two types of compartments divide the genome into regions of open and active chromatin (compartment A) versus inactive and closed chromatin (compartment B)[13]. Further analysis of each compartment revealed subsets of regions with markedly different properties within each class called subcompartments[14,15] as well as a putative third class (intermediate or I) that is at the interface between A and B and is reorganized in cancer cells[9].

The main method to extract compartment information has been to analyze high-throughput chromosome conformation capture (Hi-C) contact maps using Principal Components Analysis (PCA)[13,16,17]. Briefly, this process involves distance normalization (observed/expected for each genomic distance) of the Hi-C contact map for each chromosome at a particular resolution (generally between 100 kb to 1 Mb) followed by transformation into a correlation matrix, where each entry ($i$, $j$) denotes the correlation of row $i$ and row $j$ (or column $i$ and $j$ since symmetric) of the distance-normalized Hi-C map. The eigenvalue decomposition of the correlation matrix provides eigenvalues and eigenvectors, and typically the first principal component (PC1) derived from them represents the genomic compartments A and B. If PC1

[1]Centers for Autoimmunity, Inflammation and Cancer Immunotherapy, La Jolla Institute for Immunology, La Jolla, CA 92037, USA. [2]The Bishop's School, La Jolla, CA 92037, USA. [3]Bioinformatics and Systems Biology Program, University of California San Diego, La Jolla, CA 92093, USA. [4]Department of Pediatrics, University of California San Diego, La Jolla, CA 92093, USA. [5]Present address: Harvard College, Cambridge, MA 02138, USA. [6]These authors contributed equally: Abhijit Chakraborty, Jeffrey G. Wang. ✉e-mail: abhijit@lji.org; ferhatay@lji.org

corresponds to chromosome arms or other broad patterns in the Hi-C map (e.g., copy number differences), the second principal component (PC2) is likely to represent A and B compartments. The A and B compartment labels are assigned to the positive and negative stretches of the selected PC, respectively. However, depending on the implementation of eigenvalue decomposition, it may be necessary to reorient these assignments correctly using GC content or gene density.

Whether one is interested in the two major compartments or their more nuanced subsets, the principal components derived from PCA have been the major determinants of compartment type. However, standard PCA is limited to analyzing each Hi-C contact map individually, and computational methods that can compare compartmentalization across multiple (>2) Hi-C datasets are needed. This is becoming an obstacle in analyzing the ever-increasing chromatin conformation data, either from Hi-C or its variants[18–25], generated across many cell types and conditions[26]. In addition, single-cell Hi-C datasets now provide a rich test bed for studying cluster-specific and/or time dependent changes in compartmentalization[27–31]. Technical challenges such as selecting the correct PC and sign that represents A/B compartments and their scaling across different datasets become larger problems when comparing many bulk or single-cell Hi-C contact maps. Thus far, comparative compartment analysis has been mainly limited to examining compartment flips between two Hi-C maps at a time[32,33].

Here, we introduce dcHiC (differential compartment analysis of Hi-C), a method that identifies statistically significant differences in compartmentalization among two or more contact maps, including changes that are not accompanied by a compartment flip. Our method implements a parallelized partial singular value decomposition (SVD) that uses a memory-efficient data structure (Filebacked Big Matrix or FBM) and efficiently computes only the first few singular vectors (i.e.,

eigenvectors) that are need for compartment analysis[34]. We follow this by quantile normalization to obtain comparable compartment scores across two or more Hi-C maps/replicates at a time (Fig. 1, Step 1). dcHiC then utilizes the normalized component scores to derive a multivariate distance measure[35] (Fig. 1, Step 2) to estimate the statistical significance of compartment differences. If available, dcHiC utilizes variance among Hi-C replicates as covariates for independent hypothesis weighting (IHW)[36] to correct for multiple testing. With our methodology, compartment analysis can be conducted on Hi-C maps with or without replicates at resolutions up to 10 kb and for pseudo-bulk Hi-C profiles from as few as 100 cells per condition. Further downstream, dcHiC provides a number of analysis features, including standalone IGV browser[37] visualization of results, detection of differential interactions involving significant differential compartments, time-series clustering of compartment scores, and a module for determining enriched Gene Ontology terms from differential compartments.

To assess the biological relevance of the identified differences, we apply dcHiC to several different collections of bulk and single-cell Hi-C datasets, including mouse neuronal development ($n = 3$), mouse hematopoiesis ($n = 10$), a set of lymphoblastoid cell lines (LCLs) from different human populations ($n = 20$) and single-cell Hi-C data from post-natal mouse cortex and hippocampal brain regions at six different time points clustered into three developmental stages ($n = 3$). Analyzing each Hi-C dataset at resolutions ranging from 10 Kb to 250 Kb, we identify relevant compartmentalization differences reflecting the underlying biology in the respective scenarios. In the mouse neuronal differentiation model, dcHiC identified compartmental changes for loci involving critical genes associated with cellular identities in mouse embryonic stem cells (mESC) and neuronal differentiation, such as *Dppa2/4, Zfp42, Ephb1*, and *Ptn*, as well as GO term

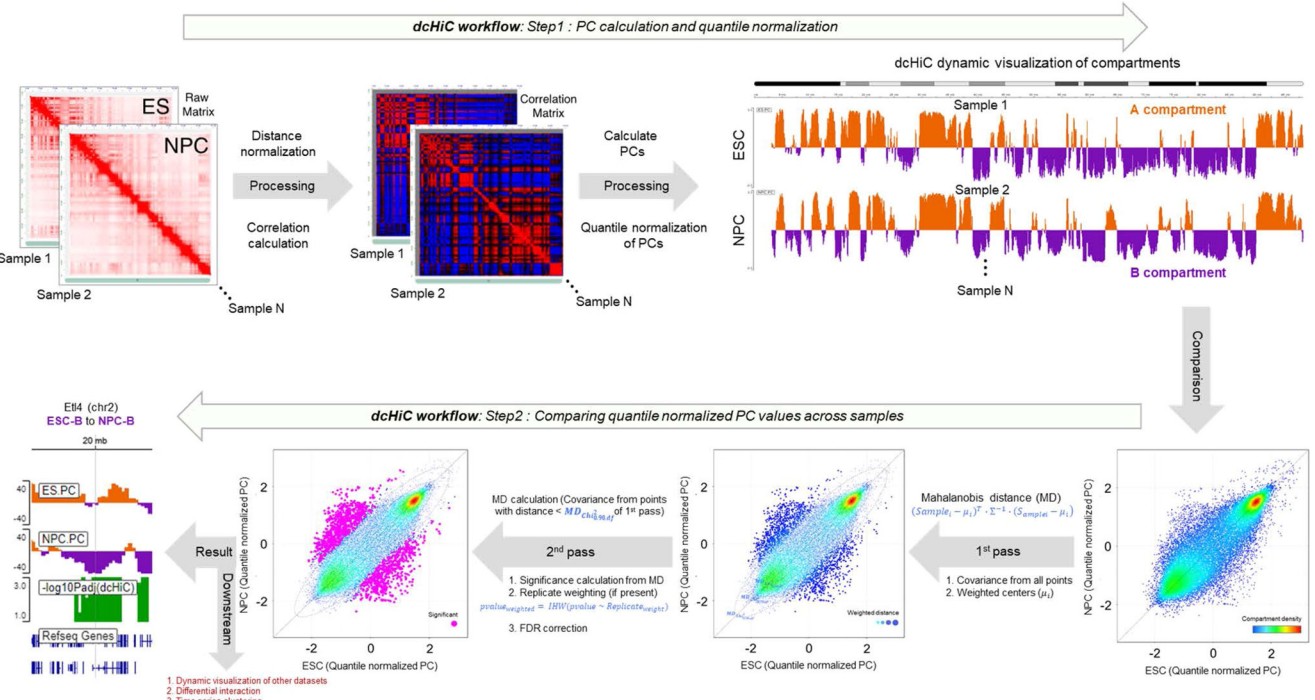

**Fig. 1 | Outline of the method.** The figure panel shows the dcHiC workflow in two steps. In step 1, dcHiC calculates the principal components followed by the quantile normalization of the compartment scores across all input Hi-C data. In step 2, for each genomic bin, dcHiC calculates the Mahalanobis distance, a multi-variate z-score that measures the extent to which each bin is an outlier with respect to the overall compartment score distribution across all input Hi-C maps (Methods - differential compartment identification). dcHiC then utilizes the Mahalanobis

distance to assign a statistical significance using the chi-square test (p-value) for each compartment bin and employs independent hypothesis weighting (IHW – when there are replicate samples) or FDR (when no replicates are available) correction on these p-values. dcHiC outputs a standalone dynamic IGV web browser view and enables the user to integrate other datasets into the same view for an integrated visualization. Source data are provided as a Source Data file.

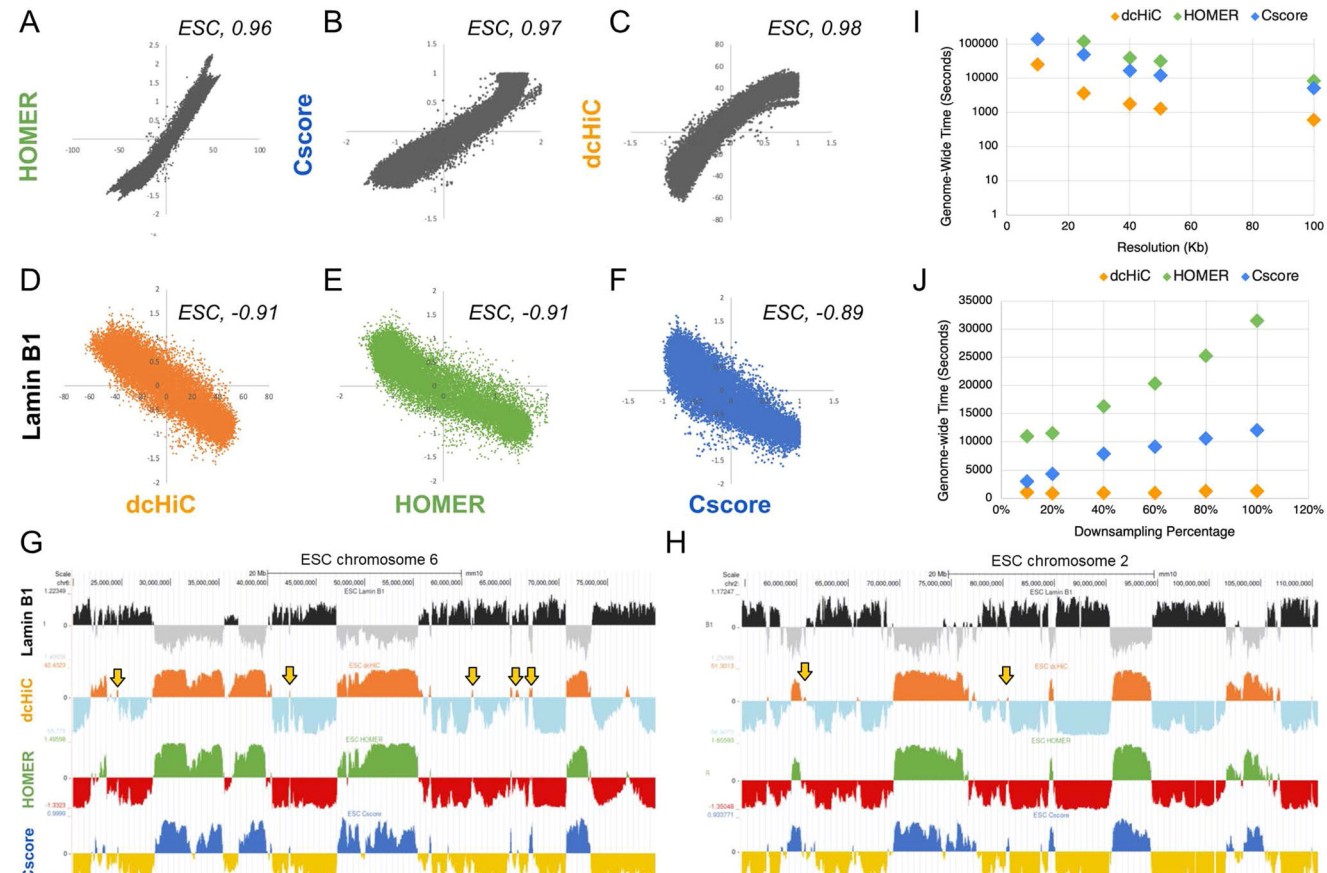

**Fig. 2 | Comparison of *dcHiC* compartment scores with HOMER compartment scores and Lamin B1 association data. A–C** Genome-wide comparison of dcHiC (orange), HOMER (green), and Cscore (blue) compartment scores against each other for mouse ESCs. **D–F** Genome-wide comparison of dcHiC, HOMER, and Cscore compartment scores against Lamin B1 profiles for mouse ESCs.
**G, H** Browser views of the compartment scores from three different methods and Lamin B1 signal in chromosomes 6 and 2 in mouse ESC. Arrows highlight a subset of regions where the compartment assignment was not consistent among the three

methods. **I** Genome-wide runtimes for compartment calling by each of the three methods at 10 Kb, 25 Kb, 40 Kb, 50 Kb, and 100 Kb resolution for mESC Hi-C data. The runtimes include a sequential run of compartment calling for each chromosome and this is repeated for two pseudo-replicates of mESC data with runtimes summed up. **J** Genome-wide runtimes for 50 kb resolution mESC Hi-C maps at 10%, 20%, 40%, 60%, 80%, and 100% down-sampling rate (100% = 500 million reads) for two pseudo-replicates (similar to Fig. 2I). Source data are provided as a Source Data file.

enrichments consistent with these cellular identities. In a ten-way comparison (*n* = 10) of major progenitor and differentiated cell types from mouse hematopoiesis, dcHiC revealed significant compartmental changes involving key genes such as *Sox6, Meis1, Runx2, Klf5*, and many others. Across both neural and hematopoietic differentiation models, our results also highlight the importance of generally ignored compartmental differences within the same compartment type (within A or within B - Fig. 1). We also demonstrate the biological significance of our differential calls through strong correlations with cell-type specific differences in lamin B1 association, histone modifications and gene expression. For human LCLs, comparing twenty Hi-C maps from a diverse set of individuals, dcHiC confirms the previous findings, with significant enrichment of various biological signals within the differential compartment regions across the population. Finally, using single-cell Hi-C data from two post-natal developing mouse brain regions across different time points, we demonstrate the utility of dcHiC in identifying compartments and in performing differential compartment analysis on pseudo-bulk data from as few as 100 cells per time point. Our results from this analysis reveal dynamic and tissue-specific compartmental differences for key genes related to synapse assembly and adult brain development that were not highlighted before.

Overall, dcHiC provides an integrative framework and an easy-to-use tool for comparative analysis of bulk and single-cell Hi-C maps and

identifies biologically relevant differences in compartmentalization across multiple cell types and/or conditions. With immediate application to hundreds of publicly available datasets, dcHiC will play an essential role in providing deeper insights into dynamic genome organization and its downstream effects.

## Results

### dcHiC identifies compartments consistent with other approaches

As more complex experimental designs emerge that compare multiple different Hi-C profiles, a comprehensive method to compare the spatial organization of the genome is necessary. To do this, dcHiC first employs a time- and memory-efficient R implementation of singular value decomposition (SVD) to achieve the eigenvalue decomposition of each Hi-C contact map[34]. This is followed by automated selection to find the principal component and its sign (reoriented if needed) that best correlates with gene density and GC content per sample (Methods – computation and quantile normalization of compartment scores for comparison). The resulting compartment scores are quantile normalized, and a multivariate score (Mahalanobis distance) is computed based on an initial covariance estimation. We then refine the null distribution by removing outliers before calculating new covariance estimates that will be used for computing the final statistical significance ($\chi^2$ test) of differences in compartmentalization (Methods –

differential compartment identification). dcHiC provides standalone browser visualization as well as several other features facilitating the interpretation of its results. Figure 1 summarizes the overall workflow of dcHiC.

To establish the validity of the dcHiC results, we first compared compartment calls to two other common compartment-finding approaches: a canonical PCA-based approach (HOMER[33]), and the CscoreTool[17], a method that uses a likelihood function over a sliding window to infer compartment scores. The resulting compartment scores were highly similar among the three methods at 100Kb resolution using mouse ESC Hi-C data, with Pearson's $r = 0.96$ between dcHiC and HOMER, 0.97 between HOMER and CscoreTool, and 0.98 between CscoreTool and dcHiC (Fig. 2A–C). Similar to A/B compartment decomposition from Hi-C data, association with the nuclear lamina (or radial position) is another strong indicator of a broad-level chromatin state with heterochromatin localizing at the periphery and euchromatin at the nucleus center. All three methods also showed strong negative correlation with Lamin B1 data, confirming the previous findings[27,36], with $R$-values of −0.91, −0.91, and −0.89 for dcHiC, HOMER, and CscoreTool, respectively (Fig. 2D–F). We further plotted the compartment scores for chromosome 2 and chromosome 6 for ESCs and NPCs from dcHiC, HOMER and CscoreTool alongside Lamin B1 association signal confirming the high concordance (Fig. 2G, H). These results established that dcHiC, similar to existing approaches, accurately captures compartment patterns. Next, we further analyzed the 4–7% of the genome that is labeled in opposite compartments by dcHiC in comparison to HOMER for ESC and NPC (Supplementary Fig. S1A, B). Overall, dcHiC-B but HOMER-A regions (~1% for ESC and NPC) showed positive lamin B1 signal and lower gene expression levels compared to dcHiC-A but HOMER-B regions (Supplementary Fig. S1C, D). The latter set (3% for ESC and 6% for NPC) had a mix of regions with positive and negative lamin association as well as gene expression values that are lower than constitutive A but higher than constitutive B compartment regions (compare to Fig. 3) suggesting a weak compartmentalization for these regions into either A or B compartment.

## Performance evaluation of compartment calling by dcHiC and other approaches

Next, we assessed the resource utilization of dcHiC against HOMER and CscoreTool for compartment calling, a prerequisite to differential compartment analysis as well as the major bottleneck for high-resolution analysis in general. We evaluated the time and memory utilization of these three methods using two mouse ESC pseudo-replicates (~500 M reads each), from which we generated contact maps at 5 different resolutions and 6 different sequencing depths (30 combinations; Supplementary Information, Table S1–4). In Fig. 2I, J, we plotted genome-wide runtimes at 100% sampling rate for 5 different resolutions and for 50 kb resolution at six different down-sampling rates showing that dcHiC runs 4–13x faster than CscoreTool and 22–33x faster than HOMER across these conditions. Across all read depths and all resolutions we tested, dcHiC ran 1.3–15x faster than CscoreTool and 10–52x faster than HOMER genome-wide (Supplementary Tables S1 and 2). Figure 2J also demonstrates that dcHiC scales better with increasing sequencing depth. With respect to memory use, at full read depth and 100 kb resolution, CscoreTool had a lower peak memory (~0.24 Gb) usage than dcHiC (~0.34 Gb) and HOMER (~1.2 Gb). For resolutions of 50 Kb, 40 Kb, and 25 Kb Hi-C data at 100% sequencing depth, all the three tools were within 30% of each other (~1.13 Gb, ~1.25 Gb, and ~1.3 Gb for CscoreTool, dcHiC, and HOMER, respectively) with CscoreTool utilizing the least amount memory for computing the compartment score at every resolution (Supplementary Tables S3, 4). For this time and memory profiling, we ran all tests genome-wide, and used one CPU per chromosome (Intel Xeon Gold 6252 CPU @ 2.10 GHz). Running HOMER genome-wide at 10Kb resolution did not finish after 100 h of compute time for ESC data.

## Pairwise differential compartment analysis of mouse neuronal differentiation

Previous studies have reported substantial compartment flips during the mouse embryonic cell (ESC) to neuronal progenitor cell (NPC) transition, a well-studied in vitro differentiation system[38,39]. These differences have been studied further using replication timing profiling, lamin B1 association mapping, and fluorescence in situ hybridization (DNA FISH)[8,32,40]. Therefore, we chose these two cell types to demonstrate dcHiC's utility in a pairwise comparison to replicate known compartment flips and identify additional significant changes that are supported by other data. This also allowed us to compare pairwise differential compartment calls from dcHiC and HOMER.

Overall, dcHiC identified 1981 100 kb bins with statistically significant differential compartmentalization (FDR < 0.1), covering up to 7.5% of the genome. For ESC and NPC, ~37% (72.8 Mb) and ~51% (101.6 Mb) of these differences involved A (active) compartment, respectively. The differential compartments are further subdivided into flipping (A->B or B->A) or matching (A->A or B->B) compartment transitions. We observed that ~74% of all the differential compartments were flips (A to B: ~30%, B to A: ~44%) during ESC to NPC transition, whereas the remaining ~26% were within matching compartments (Fig. 3A). We further classified significant changes within the same compartments (A to A or B to B) based on whether the compartment scores were higher in ESC or NPC (Fig. 3B–E). For the resulting set of six different types of differential compartments, we plotted the distributions of compartment scores (Fig. 3B), lamin B1 association (Fig. 3C), replication timing (Fig. 3D), and gene expression (Fig. 3E). As expected, more euchromatic compartments were associated with lower lamin B1 attachment, early replication timing and higher gene expression. These trends were consistent for compartment flips as well as changes within matched compartments (e.g., strong A in ESCs to weak A in NPCs).

Next, we compared the differential ESC vs NPC compartments from dcHiC to those from HOMER. HOMER reported a total of 3,042 100Kb bins with significant differential compartmentalization (FDR < 0.05) and 1,355 of these 100Kb bins overlapped with dcHiC differential calls (±1 bin slack; Fig. 3F). To compare the calls made by the two different methods, we plotted the absolute differences in laminB1 signal, replication timing and log2 gene expression values of all the reported differential compartments (Fig. 3G) or method-specific differential compartments (Fig. 3H) for each method. These results show that dcHiC differential compartments are significantly (unpaired $t$-test $p$-values < 0.05) enriched for regions with higher ESC and NPC differentials for lamin association and replication timing signals although both methods captured regions with signal differences in all three measures. We also performed differential expression analysis between ESCs and NPCs to map the differentially expressed (DE) genes (DEseq2[41], FDR < 0.05, fold change>4) on the differential compartments. We observed that dcHiC differential compartment bins were enriched in the number of DE genes (Fig. 3I) as well as the fold change (log2) and significance (DESeq2) of the difference for those DE genes (Supplementary Fig. S2A, B). Further, we also looked at the average number of histone modification peak (MACS2 $p$-value < 1e-5) differences between ESC and NPC per 100 Kb for the regions from dcHiC and HOMER's differential calls (Supplementary Fig. S3). For all three different histone marks (H3K4me1, H3K4me3, H3K27ac), we observed a higher number of peak differences per 100 Kb for dcHiC compared to HOMER. These observations imply that the differential calls made by dcHiC are accompanied by larger changes between ESCs and NPCs in other biological signals relevant to compartmentalization.

## Robustness of dcHiC differential compartment calls

Next, we sought to see how well the pairwise differential compartment calls between different Hi-C profiles are preserved through down-sampling and at different resolutions. We used the 4 ESC biological

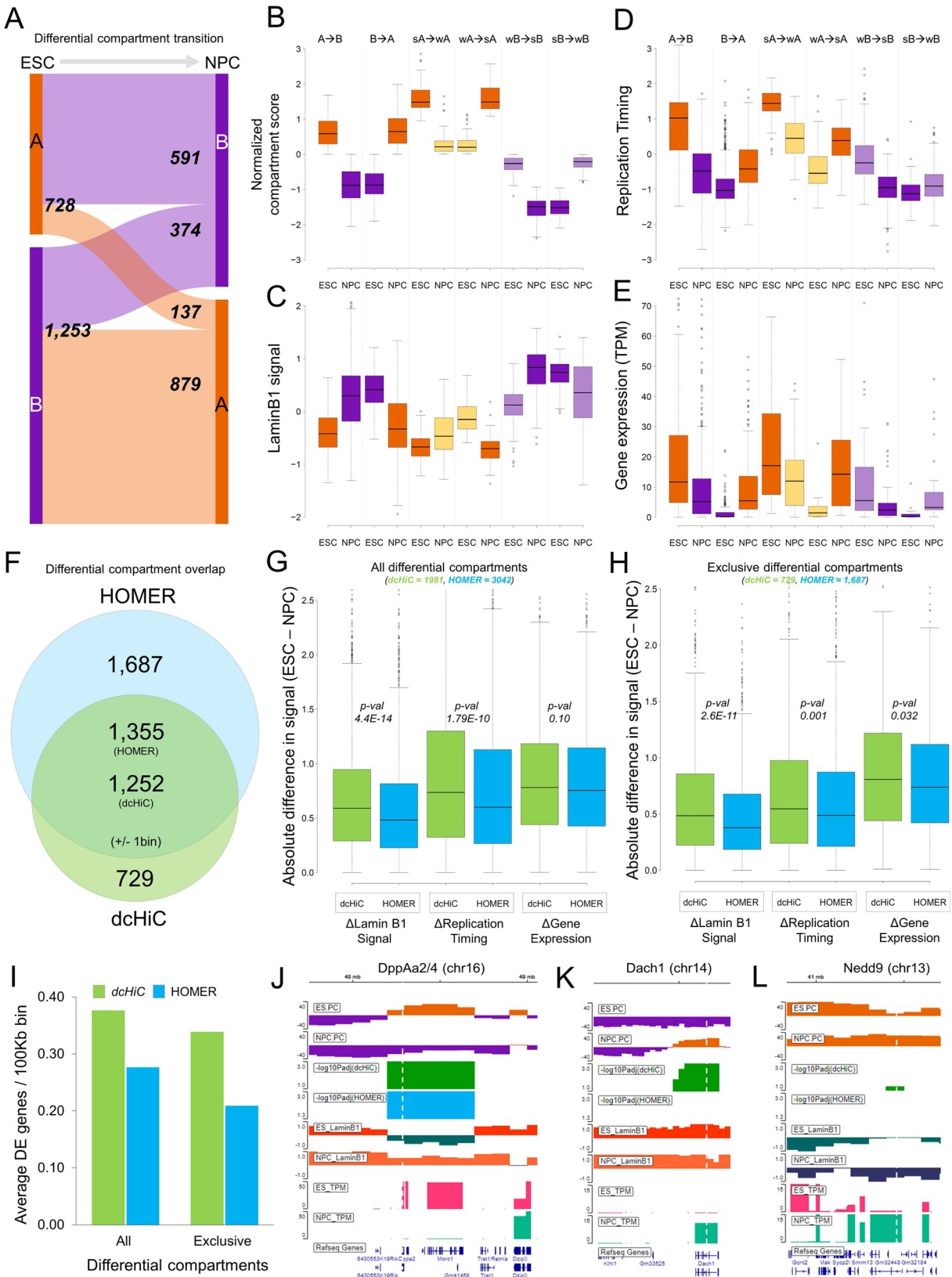

replicates (230M–1.2B reads) and the 3 NPC biological replicates (720M–1.5B reads) at their full sequencing depth (100%) and then down-sampled each replicate separately to 5 different read depths: 80%, 60%, 40%, 20%, and 10%. To profile the effects of down-sampling, we first compared ESC and NPC replicates at each read depth using 100Kb resolution contact maps. We found that there over 80% recall of

differential compartments called from full sequencing depth ("ground truth") for down-sampling rates of 40% or more (Supplementary Fig. S4). In order to also assess the role of resolution in recall of differential compartments, we repeated the same down-sampling experiments for 4 other resolutions: 50Kb, 40Kb, 25Kb, and 10Kb. We observed that except from 10 Kb resolution, all other cases were

**Fig. 3 | Pairwise differential compartment analysis between ESCs and NPCs.**
**A** The breakdown of the numbers of differential compartment calls (100 kb resolution) belonging to different types. A compartments are in orange and B compartments are in purple. **B–E** The distributions of compartment scores, replication timing, Lamin B1 signal and gene expression values for all statistically significant compartment changes identified by dcHiC. Strong (s) and weak (w) were used to indicate the relative compartment strength (or absolute value) between the two cell types. The breakdown of differential compartment calls across different subtypes with respect to compartment association were: A->B $n = 659$; B->A, $n = 819$; sA->wA, $n = 70$; wA->sA, $n = 50$; wB->sB, $n = 168$; sB->wB, $n = 222$ for each subpanel with zero or N/A values removed from each data type. **F** The Venn diagram shows the overlap between dcHiC (green) and HOMER (blue) differential compartment calls. **G, H** The absolute difference in Lamin B1, replication timing signal and gene expression values (TPM) overlapping with all and exclusive differential compartments identified by dcHiC and HOMER, respectively (statistical significance was calculated by unpaired, two-sided $t$-test). **I** The average number of differentially expressed (DE) genes overlapping with differential compartment bins (100 kb) identified by dcHiC and HOMER. **J–L** dcHiC differential compartments involving three DE genes: *DppA2/4*, *Dach1* and *Nedd9*. In **B–E**, **G**, and **H**, data are presented as mean values ±SEM. Source data are provided as a Source Data file.

similar to 100Kb where 40% down-sampling still led to a high recall (>75%), whereas for 10 Kb resolution, the results at 60% down-sampling had a recall of 80% that dropped to 61% for 40% down-sampling (Supplementary Fig. S5A–D).

To better understand the role of low sequencing depth replicates in this recall analysis, we repeated the down-sampling by leaving out the one ESC replicate with 230 M reads, which pushed the sequencing depth range to 600 M to 1.5B reads per replicate across 3 remaining ESC and 3 NPC replicates (3v3 analysis). First, at 100% sequencing depth, we recovered 1906 differential bins out of 1981 we found from the 4v3 analysis above suggesting a minimal effect of the low depth replicate for differential compartment identification. Next, we observed that over 80% recall for these 1906 bins was kept up until 20% down-sampling (~120 M reads for the lowest depth replicate; ~90% recall for 40% down-sampling) in this 3v3 analysis as opposed to 40% for 4v3 analysis. Overall, these results suggest that replicates with substantially lower sequencing depth than others may not contribute much to overall discovery power of dcHiC especially if they are sequenced below 100 M reads. Our data also shows that with replicates of at least 100 M reads, differential compartment analysis at 25Kb or lower resolution (50 kb, 100 kb) can be carried out with an acceptable recall of all compartmentalization changes that can be detected with deeper sequencing.

To show the utility of our tool in detecting differences at higher resolution, we ran dcHiC at 10Kb resolution to call differential compartments between ESC and NPC. We found a total of 16,581 10Kb bins (165.81 Mb) differential compartments between the conditions. Among the 1981 100Kb dcHiC differential bins (198.1 Mb), 72% exactly overlapped at least one 10Kb differential bin (over 86% within ±2 bins). This suggests a significant overlap across resolutions but also highlights the prevalence of regions that are detectable only at higher or lower resolution compartment analysis (Supplementary Fig. S6). We also evaluated the potential of false-positive discoveries from *dcHiC* by running it to compare replicates of the same conditions/sample. We used all four biological Hi-C replicates available for ESC in different combinations (all 1 vs 3 and 2 vs 2 combinations of splitting the replicates). When we ran dcHiC on these combinations, the number of significant compartment changes (i.e., false positives or type-1 error) ranged from 1 to 32 with a median value of 2 bins (compared to 1,981 100 kb bins when ESC was compared to NPC), suggesting a low false-positive rate for identifying differential compartments. When we ran the same analysis using 10Kb bins, we identified a median value of 751 differential bins (~0.2% of the genome), suggesting that higher resolution differential analysis may be more prone to false positives.

To further assess the type-1 error rate, we carried out a series of differential compartment analysis between mouse ESC pseudo-replicates (2 replicates) at different resolutions (100 Kb, 50 Kb, 40 Kb, 25 Kb, and 10 Kb) and down-sampling rates (100%, 80%, 60%, 405, 20% and 10% of 500 million sequencing depth). We measured the number of differential compartments when running two down-sampled replicates against each other at different resolutions and our results indicate that dcHiC is robust to type-1 error when

comparing replicates at different resolutions and read depths (Supplementary Table S5). We also evaluated the type-1 error rate, when two mouse ESC pseudo-replicate Hi-C maps of different sequencing depth are compared by dcHiC. Across the 21 comparisons, we first see that the compartment calls are highly correlated within 100% to 40% (500M-200M reads) of read depth (Supplementary Table S6). The correlations with high read depth samples drop substantially for 20% (100 M reads) and further for 10% (50 M reads) sample. We noticed this occurred because compartment scores for some chromosomes started to not fully reflect the compartmentalization pattern at lower read depths. Removing the 5 chromosomes (chr 4, 5, 14, 17, X) with such issues, we see correlations at lower read depths improve, however not to the point that we highly concordant (correlation >0.9) compartment calls between two pseudo-replicates (Supplementary Table S7). While evaluating the false-positive calls, we first observed that correlations between compartment scores are closely related to the number of differential calls. When we utilized dcHiC to find differential compartments (i.e., false-positive calls) between two replicates of different sequencing depth by down-sampling Hi-C maps at 100Kb resolution, we see no false positives among samples with 100% to 60% down-sampling (500M-300M reads) (Supplementary Table S8). We also do not obtain any false positives even for lower depth samples when they are compared against the sample with the same rate of down-sampling. However, a substantial number of false-positive differential calls appear when 20% or 10% down-sampled samples are compared to higher depth samples (Supplementary Table S8). Like compartmental correlations, here also when we filter out the 5 chromosomes with issues in compartment calls at low read depths, we see that the false-positive rates dramatically improve for 40% and for 20% down-sampled samples (Supplementary Table S9). Based on these results, we believe compartment scores and differential compartment calls are robust when comparing Hi-C maps that are sufficiently sequenced (100 M or more reads) and are within 2–3-fold read depth of each other.

Example genes from ESC vs NPC differential compartments: Within dcHiC's calls, we also analyzed a set of key genes known for their critical role in ESC or NPC state that have been studied extensively for changes in their nuclear organization during the transition. For instance, we analyzed a set of genes for which fluorescence in situ hybridization (FISH) experiments were performed to study changes in radial positioning during the ESC to NPC transition. These included pluripotency markers specifically expressed in ESCs (e.g., *Zfp42/Rex1* and *Dppa2/4*) as well as EPH Receptor B1 (*Ephb1*) and other marker genes specific to neuronal differentiation. Figure 3J shows the *Dppa2/4* region in mouse chromosome 16 that is shown to change radial positioning, chromatin state, lamin B1 association and replication timing during differentiation[40,42]. Consistent with these data, both dcHiC and HOMER reported a significant shift from the A (active) to the B (inactive) compartment during mouse ESC to NPC differentiation (Fig. 3J). In addition, dcHiC reported significant compartment changes for several other important genes that HOMER missed. Figure 3K, L displays two genes, namely, Dach1 and Nedd9, which are known to play a critical role in organogenesis and signal transduction pathways for mouse neuronal development[43,44]. We also detected these genes in our

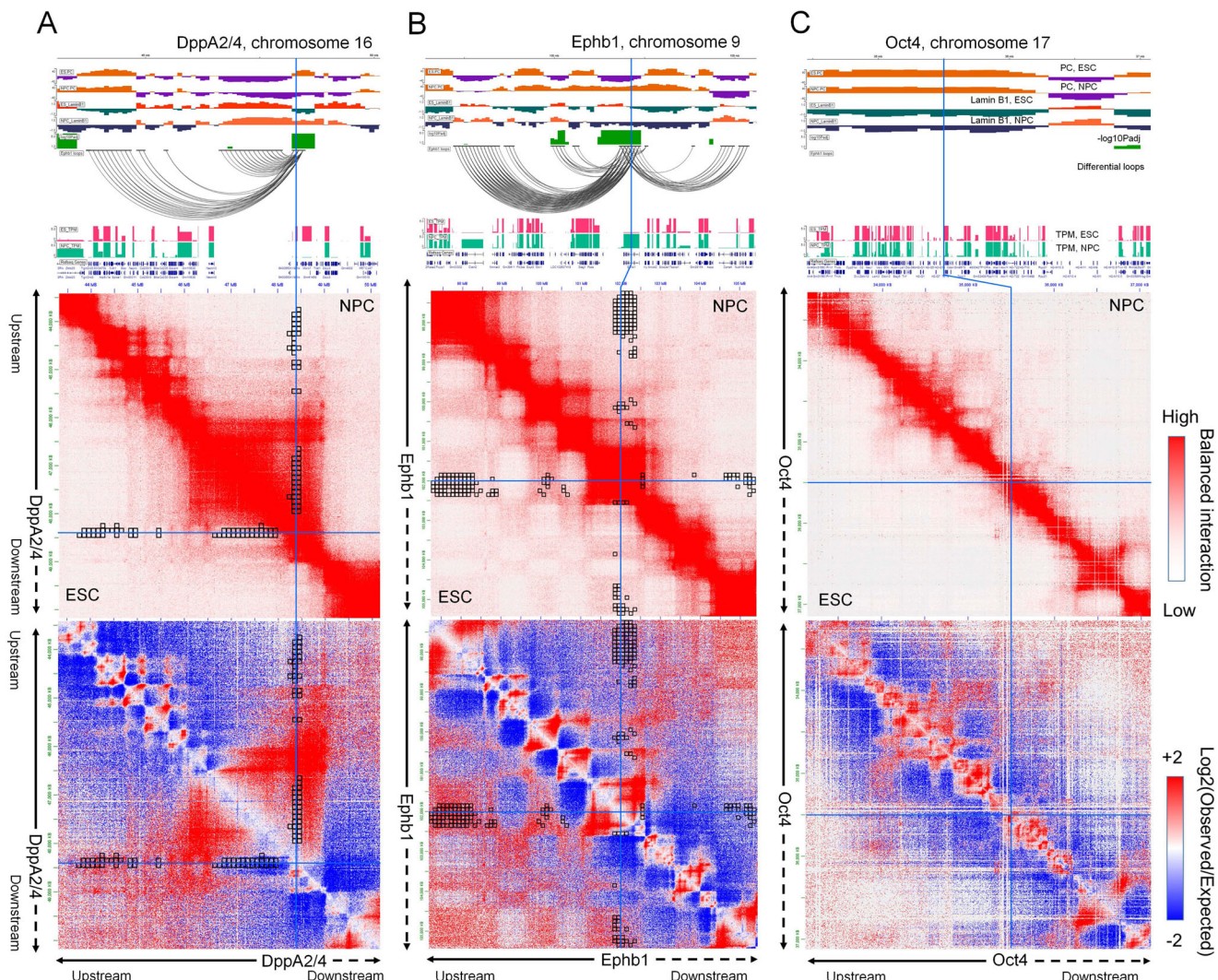

**Fig. 4 | Differences in local chromatin interactions of differential compartments.** Detailed browser views (top), Hi-C contact maps (mid) and differential chromatin interactions (bottom) of three gene loci – **A** *DppA2/4*, **B** *Epbh1*, and **C** *Oct4*. Visible changes in interactions involving the *Dppa2/4* locus and *Ephb1* locus are highlighted through each plot. Although *Oct4* shows a dramatic change in gene expression, the region does not alter its radial position within the nucleus (FISH experiments), which is also consistent with the lack of change in compartmentalization reported by dcHiC. Source data are provided as a Source Data file.

differential gene expression analysis of ESCs vs NPCs as significantly upregulated in NPCs (FDR < 0.05; >160x for Dach1 and >30x for Nedd9). Dach1 lies in a compartment reported to be flipped from ESC-B to NPC-A by dcHiC (Fig. 3K). Nedd9 gene overlapped with the A compartment in both cell types but with stronger compartmentalization in NPC, which was detected as a significant change by dcHiC (Fig. 3L).

To determine whether the compartmental changes are accompanied by specific differences in local chromatin interactions, we implemented an extension of our comparative approach to identify differences in contact counts involving the differential compartments (Methods – differential interaction identification). This feature allows users to input a set of significant chromatin interactions (e.g., from Fit-Hi-C[45]) or chromatin loops (e.g., from HiCCUPS or Mustache), which will then be filtered for their overlap with differential compartments and tested for their difference across the compared conditions. The black square boxes in Fig. 4A represent the dcHiC-identified differential interactions (ESC vs NPC) that are anchored in the *Dppa2/4* region. These interactions are identified among FitHiC2 calls[46] (FDR < 0.05) that are reported as significant in at least one replicate of ESC and/or NPC datasets. The results show that the *Dppa2/4* domain in NPC specifically

interacts with its upstream region compared to ESC, while the interactions with the adjacent downstream region remained unchanged, a change that can be visualized on the Hi-C map (Fig. 4A). Previous studies on *Ephb1* have demonstrated significant subnuclear repositioning of the gene from the periphery to the nuclear center during ESC to NPC differentiation[40] accompanied by higher gene expression later. A similar analysis of the *Ephb1* region shows that it has enriched interactions with a pair of upstream B compartments in ESCs, which are weakened in NPCs where *Ephb1* is transitioned to the A compartment (Fig. 4B). In addition, the same region gained interactions with a downstream A compartment in NPC. These results highlight the value of differential interaction analysis coupled with differential compartmentalization to better delineate important changes in the local chromatin environment. Finally, even though the above examples highlight cases where gene expression is tightly correlated with compartment changes and radial positioning, this is not necessarily the case for all genes. Figure 4C shows the pluripotency marker gene *Pou5f1/Oct4* region with ESC-specific gene expression. The radial positioning of this gene locus was shown to remain unchanged during the ESC to NPC transition[40], consistent with our results (Fig. 4C). Overall, dcHiC identified both known compartment flips (A to B or B to A) as well as

compartmentalization differences within the same compartment for important genes.

## Differential compartments are associated with sub-compartment transitions during ESC to NPC lineage differentiation

Recent studies have shown that beyond open and closed chromatin, genome activity encompasses multiple states of compartmentalization which can be captured via a more refined sub-compartment analysis[14,15,47]. Therefore, we hypothesized that differential compartments identified by dcHiC, whether they involve compartment flips or not, should also be associated with changes in sub-compartments between conditions. To compare the changes in sub-compartments with differential compartments, we mapped the dcHiC differential calls on the 'Calder'[47] derived sub-compartments within mouse ESC and NPC cell lines. The Calder algorithm infers a complete hierarchy of compartment domains using intrachromosomal interactions and classifies each A/B compartment into 4 sub-compartments each (8 in total; A/B.1.1, A/B.1.2, A/B.2.1, A/B.2.2) adopting a more nuanced representation of the two primary compartment classes. We applied Calder on ESC and NPC Hi-C maps separately and retrieved a total of 24,546 100Kb bins (-2.4 GB) with sub-compartment assignments for both ESC and NPC. For these bins, we then assessed the overlap of differential calls from dcHiC with the differences in sub-compartment labels. Out of 1,981 dcHiC bins, for 1,862 we had Calder labels on both cell types and among those 97.5% (1,816 bins) overlapped with differential sub-compartment labels. For the remaining 22,684 bins with Calder labels that do not overlap with dcHiC differential calls, still a high but smaller percentage (57.5%) corresponded to differences in sub-compartment labels. Supplementary Fig. S7A, B shows the total number of differential compartment transitions, grouped based on their sub-compartment classes within ESC and NPC lineages. These results highlight that nearly all dcHiC differential compartments have underlying changes in sub-compartment assignments consistent with our initial hypothesis. In terms of being able to do a differential analysis directly from sub-compartments, however, a large percentage (-60.5% or 14,866 out of 24,546 100 kb bins) of sub-compartment transitions/flips suggest that this approach may lead to low specificity in detecting important differences and would need to be coupled with additional filters and/or supplemented by further statistical assessments.

To better understand the type of sub-compartment flips that are overrepresented in dcHiC calls, we compared the transition probabilities among sub-compartment labels (ESC vs NPC) obtained from dcHiC differential calls versus non-differential regions (Supplementary Fig. S7C, D). The fold-change values show that dcHiC differential calls are significantly enriched for sub-compartment transitions with a distance of 3 or more in the sub-compartment hierarchy (e.g., A.1.1 to A.2.2 (distance of 3) or A.1.1 to B.1.1 (distance of 4)) supporting the strong compartmentalization change of these bins (Supplementary Fig. S7C, D). We observed highly enriched transitions from ESC-A subcompartments to strong NPC-B subcompartments (B.2.1 and B.2.2) that corresponded to substantial reduction in the transcriptional activity of overlapping genes going from ESC to NPC (Supplementary Fig. S7E). An example of such sub-compartment transition was the 145–148 Mb region in chromosome 4 encompassing 71 unique genes (Supplementary Fig. S7F). This locus harbored genes with known functions including pluripotency (*Rex2*) and migration and invasion inhibition (*Miip*)[48].

Although the broad classification of A and B compartments is likely insufficient to capture the multistate genomic activity across cell lines, our sub-compartment analysis suggested that differential analysis using compartment scores is able to effectively capture changes involving sub-compartments with biological significance. Sub-compartment inferring algorithms such as Calder[47] provide a useful approach to decipher the underlying epigenetic and transcriptional

heterogeneity within tissue types, differentiation stages and other conditions but are not directly applicable for the task of de novo detection of compartmentalization changes across samples due to a large number of transitions involving sub-compartment types that are very similar (distance of 1 or 2).

## Multicell-type differential compartment analysis of the mouse neuronal system

The same in vitro system used to differentiate from ESCs to NPCs also allows further differentiation of NPCs to cortical neurons or CNs[39]. This developmental lineage provides an approach to demonstrate how dcHiC uses a multivariate distance measure to compare the compartmentalization of more than two cell types simultaneously. For such multiway comparisons, dcHiC provides a quick and straightforward approach to detect outliers in compartment scores and associated differential interactions, an approach far easier with many experiments than the traditional paradigm of taking pairwise comparisons. In this section, we first illustrate the biological significance of dcHiC's differential compartments using multiple lines of biological data. We then demonstrate functional term enrichments and show specific differential genes that illustrate the application's breadth of analysis.

Applying dcHiC at 100 kb resolution to intrachromosomal Hi-C data from ESC, NPC, and CN samples, we identified a total of 5055 significant differential bins covering approximately 19.2% of the genome. Compartments A and B were evenly split for NPC and CN, whereas ESC had ~63% B compartments. Overall, regions in the B compartment for each cell type were more likely to exhibit statistically significant compartment changes compared to the A compartment (21–23% vs 16-18%). Figure 5A summarizes the number of differential compartment bins that involve flips (A->B or B->A) or remained within the same compartment throughput the lineage transition. Consistent with the literature[2,5,49], we showed that compartmental dynamics are strongly associated with the variability of gene expression and histone modifications (Fig. 5B).

To further analyze these changes simultaneously, rather than one transition (or pair) at a time, we utilized time-series analysis to cluster the compartmentalization score patterns of these differential bins across (Fig. 5C) and plotted the expression pattern of the overlapping genes in each cluster across three different time-points. To focus on relative changes in compartmentalization, we further z-transformed the quantile normalized PCA scores for each 100 kb bin across the three cell types and applied TC-seq[50] to identify 6 major clusters. Two major clusters corresponded to regions that progressively became more euchromatic (clusters 1 and 6), and one corresponded to more heterochromatic regions (cluster 4). We observed other clusters that corresponded to one cell type showing highly different compartmentalization with respect to the other two (e.g., clusters 3 and 5 with NPC-specific patterns). To link these compartmentalization patterns to gene function, we identified genes overlapping with each differential compartment bin for each cluster. Performing functional enrichment analysis on these gene sets[51], we identified signatures that are consistent with the cellular identity of the cell type with the highest compartment z-scores (i.e., more euchromatic). For instance, for the genes overlapping with clusters 1 and 6 with compartment scores increasing from ESC to NPC to CN, the enriched terms included neurogenesis and neuronal development (Fig. 5D). For cluster 3, where CN compartment scores were highest, the enriched terms (cell-cell adhesion, biological adhesion, and others) were consistent with a general pattern for genes involved in regulating cell-type specific migration and development. We also observed that cluster 3 overlapped with an important class of gene family known as protocadherins[52]. Protocadherins are highly conserved genes across species, and most of them are clustered in a single genomic locus in vertebrates[53]. They are shown to be differentially expressed in individual neurons and involved in diverse neurodevelopmental processes[54]. When we repeated the

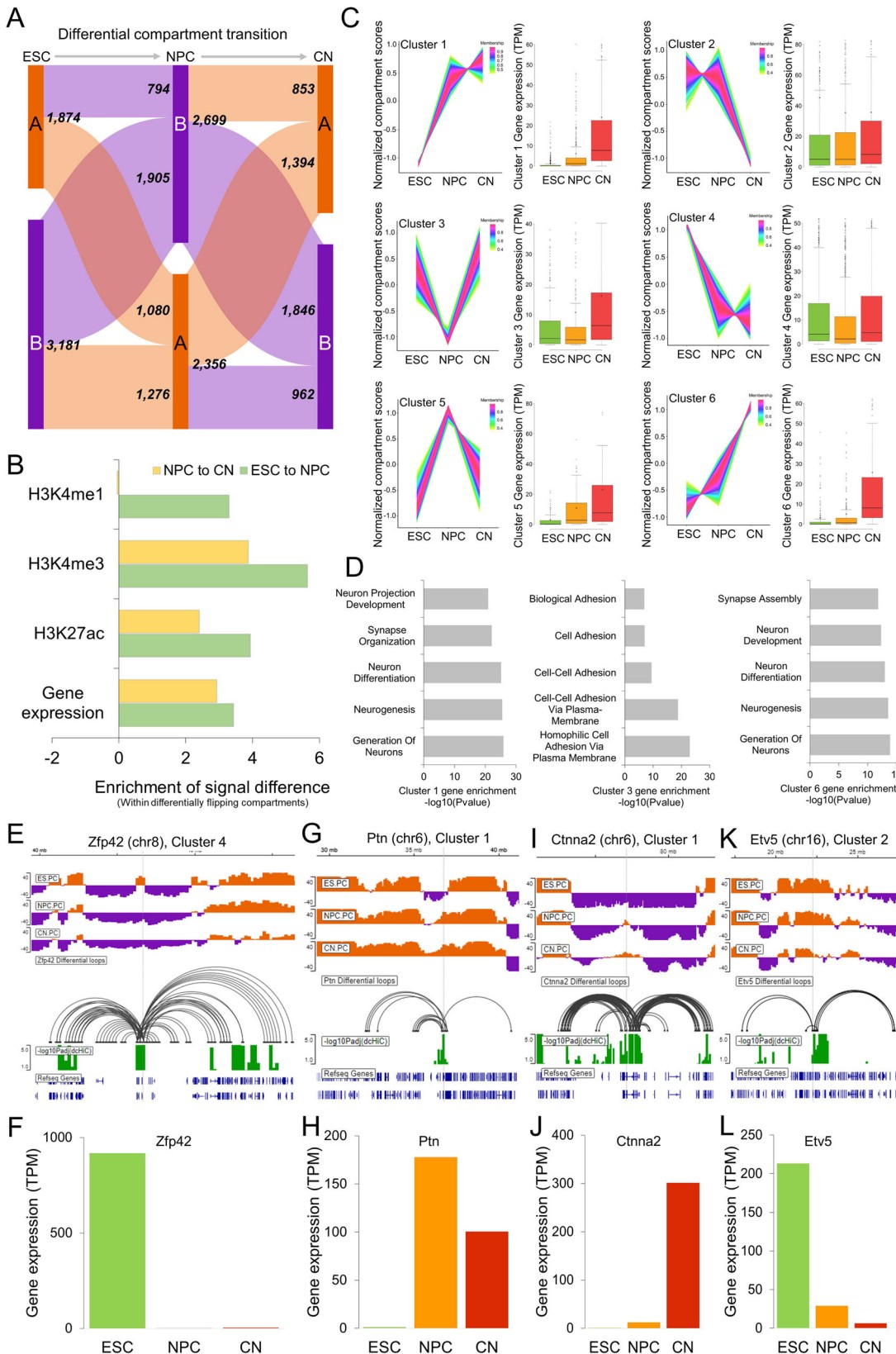

functional enrichment analysis per cell type using genes overlapping A compartments with the highest compartmentalization score for that cell type compared to the other two, we also observed cellular identity-related annotation terms (Supplementary Data 1). While annotations related to cell adhesion were enriched in ESCs as well as CN, CN specifically showed enrichment for neurogenesis, neuron differentiation

and development (Supplementary Data 1). CN, but not NPC, also showed enrichment for synaptic signaling, synapse organization and neuron projection development, potentially related to its further differentiated state with respect to NPC.

Example genes from ESC-NPC-CN differential compartments: The differential compartments captured by dcHiC encompass a variety of

**Fig. 5 | Three-way differential compartment analysis of ESCs, NPCs, and CNs.** **A** The breakdown of the numbers of differential compartment calls belonging to different types (ESC green, NPC orange, CN red). **B** The enrichment of signal differences in different histone marks and gene expression in dcHiC differential bins with compartment flips (A->B or B->A) compared to bins with nonsignificant compartment flips. **C** Time-series clustering of normalized compartment scores into six different clusters (n = 548, 972, 369, 1809, 93, 257, respectively) from the three cell types along with their overlapping gene expression profile. For the clustering analysis, the quantile normalized PCA scores for each 100 kb bin across ESC-NPC-CN were further z-transformed to focus on relative changes in compartmentalization. Data are presented as mean values ±SEM. **D** Gene term enrichment results of GO biological functions from genes overlapping with clusters 1, 3, and 6 compartments. **E–L** Differential compartments overlapping with representative genes in each of the cell types shown along with differential chromatin interactions involving respective compartments and gene expression values (TPM) across the ESC-NPC-CN transition for four example genes, each representing one cluster pattern. Source data are provided as a Source Data file.

---

traditionally studied as well as more nuanced scenarios. For instance, similar to *Dppa2/4*, *Zfp42/Rex1* is a well-studied pluripotency marker primarily expressed in undifferentiated stem cells (Fig. 5E). As is the case for *Dppa2/4*, *Zfp42* is also in a small A compartment region surrounded by large stretches of B compartments in ESCs. As expected, this region flipped into the B compartment in NPC and stayed in CN, consistent with the lack of gene expression in these two cell types (Fig. 5F). *Ptn* or pleiotrophin, on the other hand, exhibits mitogenic and trophic effects on dopaminergic neurons and is instead a marker gene for neuronal lineage. dcHiC reported this gene in a differential compartment that is B in ESC but A in NPC and CN, in concordance with gene expression (Fig. 5G, H), which fits the compartmentalization pattern of cluster 1 (Fig. 5C). These two examples represent strong compartment flips from A to B or B to A. An example of a more gradual compartmental change is the CN-specific *Ctnna2* gene, which functions as a linker between cadherin adhesion receptors and the cytoskeleton to regulate cell-cell adhesion and differentiation in the nervous system. The B compartment encompassing *Ctnna2* in ESCs gradually weakens during the ESC-NPC-CN transition, leading to a transcription-permissive A compartment that starts in NPCs and expands further in CNs (Fig. 5I, J).

Compartment shifts within the same compartment are also captured by dcHiC (Fig. 5K, L). *Etv5* encodes a transcription factor that plays an important role in the segregation between epiblast and primitive endoderm specification during ESC differentiation[55]. *Etv5* is highly expressed in ESCs but gradually loses its expression (Fig. 5L) as well as strong compartmentalization during the ESC-NPC-CN transition while remaining in the A compartment at all times. This locus belongs to cluster 2 with enrichment for more euchromatic association specifically in ESCs, consistent with the highest expression for *Etv5* for this cell type. Beyond *Etv5*, we also found a list of 179 other genes (199 bins) within the A compartment throughout the ESC-NPC-CN transition, for which the variation in the expression profile strongly correlated with changes in compartmentalization (Pearson correlation > 0.7; Supplementary Data 2). A similar analysis within differential B compartments revealed 190 genes (245 bins) with a strong positive correlation between expression and compartmentalization change (Supplementary Data 2). Overall, our results demonstrate that dcHiC can comprehensively analyze multiple different Hi-C maps simultaneously and identify compartmental changes involving abrupt (e.g., compartment flips) as well as gradual changes.

## Differential compartment analysis of the mouse hematopoietic system

The hematopoietic system is a developmentally regulated and well-characterized cell differentiation model[56,57]. This system provides an opportunity to understand the dynamic changes in chromatin structure together with transcriptional and other epigenetic changes during differentiation in detail. The study of genome organization changes during this complex process—involving many different progenitors and differentiated cell types—requires a systematic approach. A recent study by Zhang et al.[33] profiled chromatin organization in a classic hematopoietic model with ten primary stem, progenitor, and terminally differentiated cell populations from mouse bone marrow (Fig. 6A). In this model, long-term hematopoietic stem cells (LT-HSCs) represent the starting point of the hematopoietic hierarchy with self-renewal and multilineage differentiation capability. LT-HSCs first differentiate into short-term hematopoietic stem cells (ST-HSCs) and then multipotent progenitor cells (MPPs). MPP cells differentiate into either common lymphoid progenitor (CLP) or common myeloid progenitor (CMP) cells. CMPs then further branch out into granulocyte-macrophage progenitors (GMPs) and megakaryocyte-erythrocyte progenitors (MEPs). The GMP cells are then terminally differentiated into granulocytes (GR), while MEP cells are further differentiated into megakaryocyte progenitors (MKP) and then terminally differentiated into megakaryocytes (MK).

Using the Hi-C data from this system, we carried out multivariate differential analysis using dcHiC at 100 kb resolution. We detected a total of 6,061 (60.61 Mb of the genome) differential compartment bins across the ten cell types encompassing many of the genomic regions previously shown to undergo hematopoiesis-related dynamic changes[33]. Figure 6A shows an overall summary of the significant compartment changes identified by dcHiC across these cell types. We observed that the number of A to B transitions continued to increase from the LT-HSC stage to the MEP and GMP progenitor stages. The differentiation of CMP into MEP and GMP cells represents two of the most frequent A to B transitions (~27.4% and ~15.7% A->B transition, respectively) within the hematopoietic hierarchy, likely reflecting the need for suppression of certain transcriptional profiles for commitment into each branch. This is consistent with the largest proportion of differential B compartments in MEP (~46.5%) and GMP (~42%) compared to all other cell types. With respect to the top of the hematopoietic tree (i.e., LT-HSC), early progenitors such as MPP have 571 100 kb bins with a significant compartment flip (either A to B or B to A), whereas the differentiated cells such as MK and GR had 949 and 1212 such bins, respectively. This confirms the gradual divergence of chromatin compartmentalization from hematopoietic stem cells as cells progress further into differentiation.

Next, similar to the ESC-NPC-CN transition, we also carried out functional enrichment analysis of differential regions with the highest A compartment score in each group and specific cell type. Figure 6B–E shows these enrichments for four different stages of hematopoiesis (pre-bifurcation stage: LT-HSC, ST-HSC, progenitor stage: MPP, CMP, granulocyte branch: GMP, GR and the terminally differentiated granulocytes or GR) with respect to the rest and for a specific cell type within each of these stages highlighting biologically relevant processes in each case. For example, morphogenesis- and development-related biological processes were enriched in the overall pre-bifurcation stage (set of genes with the highest A compartment score in either LT-HSC or ST-HSC; Fig. 6B), and progenitor stage cells were enriched in morphogenesis-, adhesion- and migration-related terms (Fig. 6C). The granulocyte branch (GMP and GR) as well as the terminally differentiated granulocytes (GR) showed significant enrichment related to the activation and regulation of neutrophils and granulocytes (Fig. 6D, E). For the megakaryocyte branch (MEP, MKP, MK), however, we did not observe any statistically significant GO term biological process enrichment.

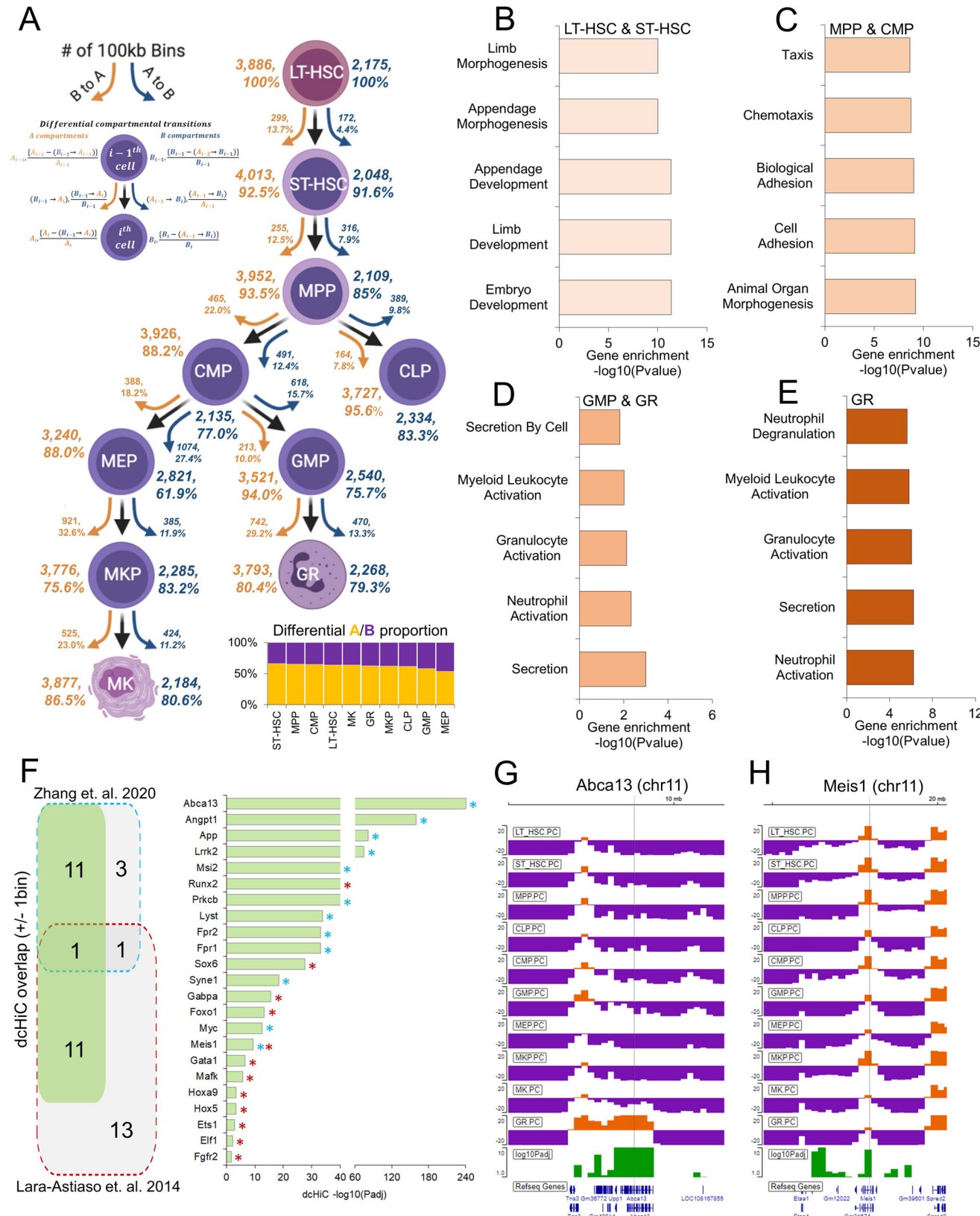

Example genes from mouse hematopoiesis differential compartments: After investigating the significance of the differential compartments from a high level, we examined the genes overlapping with the differential compartments involved in hematopoietic lineage differentiation and chromatin dynamics[58]. Figure 6F shows a set of important genes overlapping with differential compartments from

our multivariate analysis. Zhang et al. showed that increased gene-body associating domain (GAD) scores are linked to active transcription and indicate cell-type specific features. We identified 12 out of 16 such differential GAD genes between ST-HSC and GR as part of dcHiC differential compartments identified across the system (FDR < 0.1; Fig. 6F, marked by cyan stars). In addition, a previous

**Fig. 6 | Ten-way multivariate differential compartment analysis of mouse hematopoiesis. A** Summary of overall compartment decomposition and significant compartment changes observed across the 10 cell types. The orange and blue arrows represent A to B and B to A compartment flips, respectively. The numbers next to the arrows represent the total number of flipping compartments, and the numbers within the parentheses next to arrows show the significantly different flipping compartments. The bottom-right plot shows the proportion of A and B bins among dcHiC differential compartments for each cell type. Figure adopted from Zhang et al.[33]. **B–E** The functional enrichment of genes overlapping with differential compartments from 10-way comparison that have the strongest A compartment scores in either **B** LT-HSC or ST-HSC, **C** MPP or CMP, **D** GMP or GR, or **E** GR alone. **F** Differential compartments identified by dcHiC overlap a set of critical genes previously known to play a role in mouse hematopoiesis (blue/red boxes and stars indicate genes and their source). **G** An IGV browser snapshot of the *Abca13* gene and its overlapping differential compartment across the 10 cell types. The *Abca13* gene is exclusively found to be a part of the A compartment in GR. **H** An IGV browser view surrounding the *Meis1* genic region. This region overlaps with the A compartment for all cell types but with varying magnitudes of strength. dcHiC captured this region as a differential compartment. Source data are provided as a Source Data file.

analysis by Lara-Astiaso et. al.[58] also reported a set of critical genes for hematopoietic lineage differentiation. We identified 12 of these 26 genes within differential compartments (FDR < 0.1; Fig. 6F, marked by red stars), supporting dcHiC's ability to detect changes in regions harboring genes that are dynamically regulated during hematopoiesis. Among these genes, one example is the transmembrane transporter gene *Abca13*, which was the exclusive differential A compartment within GR but in the B compartment for all other cell types (Fig. 6G). Other notable examples include *Meis1*, a transcription factor required to maintain hematopoiesis under stress and over the long term[59]. Notably, this particular example was a significant change solely within the A compartment (Fig. 6H). Apart from *Meis1*, dcHiC also detected differences for other transcription factors, such as *Runx2* and *Sox6*, that are essential for progenitor cell differentiation (Fig. 6F)[60,61]. We also identified *Myc*, known for its role in balancing hematopoietic stem cell self-renewal and differentiation[62] adjacent to a significant change within the A compartment that encompasses the *Pvt1* gene. The long noncoding RNA *Pvt1* harbors intronic enhancers that interact with *Myc* and promote *Myc* expression during tumorigenesis[63]. Overall, this complex system demonstrates the utility of dcHiC's multivariate compartment analysis, which discovers important changes in compartmentalization without requiring a large number of pairwise comparisons.

Further, we have performed time-series analysis of the differential compartments on the Long-Term Hematopoietic stem cells (LT-HSC) to Granulocytes (GR) (6 time-points) and LT-HSC to Megakaryocytes (MK) (7 time-points) lineage differentiation separately. For LT-HSC to GR differentiation, the first 3 clusters show a general pattern of differential compartments with decreasing genomic activity while the last 3 shows an increase (Supplementary Fig. S8A). The functional enrichments for genes within each cluster involved general terms such as 'morphogenesis', 'development' and 'organization' (Supplementary Fig. S8B). When we repeat the same analysis for LT-HSC to MK lineage differentiation, we observed more nuanced patters involving four clusters with distinct signatures in MEPs (Supplementary Fig. S9A). For these clusters (Cluster 1, 2, 4, and 5), the change in compartment score is most prominent at the MEP stage and is generally prominent after this stage. We believe this is due to the unique condensed chromosomal organization observed in MEP stage along with MKs[33]. This previous study proposed that in these cell types, there is a reduction in long-range chromatin interactions, which resembles the condensed chromosome structures found in mitotic metaphase cells[33]. We believe the time-series analysis of differential compartments from dcHiC thus captured this feature of MEPs while also capturing two clusters (cluster 3 and 6) with gradual increase or decrease in their compartment scores (Supplementary Fig. S9A). The functional enrichment of genes in each cluster again involve general terms such as 'morphogenesis', 'development' and 'differentiation' (Supplementary Fig. S9B).

**Multiway differential compartment analysis across human-derived cell lines**

Measuring the extent to which genetic variation across individuals influences chromatin features, including 3D organization, has significant implications in our understanding of human disease. Previous studies have revealed that the presence of variations such as quantitative trait loci (QTLs) can affect histone modifications, transcription factor binding, and enhancer activity across populations[64,65]. More recent work by Gorkin et al[66]. studied variation in chromatin conformation across individuals from different human populations. Using dilution Hi-C, they profiled lymphoblastoid cell lines (LCLs) derived from 13 Yoruban individuals, one Puerto Rican trio, one Han Chinese trio, and one European LCL (GM12878). They measured significant differences in 3D genome organization across individuals using different metrics, including the Directionality Index (DI), Insulation Score (INS), Frequently Interacting REgions (FIREs), and compartment scores[66]. The study also carried out differential analysis of compartments across individuals and provided both compartment scores and "variable regions" at 40 kb resolution (after excluding chromosomes 1, 9, 14, 19 and X). To minimize technical variation and ensure a fair comparison, we started directly from the 40 Kb compartment scores reported by Gorkin et al. and ran dcHiC on these values (starting from quantile normalization). dcHiC allows direct utilization of precomputed compartment scores, such as in this case, when available.

The Venn diagram (Fig. 7A) of differential compartments from dcHiC and Gorkin et al. using the same set of 40 kb genomic bins shows a large overlap between the methods. A large fraction of dcHiC calls (7524 out of 7,876 or ~96%) were also reported by the original paper. However, Gorkin et al. reported an additional 765 Mb of the human genome as variable compartment regions (Additional_file_4.xlsx from the original publication filtered for phenotype=PC1 and discover_set=20 LCLs), which amounts to ~11 K more bins at 40Kb resolution. To further study the overlap and differences between the two approaches, we plotted two statistical significance score distributions (-log10 of the adjusted p-value calculated by Gorkin et al.) for regions that the Gorkin study reported as differential, one with regions overlapping with dcHiC calls and the other with non-overlapping regions (Fig. 7B). Variable compartments from the previous study that were not deemed significant by dcHiC have substantially lower statistical significance, as computed by the original paper suggesting dcHiC calls are enriched for stronger differences. Next, we compared the full set of differential compartments called by both methods and their fraction covering each individual chromosome (Fig. 7C). The figure shows that Gorkin et al. calls cover a larger fraction of smaller chromosomes, with more than half the entire length reported as variable compartments for some chromosomes (e.g., chr18). dcHiC, on the other hand, has a more uniform representation of differential compartments across chromosomes, with differential fractions ranging between 10% and 20% for most chromosomes. Finally, we compared the top 5000 differential compartment bins ranked by their significance scores from each approach. Figure 7D shows that ~61% of these top 5000 differential bins are identical, suggesting substantial differences in each approach's ranking with respect to statistical significance (Spearman rank correlation of 0.55). Although the ranking is substantially different between the methods, the overlapping portion of the differential compartments were enriched for higher statistical significances in terms of their differences (Fig. 7E). Using other chromatin organization metrics that were deemed variable across individuals by the Gorkin paper, we observed

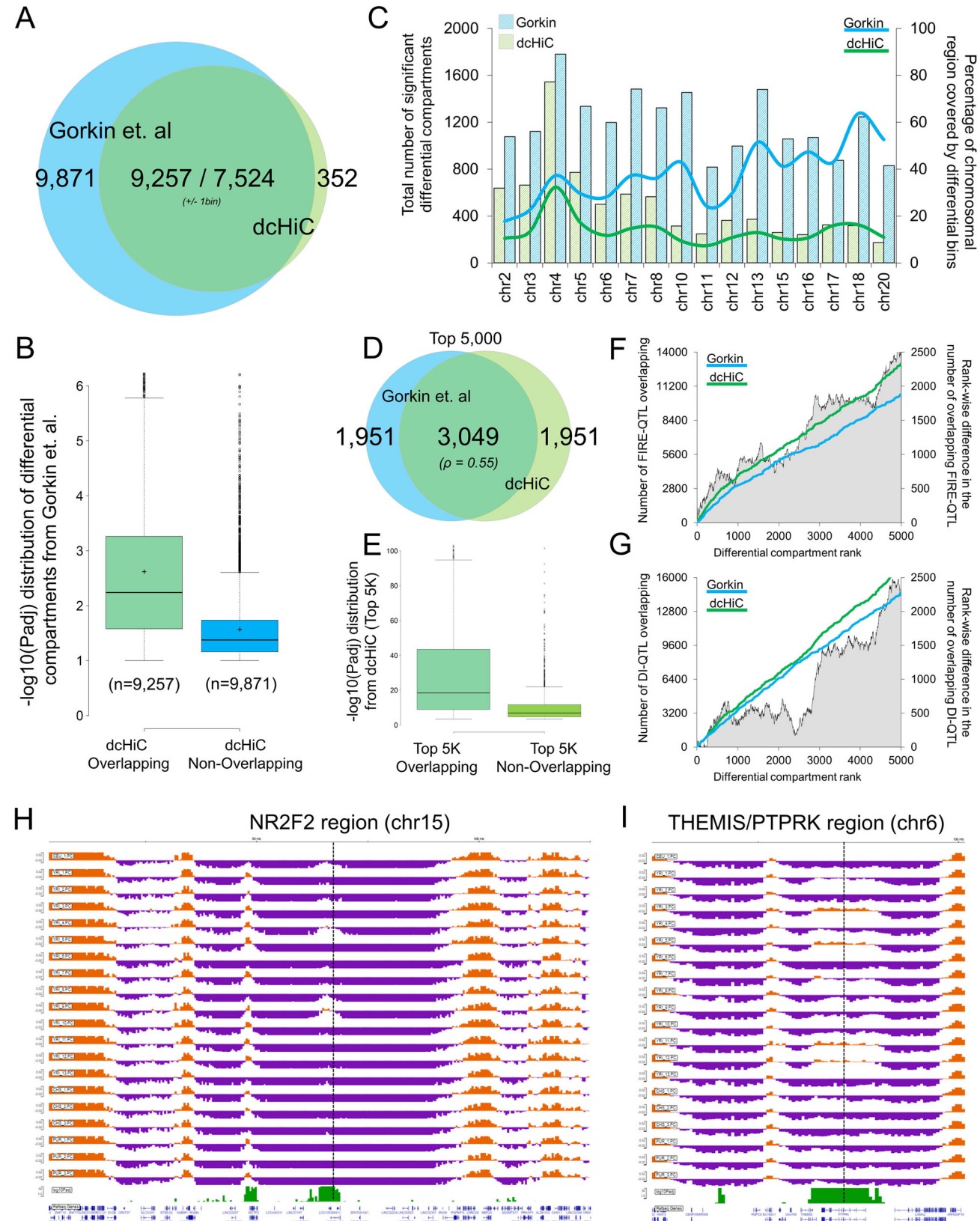

that dcHiC calls were more enriched in FIRE-QTLs (Fig. 7F) as well as DI-QTLs (Fig. 7G). Preferential enrichment of such signals suggests a better concordance of dcHiC identified compartmental differences and chromatin organization variability at other levels across individuals. We then asked whether the identified differential compartment regions were enriched in regions with variability in histone marks

(H3K27ac, H3K4me3, H3K4me1 and H3K27me3) across different individuals. The variable histone modification regions/peaks identified for human LCLs by Kasowski et al.[65] were mapped on differential compartments identified from dcHiC and by Gorkin et al.[66]. Using the non-differential compartment regions as background for each method, we observed nearly no enrichment for regions called differential only by

**Fig. 7 | Twenty-way multivariate differential compartment analysis of human lymphoblastoid cell lines (LCLs). A** A Venn diagram of the overlap of differential compartments called by dcHiC (blue) and the variable compartment regions from Gorkin et al. (green). **B** The distribution of −log10(p-adj) values of dcHiC-overlapping and non-overlapping variable regions calculated by the previous study, reported by Gorkin et al. **C** The total number of chromosome-wise differential compartments and the fraction of each chromosome (except those filtered by Gorkin et al.) covered by such calls for dcHiC and the previous study. **D**, **E** Venn diagrams of overlapping compartments of the top 5000 differential regions from

both approaches and the −log10(p-adj) value distribution of the overlapping and non-overlapping sets from dcHiC. **F**, **G** The cumulative number of FIRE-QTLs and DI-QTLs overlapping the top 5000 differential compartment calls by dcHiC and Gorkin et al. **H**−**I** Two example regions with differential compartments overlapping genic regions of *NR2F2* and *THEMIS/PTPRK*. Both genes and especially the *NR2F2* region were shown to be variable regions across the population through FISH experiments (Gorkin et al.). In **B** and **E**, data are presented as mean values ±SEM. Source data are provided as a Source Data file.

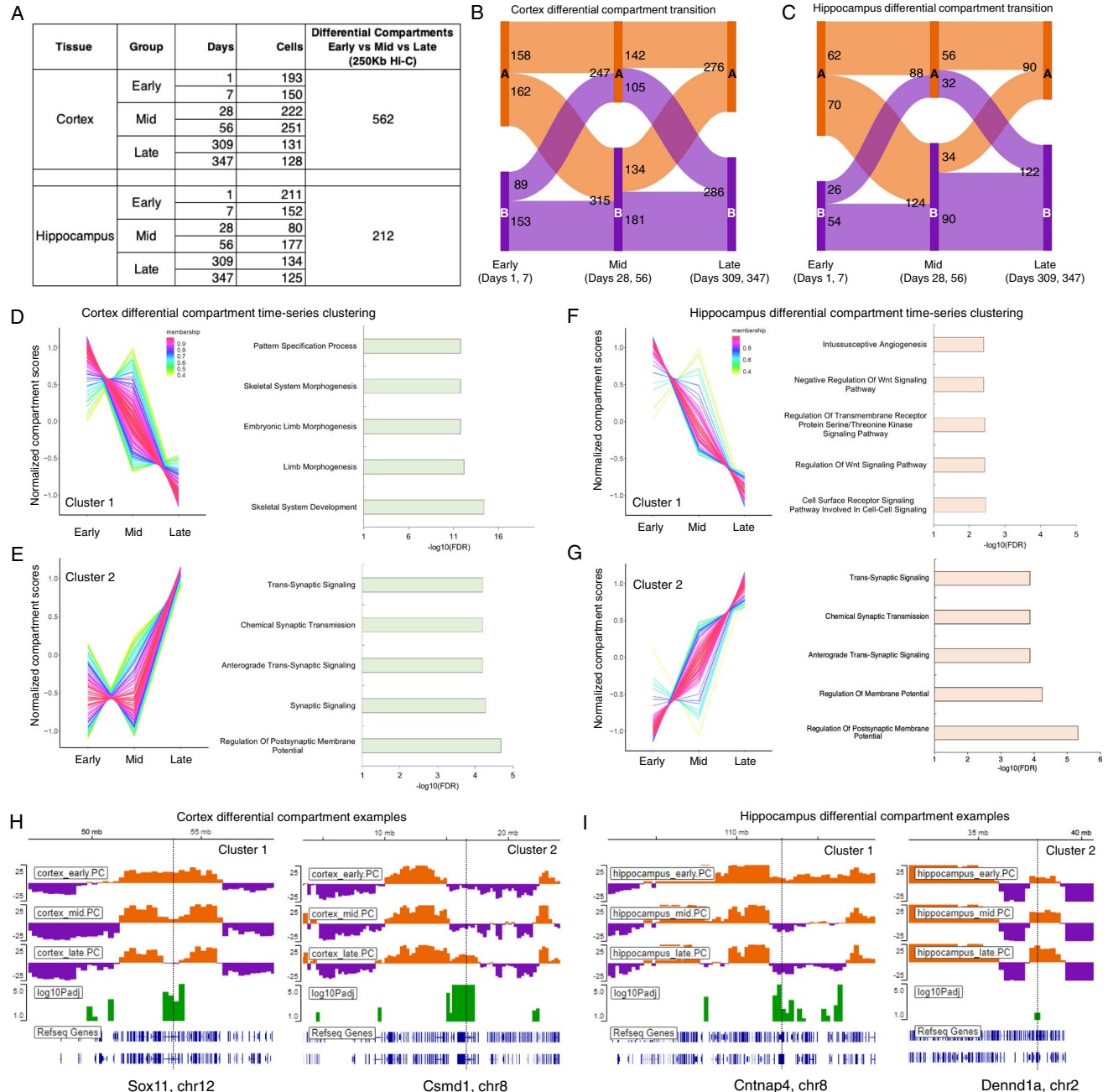

**Fig. 8 | Differential compartment analysis of pseudo-bulk single-cell Hi-C data from post-natal mouse brain development. A** Single-cell Hi-C summary and our categorization of 6 time points from each mouse brain region into three groups, namely – Early (includes Day 1, 7), Mid (Day 28, 56), and Late (Day 309, 347). **B**, **C** The transition of differential compartments in Early, Mid and Late groups within cortex and hippocampal region. **D**, **E** Time-series clustering of differential

compartments in cortex and the functional enrichment of genes overlapping with these compartments. **F**, **G** Time-series clustering of differential compartments in hippocampus and the functional enrichment of genes overlapping with the compartments. **H**, **I** Example genes overlapping differential compartments from cortex and hippocampus. Source data are provided as a Source Data file.

Gorkin et al. (Supplementary Fig. S10) while calls from dcHiC showed 26–45% of enrichment. The proportion of differential calls that overlapped with at least one variable region for each histone mark was also substantially higher for regions from dcHiC in comparison to Gorkin et al. specific regions (Supplementary Fig. S11A–D).

Example genes from differential compartments among human-derived LCLs: Fig. 7H, I show two examples of a variable region (overlapping with *NR2F2* and *THEMIS/PTPRK* genes) identified by dcHiC. The *NR2F2* region was investigated using FISH by Gorkin et al., which confirmed individual-specific changes in 3D chromatin conformation. Two of the individuals from the cohort (YRI-4 and YRI-8) showed enriched interaction between the *NR2F2* FISH and another placed upstream compared to YRI-3 and YRI-5. The variability of 3D genome organization among individuals is also apparent from compartment scores for this region. The *NR2F2* locus across the cohort was found to be a part of a strong B-compartment for all Yoruban individuals except for YRI-4, YRI-8, and YRI-9 (Fig. 7H). Figure 7I shows another example of such a variable region with coordinated changes in epigenetic marks across individuals with support from differential compartments documented in a previous paper. Gorkin et al. identified variations in different epigenetic marks, such as H3K4me1 and H3K27ac, binding of CTCF and, most importantly, gene expression patterns within this region across different individuals (YRI-2 and 13 vs 11 and 12). The PC score track in Fig. 7I also supports the previous findings, as some of the individuals from the YRI population, especially YRI-3, YRI-5, YRI-11, and YRI-12, showed a clear flip from the B to A compartment, and both our approach and Gorkin et al. labeled this region as a differential compartment. Taken together, dcHiC identified fewer differential compartment bins with enrichment towards capturing regions with higher variability in different levels of chromatin organization and those with additional evidence for differences among individuals.

## Differential compartment analysis of pseudo-bulk single-cell Hi-C data from post-natal mouse brain development

The post-natal dynamics of mammalian brain development is still a fundamental question in developmental biology[27]. Although, gene expression dynamics has been studied in developing adult and embryonic brains[67–70], the dynamics of 3D genome organization in conjunction with transcriptional changes remain largely uncharacterized. Tan et al.[27] attempted to address the issue by integrating single-cell gene expression and single-cell Hi-C data from two mouse brain regions (Cortex and Hippocampus). They employed diploid chromatin conformation capture (Dip-C) method and generated over 3k single-cell Hi-C maps from cortex and hippocampus encompassing 6 different time points that comprehensively describe the dynamic 3D genome organization. This high-quality, time-course and single-cell resolution Hi-C data provided us with an opportunity to showcase the expansion of dcHiC's utility in performing differential analysis on the pseudo-bulk single-cell Hi-C data. Tan et al. 2021 comprehensively described neuronal sub-types from the single-cell 3D genome data and most of their analysis focused on comparisons among the sub-types. Here we performed differential compartmental analysis among the time points to demonstrate the utility of dcHiC in identifying dynamic changes in compartmentalization for both brain regions. To perform the differential analysis, we first categorized the 6 time points from each brain region into three groups, namely – Early (includes Day 1, 7), Mid (Day 28, 56), and Late (Day 309, 347) as shown in the Fig. 8A. The two time points in each group were treated as replicates and pseudo-bulk Hi-C maps were analyzed at 250 kb resolution. We then used dcHiC with default parameters to perform differential compartment analysis and reported compartmental changes below an FDR threshold of 10% (Fig. 8B, C). Comparing the three groups (Early, Mid and Late), dcHiC found a total of ~140 Mb of the genome (562 Hi-C bins at 250Kb

resolution) in cortex and ~53 Mb of the genome (212 Hi-C bins) in hippocampus to be differential in their compartmentalization (Fig. 8B, C). The higher number of differential compartments in cortex may be reflective of its sudden change in compositional structure in the "Mid" group (i.e., higher fraction of oligodendrocytes) previously identified by Tan et al. 2021. Intersecting the differential compartments between the tissues revealed 89 Hi-C bins that are common as well as 473 Hi-C bins uniquely differential in cortex and 31 Hi-C bins uniquely differential in hippocampal tissue. The 562/473 bins in cortex overlapped with 2,973/2,566 genes while 212/31 bins in hippocampus encompassed 873/466 genes. A subset of the top cell-type specific marker and variable genes ranked by PC1 loading identified in the previous study (Tan et al., 2021) such as *Tshz2, Vip, Sst, Sox11, Tubb2b, Nrep*, and *Syt1* were also found to be part of dcHiC-identified dynamic changes (Supplementary Fig. S12A–F). We also observed other important genes in differential compartments that are either specific to one region or are common (Supplementary Fig. S12G–L). For example, we identified *Grin2a* and *Nrg3* (Supplementary Fig. S12) as part of differential compartments in both brain regions. *Grin2a* provides instructions for making a protein called GluN2A, which is a component of NMDA receptors. This protein is found in nerve cells of the brain region involved in speech and language processing[71]. Neuregulin 3 (encoded by *Nrg3*) is structurally related to neuregulin 1 (NRG1)[71], which plays a critical role in controlling the growth and differentiation of glial, epithelial and muscle cells[72]. The expression of *Nrg3* is known to be highly restricted within developing and adult nervous system. Tissue-specific differential compartments encompassing important genes included the region containing *Chrm5*, which encodes a muscarinic cholinergic receptor that binds acetylcholine and was found within a differential compartment in hippocampus but not in cortex (Supplementary Fig. S12). On the other hand, *Clstn2*, which is predicted to bind calcium ion and help positive regulation of synapse assembly and synaptic transmission was observed in a differential compartment that is specific to cortex (Supplementary Fig. S12).

The time-course analysis of three time points (early, mid, and late) also helped us identify significantly differential compartments falling into similar dynamic patterns (e.g., descending or ascending) for each brain region. The Fig. 8D, E shows the descending (cluster 1) and ascending (cluster 2) differential compartments across early, mid and late group in cortex. The genes overlapping with cluster 1 (descending pattern) of Fig. 8D showed enrichment in development and morphogenesis related GO terms, while differential compartments of cluster 2 (ascending pattern) in Fig. 8E were enriched in terms like membrane potential and synaptic signaling related biological functions. We also observed two other cluster patterns for cortex that are worth mentioning (Supplementary Fig. S13A, B). The first one corresponded to a peak in 'Mid' stage (cluster 3, Supplementary Fig. S13A) and the second one corresponded to a dip (cluster 4, Supplementary Fig. S13B). Cluster 3 genes showed an enrichment of non-specific functional terminologies while genes belonging to cluster 4 showed specific enrichment of terms related to nervous system (Supplementary Fig. S13C, D). Tan et.al. 2021 previously described a sudden change in compositional structure types where they observed a higher fraction of glial cells in the 'Mid' stage (Day 28, 56) within cortex. Interestingly, we found *Cd33*, a gene that is known to be expressed in microglia[73], as part of cluster 3 (peak in 'Mid'; Supplementary Fig. S14A). The region containing *Trex1*, a gene with enriched expression in glial cells in human brain[74] also belonged to cluster 3 (Supplementary Fig. S14A). Two example genes overlapping cluster 4 regions were *Gabra5* and *Anks1b*, both of which were specifically enriched for high expression in excitatory and inhibitory neurons[74] (Supplementary Fig. S14B–D). Figure 8F, G shows the significantly differential compartments with descending or ascending patterns in hippocampus. Unlike cluster 1 of cortex, the genes within differential compartments following a descending pattern in hippocampus are marginally enriched in Wnt

signaling, cell surface and transmembrane receptor signaling pathways. The genes overlapping with cluster 2 (Fig. 8G) in hippocampus, like those in cortex, are also enriched for biological functions such as membrane potentials and synaptic signaling terms.

Example genes from differential compartments among different stages of post-natal mouse brain development: Fig. 8H, I shows a pair of differential compartments from each region that follows the descending and ascending pattern of score transition among three time points and are overlapping with interesting genes. We observed *Sox11* (Fig. 8H, left panel), a gene identified by Tan et al. 2021[27] as one of the top variable genes, as part of the cluster 1 (Fig. 8D) in cortex. In the early stages of cortex, *Sox11* resides within an active compartment but with more differentiation the region overlapping with the gene undergoes a gradual A- > B transition. The right panel of Fig. 8H shows another gene *Csmd1* from cluster 2 in cortex. CUB and SUSHI multiple domains 1 (*Csmd1*) is known to be expressed in developing neurons[75] and plays critical role in learning and memory formation[76]. We found this gene as part of a specific differential compartment in cortex that gradually changes from B in early to A in late stages of development. The left panel in Fig. 8I shows the differential compartment encompassing the *Cntnap4* gene in the hippocampus following cluster 1's pattern shown in Fig. 8F. *Cntnap4* is part of the common set of regions that are differential in both brain regions. The right panel in Fig. 8I shows a hippocampus-specific differential compartment that overlaps *Dennd1a*, a protein coding gene known to be involved in vesicle-mediated transport pathways[77] and Rab regulation of trafficking[78]. Although highly expressed in neuronal as well as glial cells of the brain, the specific role of this gene in the hippocampus remains to be investigated.

Overall, these results showed that dcHiC addresses a need in the differential analysis of single-cell Hi-C data by first utilizing pseudo-bulk profiles from a low number of cells (80–251) to characterize compartmentalization of each condition, and then systematically comparing multiple conditions such as timepoints or clusters with the same approach we use for the bulk cell Hi-C data. Our analysis here offers an example scenario where comparing multiple conditions with replicates (three developmental stages) was essential to identify important known and new genes and to characterize dynamic patterns shared across different regions.

## Discussion

This paper presents an application, dcHiC, to compare compartmentalization across Hi-C datasets. dcHiC facilitates comparative analysis across multiple contact maps and helps identify biologically relevant compartmentalization differences with statistical confidence scores. Along with conventional pairwise differential analysis, dcHiC allows a single multivariate differential comparison of Hi-C datasets, utilizing replicates when available, and provides an efficient approach to analyze multiple Hi-C maps without the need for generating many different combinations. In terms of its approach, dcHiC first identifies principal components efficiently using a parallelized partial singular value decomposition (SVD). It then uses quantile normalization on the resulting compartment scores followed by computation of a multivariate distance measure to systematically identify significant compartmentalization changes among multiple contact maps. Through an extensive set of technical comparisons and analyses, we showed that dcHiC enables high-resolution compartment analysis in a more time-efficient manner compared to the existing approaches. Using replicates and pseudo-replicates, we show that dcHiC has a low type-1 error rate and the identified differential compartments are fairly robust to varied levels of sequencing depths and resolutions.

We applied dcHiC to various biological scenarios, ranging from neuronal and hematopoietic stem cell differentiation in mice to Hi-C data from different human populations along with pseudo-bulk single-cell Hi-C maps of cells from cortex and hippocampus during post-natal

mouse brain development. Our results confirmed that dcHiC detects known compartmental changes among cell types, including those previously shown to play a role in neuronal and hematopoietic differentiation. When comparing dcHiC to existing approaches, we showed that it identifies regions with higher differences in replication timing, Lamin B1 signals, differentially expressed genes, and histone marks suggesting a better prioritization of relevant biological changes. Even though differences in compartmentalization between ESCs and NPCs are generally aligned with changes in the lamin B1 association, a recent work highlighted the importance of nucleolus association in revealing layers of compartmentalization with distinct repressive chromatin states[79]. Our initial analysis showed that over 10% of all significant compartment differences we found between ESC and NPC belong to nucleolus-associated domains (NADs) that were deemed exclusive to either ESC or NPC[79], providing an explanation for a subset of differences in compartmentalization during differentiation. In the same pairwise analysis, we also found that differential compartments are enriched for sub-compartmental flips (changes in subcompartment labels) especially with an enrichment of strong heterochromatin ESC sub-compartments transitioning to euchromatin sub-compartments in NPC as compared to the background. Such changes were also associated with strong alterations in the transcriptional activity of overlapping genes.

We next expanded our analysis to a three-way ($n = 3$) comparison of cell types during in vitro mouse neuronal differentiation and showed that dcHiC continues to systematically identify critical marker genes and can recover cell-specific functions from differential compartment analysis alone. Across dcHiC's differential compartments, we observed significant and relevant enrichment of biological processes such as neuron differentiation in NPC and CN cells. More broadly, dcHiC's differential compartments also compellingly aligned with changes in Lamin B1, gene expression, and histone modification data. Taken together, these results demonstrate dcHiC's ability to find regions with the most biologically relevant changes in compartmentalization across the genome.

The hierarchical mouse hematopoietic stem cell differentiation model, consisting of ten different cell types with Hi-C data ($n = 10$), provided a unique opportunity to demonstrate a number of different utilities of dcHiC. A ten-way multivariate differential comparison of the hematopoietic system revealed previously known lineage-specific critical genes overlapping differential compartments. Notably, we identified vital transcription factors, such as *Sox6, Runx2, Meis1, Foxo1*, and many other critical genes, such as *Abca13*, by solely analyzing the differential calls. Our functional enrichment analysis of gene sets overlapping with the lineage-specific differential compartments reported from the apex to the bottom of the hematopoietic model tree reconfirmed that genome compartments play a contributory role in determining the accessibility of genes in specific cell types. We also performed time-series analysis for two different lineages (LT-HSC to GR and LT-HSC to MK) leading to different compartment change patterns some of which were gradual increase or decrease and several others were driven uniquely by the megakaryocyte–erythrocyte progenitor (MEP) state.

Measuring the extent of compartment variability across twenty-cell human types ($n = 20$) also highlighted our method's utility and strength. Most dcHiC calls overlapped with a subset of variable compartments reported by the previous study, but dcHiC calls were enriched for higher variability in a number of different biological signals. Similarly, the regions encompassing Frequently Interacting Region (FIRE)-QTLs and Directionality Index (DI)-QTLs defined by the previous study were more enriched in the top differential compartment calls of dcHiC than in the top calls defined in the previous study. The analysis also demonstrated an important feature of dcHiC: the ability to directly utilize previously computed compartment scores to run differential compartmentalization analysis.

Genomic compartment identification from single-cell Hi-C maps is still a major issue in this field due to sparsity of single-cell contact maps even at coarse resolution. A recent paper (Zhang et al. 2021) demonstrated the challenges introduced by different technical and biological factors in reliably calling and comparing A/B compartments across single cells. Here, by studying a recent single-cell Hi-C (Dip-C) data characterizing post-natal dynamics of mouse brain development in two brain regions, we showed that dcHiC is able to call compartments and perform differential analysis from pseudo-bulk profiles of as low as 80 cells per condition. We observed a higher number of differential compartments in the cortex region compared to the hippocampus. The differential compartments were overlapping with cell-type specific marker genes and previously known variable genes identified through single-cell transcriptomic analysis. The time series clustering of the differential compartments across three stages ('Early', 'Mid', and 'Late') of development helped us identify other important genes in differential compartments that were not captured in the original paper. Although we highlighted the use of dcHiC for time course analysis of single-cell data, the same can be applied to any predefined set of clusters of cells either with respect to their functional annotations (cell type or subset) or sample conditions (e.g., WT vs KO).

The framework we developed here provides a systematic way to identify differential compartments and visualize these differences in different scenarios, including multiway, hierarchical and time-series settings. Although we focused on human and mouse Hi-C data in this work, our method is readily applicable to Hi-C data or its variants (e.g., Micro-C[80]) derived from any organism with compartmental genome organization. dcHiC is also readily applicable to comparative analysis of other coarse-grain (10 kb to 1 Mb resolution) genome-wide signals such as replication timing and lamin association. With hundreds of publicly available Hi-C datasets in the 4D Nucleome Data Portal and others published every day, dcHiC will play an essential role in the comparative analysis of high-level genome organization. As single-cell Hi-C data start providing better resolution for compartment analysis, dcHiC and its future extensions will be critical to enable compartment comparison across thousands of cells and tens of conditions, clusters or cell subsets.

## Methods

### Data processing, result generation, and visualization
In the paragraphs below, we describe our Hi-C, RNA-Seq, ChIP-Seq, time series, and browser visualization methods.

All the Hi-C data, except for Gorkin et al., were mapped to the mm10 reference genome and processed using the HiCpro (v2.7.9) pipeline[81]. The raw Hi-C interaction maps retrieved after HiC-Pro processing are used for downstream compartment score calculation by dcHiC. In the section analyzing data from the Gorkin et al. study, we used the provided compartment scores (40 kb resolution) across all samples mapped to the hg19 reference genome[66]. Statistically significant interactions were called using FitHiC2[46] with default parameters and an FDR threshold of 0.05 for each replicate and/or each sample.

The RNA-seq data from Bonev et al.[39] study concerning mouse neural development were processed using our in-house and open-source RNA-seq processing pipeline (https://github.com/ay-lab/LJI_RNA_SEQ_PIPELINE_V2.git), which utilizes STAR[82]. The differential gene expression analysis between mouse ESC and NPC cell lines (two replicates each) was performed using the DESeq2 method[41] with all the default parameter settings.

For ChIP-seq peak calling (H3K27ac, H3K4me3, and H3K4me1 histone marks), we first mapped the respective fastq files to the mm10 genome using bowtie2[83] and generated the corresponding bam files (MAPQ > 20). The aligned files were then used as input to the MACS2 program[84] to call peaks (p-value <1e-5) against their respective input controls. The continuous ChIP-seq peaks were then merged, and the

unique set was mapped to the 100 kb differential compartments to calculate the average number of peaks. The enrichment of signal difference was calculated by first quantifying the absolute difference in signal (number of ChIP-seq peaks and gene expression TPM values) within ESC to NPC differential and non-differential compartments. The enrichment of absolute signal difference between the differential and non-differential compartments between ESC and NPC was then compared by unpaired T-test.

Time-series clustering was generated using the TCseq package[50]. For gene-term enrichment analysis, the differential compartments are scanned against the gene coordinates of the respective genome defined by the user using the 'bedtools map' function[85]. The unique overlapping set of genes was then extracted and used for GO biological function enrichment analysis using the ToppGene suite API function or directly from their webserver[51].

dcHiC generates a JavaScript-based stand-alone dynamic IGV-HTML page to visualize the compartments and differential compartment calls, with an option to add additional tracks.

### Computation and quantile normalization of compartment scores for comparison
To perform principal component analysis (PCA) on Hi-C maps, dcHiC utilizes the singular value decomposition (SVD) implementation of the *bigstatsr* R package[34]. The input to SVD is $K$ different distance-normalized chromosome-wise correlation matrices $(X_1, X_2, X_3 .. X_K)$ for each Hi-C data. For each such matrix, dcHiC finds the decomposition:

$$X_K = U_K \cdot \Gamma_K \cdot V_K^T \ with \ U_K^T \cdot U_K = V_K^T \cdot V_K = I \tag{1}$$

The matrices $U_K$ and $V_K$ store the left and right singular vectors of the matrix $X_K$. The singular values of $X_K$ are stored in the diagonal matrix $\Gamma_K$. The principal components for each matrix are then obtained as:

$$PC_K = X_K \cdot V_K \tag{2}$$

The eigen-decomposition of the $K^{th}$ correlation matrix provides the eigenvectors, and the sign of the first principal component ($PC1_K$) typically represents the genomic compartments A and B for the $K^{th}$ chromosome. If $PC1_K$ corresponds to chromosome arms or other broad patterns in the Hi-C matrix, the second principal component ($PC2_K$) may represent A and B compartments. The A/B compartment labels are assigned to the positive/negative stretches of the selected $PC_K$ depending on the implementation of eigen-decomposition. It may be necessary to reorient these assignments and select the correct $PC_K$ using GC content or gene density. Thus, before the quantile normalization step, dcHiC performs an intermediate correlation analysis of the first two principal component scores (user-defined) of each chromosome per sample against the GC content and gene density of that chromosome. The principal component that obtained the highest sum of GC content and gene density correlation was considered the compartment score, and the A/B compartments of the selected principal components were assigned based on the GC content correlation (A compartment and positive values representing higher GC content). These generate a set of compartment score vectors representing each sample ($M$ samples) for a given chromosome ($C_1, C_2, C_3 \ldots C_M$). Once the properly labeled compartment scores are obtained, dcHiC performs quantile normalization (QN) using the limma package[86] on the set ($C_1, C_2, C_3 \ldots C_M$) per chromosome to even out the scaling across the group for downstream analysis.

$$(q_1, q_2, q_3 \ldots q_M) = QN(C_1, C_2, C_3 \ldots C_M) \tag{3}$$

In the case of samples with replicates, dcHiC performs the above steps by including each replicate from each sample (i.e., quantile

normalizes all replicates together). dcHiC then calculates the average quantile normalized values of each genomic bin across all replicates of a given sample to represent sample-wise compartment scores.

## Differential compartment identification

Mahalanobis distance (MD) is a multivariate statistical measure of the extent to which the multivariate data points are marked as outliers based on a Chi-square distribution[87]. The Mahalanobis distance of a point $i$ from a multidimensional distribution defined by set $s(sample)$ and its center $\mu$ is defined as:

$$MD^i_{sample} = (s_i - \mu_i)^T \cdot \sum{}^{-1} \cdot (s_i - \mu_i) \quad (4)$$

where $s_i = (q^i_1, q^i_2, q^i_3 \ldots q^i_M)$ is the set of quantile normalized compartment score distributions and $\mu_i = (\mu^i_1, \mu^i_2, \mu^i_3 \ldots \mu^i_M)$ is the set of weighted centers for each point $i$ from set $s$. The inverse of the covariance matrix of set $s$ is represented as $^{-1}$. The weighted centers $\mu_i$ are calculated as:

$$\mu_i = s_i * w_i \quad (5)$$

where $0 \le w_i \le 1$ is the cumulative normal distribution probability associated with the maximum $z$-score among the $z$-scores of all samples for $i$:

$$w_i = \left\{ \Pr\left( Z^i_M \right) \right\} \quad (6)$$

where $Z^i_N$, the $z$-score for point $i$ for sample $N$, is computed as:

$$Z^i_N = \frac{(d^i_N - \overline{d_N})}{\sigma(d_N)} \quad (7)$$

Here, $\overline{d_N}$ and $\sigma(d_N)$ represent the average distance and standard deviation within sample $N$ among all $d^i_N$ values that are computed as:

$$d^i_N = \frac{\sqrt{\sum_{t=1}^{M}(q^i_t - q^i_N)^2}}{(M-1)} \quad (8)$$

Essentially, the approach provides more weight to the points that are distant from others among the samples (further from the diagonal) than to points that are closer together in the multidimensional space (close to the diagonal). Equation (4) is the standard MD formulation, which we modify using the weighted centers as computed through Eqs. (6)–(8).

To increase the sensitivity of our difference detection, we implemented an outlier removal step that eliminates genomic bins (or points) with high MD (as computed above) at the initial pass (1st pass). For calculation of the covariance matrix, we utilize the *covrob* function of the R package *robust*, which implements the Minimum Covariance Determinant (MCD) procedure that has been shown to improve multivariate outlier detection[88]. We use a predefined upper-tail critical value of the chi-square distribution with $df$ degrees of freedom as our threshold for outlier removal (the default value we used is $MD\ threshold \sim \chi^2_{0.90,df}$). We then recompute the covariance matrix $^{-1}$ after removal of these outliers and calculate the MD (through Eq. (4)) one more time for each point (2nd pass). The significance of the corresponding $MD^i_{sample}$ (2nd pass) is calculated from the critical chi-square distribution table as $\chi^2(MD^i_{sample}, df)$ using the *pchisq* function of the R programming language followed by multiple testing correction to retrieve adjusted $p$-values.

In the case of samples ($s$) with replicates, ($r$) dcHiC calculates an additional covariate $MD_{replicate}$ and applies Independent Hypothesis Weighting (IHW) to adjust the $p$-values.

The covariate is calculated as follows:

$$MD^i_{repl} = \left\{ (s^r_i - sr\mu_i)^T \cdot (diag(^{-1})) \cdot (s^r_i - sr\mu_i) | s \in (1, \ldots M), r \in (1, \ldots R) \right\} \quad (9)$$

where $R$ is the total number of all replicates combined across all samples, $s^r_i = (r^i_1, r^i_2, r^i_3, \ldots r^i_R)$ is the set of quantile normalized compartment score distributions of all replicates from samples $s \in (1, 2 \ldots M)$ and $sr\mu_i = (s1\mu_i, s2\mu_i, s3\mu_i, \ldots sR\mu_i)$ is the set of weighted centers for each point $i$ from $R$ replicates. $diag$ is an operation that masks all nondiagonal entries (sets to zero) of the covariance matrix.

The weighted centers $sr\mu_i$ are calculated as:

$$sr\mu_i = s^r_i * (1 - srw_i) \quad (10)$$

where $0 \le srw_i \le 1$ is calculated as:

$$srw_i = \{Pr(srZ_R)\} \quad (11)$$

$srZ_i$ for replicate $r$ of sample $s$ is computed as:

$$srd_i = \frac{\sqrt{\sum_{t=1}^{R}(r^i_t - r^i_R)^2}}{(R-1)} \quad (12)$$

$$srZ_i = \frac{(srd_i - \underline{srd})}{\sigma(srd)} \quad (13)$$

Here, the variables are defined as similar to Eqs. (7) and (8), and $R$ is used to represent the number of replicates of the same sample (i.e., distances across replicates of different samples are not taken into account).

This approach provides more weight to the features that are closer to each other within replicates of a sample (close to the diagonal) and as opposed to the calculation across different samples (Eq. (5)), where higher weights were given to the points with samples distant from each other (far from the diagonal). The significance of the corresponding $MD$ for each point is calculated using the chi-square distribution as mentioned above. dcHiC applies the IHW approach to adjust the $p$-values using FDR correction obtained from $MD_{sample}$ using the $MD_{replicate}$ replicate variation measure as a covariate.

## Differential interaction identification

Using the same Mahalanobis distance (MD) measure, dcHiC enables the user to find differential interactions across samples that are either linking two differential compartments together or a differential compartment with other parts of the same chromosome. The goal of this feature is to provide more information on the chromatin organization changes related to or correlated with compartmental differences. For this analysis, we used FitHiC2 to call significant interactions (FDR 5%) for each sample or replicate (when available), but users are free to provide their own set of interaction or loop calls from any other tool. Using these calls, dcHiC first finds the interaction subset that overlaps with differential compartments (on either end or both) using the bedtools 'pairtobed' function. dcHiC utilizes the $\log2(\frac{Observed}{Expected})$ values of a chromatin interaction $i$ to perform differential interaction calling as follows:

$$MD^i_{interaction} = (oe^s_i - \mu^s)^T \cdot {}^{-1} \cdot (oe^s_i - \mu^s) | s \in (1, 2, 3 \ldots M) \quad (14)$$

where $oe^s_i$ represents $\log2(\frac{Observed}{Expected})$ values for chromatin interactions of locus pair $i$ for sample $s$ and $\mu^s$ represents the vector of centers of distance normalized interactions from sample $s$. Here, $^{-1}$ represents the inverse of the covariance matrix of interactions among the samples. The approach provides more weight to the interactions that are

distant from the expected interaction strength among the samples than to the interactions that are closer to the expected range in the multidimensional space. The significance of the corresponding $MD^i_{interaction}$ is calculated from the critical $\chi^2$ distribution table as $\chi^2(MD^i_{interaction}, df)$ using the *pchisq* function embedded within the R programming environment followed by FDR correction to retrieve adjusted p-values.

## Statistics and reproducibility

No statistical method was used to predetermine sample size, and no data was excluded from the main body of the text. The Investigators were not blinded to allocation during experiments and outcome assessment. We utilized all biological replicates available to us where possible, except in one location specified in the manuscript: when testing the runtime and memory usage of dcHiC, where having evenly sized input samples helped make data collection more uniform.

## Reporting summary

Further information on research design is available in the Nature Portfolio Reporting Summary linked to this article.

## Data availability

The mouse ESC, NPC and CN Hi-C data used in this study are available in the GEO database under the following accession code GSE96107. The mouse hematopoiesis Hi-C data used in this study are available in the GEO database under the following accession code GSE152918. The single-cell Hi-C data used in this study are available in the GEO database under the following accession code GSE146397. The human LCL Hi-C data used in this study are available in the GEO database under the following accession codes GSE128678 and GSE50893. These are also listed in Supplementary Table S10. All reported compartments for all cell lines, multivariate differential scores, RNA-seq, and ChIP-seq data used in this manuscript can be viewed interactively at ay-lab.github.io/dcHiC. These standalone HTML files employ dcHiC's visualization utility through the IGV browser. Source data are provided with this paper.

## Code availability

A Python/R implementation of dcHiC is freely available at https://github.com/ay-lab/dcHiC or at this DOI: 10.5281/zenodo.7256046[89]. This application is compatible with Hi-C data in HiC-Pro,.*hic*, and .cool formats.

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

## Acknowledgements

We would like to thank Sourya Bhattacharyya, Daniela Salgado-Figueroa, and all members of the Ay lab for their input and helpful comments on the tool development. We would also like to thank Dr. Katia Georgopoulos and Dr. David Gilbert for helpful discussions about interpretation of dcHiC results. This work was funded by NIH Grant R35-GM128938 awarded to F.A.

## Author contributions

A.C., J.W., and F.A. conceived the project and designed the method. A.C. and J.W. implemented the software and carried out experiments under supervision of F.A. A.C. and F.A. interpreted the results, and drafted the manuscript with input from J.W. All authors read and approved the final version of the manuscript.

## Competing interests

The authors declare no competing interests.
