## [Peer Review File · Nature Communications]

Reviewers' Comments:

Reviewer #1:

Remarks to the Author:

The authors present dHiC for differential compartment analysis. There is a need to innovate compartment analysis in Hi-C as the bioinformatic methodology has not changed much over the years. The authors mostly benchmark against Homer and use dHiC to characterize differential compartments in published Hi-C datasets of differentiation and between individuals. However, there are a few major concerns to address in characterizing the tool that dampen my enthusiasm. As presented, it is unclear if dHiC is an improvement of current methods. Additionally, it is difficult to see the significance of the tool, as there are little to no large departures from that of published findings.

First, the authors compare only to Homer. However, both methods rely on eigenvalue decomposition. They should compare to alternate methods, like CScoreTool: <https://pubmed.ncbi.nlm.nih.gov/29244056/> .

In these comparisons, the authors should compare and report standard benchmarks, i.e. time and memory usage at various resolutions and sequencing depths.

The authors should evaluate the effects of sequencing depth on compartment calls and differential compartment calls by downsampling maps. They should then report the false-positive / false-negative rates compared to the high-read depth map. This should be done for a range of resolutions and sequencing depths.

The manuscript relies heavily on correlation with other features, such as histone marks, replications timing, etc to validate the compartment calls and differences. However, they validate that the differences between Homer and dHiC (and CScores) reflect differences in the Hi-C. They should show Hi-C examples of the loci where Homer and dHiC differ. They should also examine the compartment scores independently by computing the intensity scores (AA)/(AB) of the actual Hi-C signal for these regions.

Currently, it is difficult to tell if Homer or dHiC is more accurate. Homer detects more differences, which appear to also correspond to differences in correlated features (Fig 3F&H). This dramatically dampens my enthusiasm.

Similar to above, Gorkin et al identified far more differences than dHiC, and it is not clear whether or not these are false positives. The authors should show the Hi-C signal at Gorkin exclusive vs dHiC exclusive differences. Additionally, they should test whether Gorkin specific differences correlate with differences in the histone marks that they published.

Authors should show what the Gorkin compartment calls look like at the NR2F2 and THEMIS regions to see how dHiC compares.

Reviewer #2:

Remarks to the Author:

In this manuscript, Chakraborty and colleagues proposed dHiC to perform differential analysis of A/B compartments, from Hi-C data. Overall, there are many merits of the manuscript. The method is simple yet well motivated. Differential compartment testing is useful for Hi-C data analysis. The manuscript is well organized and easy to follow. The authors applied their methods to three datasets, leading biologically findings that are expected and supported from auxiliary transcriptomics or epigenomic data (although I didn't find anything particularly note-worthy or novel in terms of biological insights). The GitHub site is well constructed and the examples from analyses performed in the manuscript are valuable for users.

However, the biggest concern is that the method needs to be compared with the state-of-the-art alternatives. The authors started by showing that the method generates results consistent with the standard PCA analysis. How about applying PCA analyses to the datasets they analyzed and what would the results look like? Would the standard PCA approach lead to similar findings? What does dHiC offer that would have been missed by PCA?

Other major comments are listed below.

(1) As a statistical testing method, the authors need to demonstrate the validity of the methods: that is, the authors need to first show protection of type-I error. The authors can do this by applying dHiC to Hi-C datasets with replicates or randomly splitting high-depth Hi-C data into two sets. Under such scenarios, no differential compartments are expected, which would allow an evaluation of the method validity. Without establishing the validity, a statistical hypothesis testing is meaningless.

(2) The authors applied an outlier detection approach when performing differential compartment analysis, which appears somewhat ad hoc. A standard approach is to use parametric or non-parametric ANOVA analysis, which is computationally fast, as well as statistically straightforward and extensively used.

(3) The authors should be applauded for attempting finer-resolution in at least one dataset. Have the authors applied their methods at higher-resolution to the other two datasets. How would the depth of Hi-C data affect the finest resolution recommended?

(4) Have the authors considered extending the approach to single cell Hi-C data? A recent study (PMID: 33484631) has suggested that compartment score can reflect 3D chromatin structure in single cells. Can dHiC handle the sparse single Hi-C data?

Minor comments:

(1) Under the "Computation and quantile normalization of compartment scores for comparison" sub-section of the Methods section, the authors said input to SVD is "distance-normalized" matrices: how were the matrices distance-normalized?

(2) What is the computational costs of dHiC? How does it scale with respect to input data depth, number of cell types compared etc?

(3) How does the method perform when the data depth differs substantially? Conceptually, quantile normalization handles the issue but can the authors show some results with differential depths across the tested cell types?

Reviewer #3:

Remarks to the Author:

In this study, Chakraborty and colleagues introduce a new computational method to determine differential chromatin compartmentalization across 2 or more samples. The method is briefly introduced in the main manuscript, and described in more detailed in the Methods section, and then ample space is dedicated to demonstrate the power of this approach in distinct biological contexts comparing Hi-C datasets ranging from 2 to 20 in number. In addition, the authors implement and perform multiple downstream analyses to gain information from the differential compartment regions detected by dHiC. Overall, this is an interesting approach addressing an under-appreciated and under-studied problem (it's amazing how much has been done to detect and compare TADs and how little to do so for chromatin compartments).

Given the main contribution of this work is the development of a tool, I would have appreciated more insight and testing on the method itself, while the detailed description of the results from the various comparisons could sometime be reduced, especially when simply confirming previous findings. Below I highlight more detailed suggestions:

1) A major concern of using PC values as compartment scores is data resolution (total number of Hi-C contacts). The authors employ a quantile normalization to make these values comparable across different experiments (which is great), but they should still test to what extent differential compartment regions can be detected by simply changing the resolution of the same experiment.

To this extent, the authors should test the robustness of their method to data resolution in different ways such as:

- by using dHiC to compare multiple version of the same Hi-C matrix down-sampled to a different percentage of reads, this will allow to get an idea of the influence of data resolution on the extent of differential compartments expected by chance (and even provide guidelines to the users on when and in which context this comparison is reliable)

- and by repeating the comparisons in the manuscript after reads-downsampling, these comparisons would allow to assess the robustness of the results in term of both the number of significant hits and how consistent these hits are upon downsampling. (I.e., will they find the same set of hits when comparing the same datasets but at different resolution?)

2) In the first part of their algorithm, the authors present a new/fast approach to detect compartments (A and B) and compare their strategy to the more standard approach based on PCA as implemented in HOMER. Here I have a couple of questions:

- if I understand correctly the main difference between the two is that dHiC employs SVD, while HOMER standard PCA, is that the only difference? Can the author provide a few more details on the two approaches?

(Also, how does HOMER perform differential compartment analysis? Why does it return such a larger number of hits?)

- while the correlations shown in Figure 1A and 1D are strong, since compartments are called based on the sign of the PC, an additional (more proper) comparison would have been to show the fraction of bins that have different A/B assignments with the two approaches.

Indeed in both comparisons in Figure 1A and 1D it appears that there is a subset of bins in the top-left quadrant that would be called A by dHiC and B by HOMER. Of course I expect differences between the tools, 100% consistency would be unrealistic, but it is curious that while some A compartment regions in dHiC are called B by HOMER (top-left quadrant), the vice versa almost never occur (bottom-right quadrant) suggesting a systematic shift of scores. Can the authors quantify the fraction of bins where discordant calls occur? Do they have an intuition on why discordant calls are almost exclusively in one direction?

3) Along the same lines of the previous comment, the authors did not compare their compartment calls with those of other approaches, especially when these allow to call subcompartments (see PMID: 25497547, PMID: 31699985, PMID: 33972523). Besides the comparisons of A and B compartment calls made by these approaches, it would interesting to combine subcompartments inferred by them with dHiC results to have a more granular analysis of significant compartment differences: do they at least involve subcompartment flips? are regions in different subcompartment equally likely to change or certain subcompartments are more "flexible"? (E.g. in PMID: 33972523 the authors talk about subcompartments enriched for differentiation genes, the regions of frequently change compartments across cell lines)

4) To evaluate the results obtained by dHiC, the authors performed several enrichment analyses (epigenetic features, gene expression, gene ontology etc.) While these are welcome, it would be great to have a better feeling of how frequently a significant compartment change is supported by orthogonal evidence. How many compartment changes are indeed associated with differentially expressed genes (and vice versa)? How many are associated with epigenetic changes (the authors could check if ChIP-seq data for histone modifications is available)?

Also the authors always report compartment changes in terms of number of bins, but I suspect in many cases multiple bins are contiguous (a compartment change of only 1 bin is more likely to be due to noise). If they account for contiguity how many changes do they get and what is their size distribution?

Possibly using orthogonal evidence and a size threshold could help determine the true differences and further filtering false positives which could emerge for technical reasons such as data

resolution.

A few minor comments:

5) How much do the results of dHiC depend on the weighting scheme adopted (Eq. 6 in the Methods)? It would be important to understand the contribution of this parameter to the results. How would the result change without weighting? What if a different weighting strategy was adopted (e.g. 75% quantile instead of max Z)?

6) Why wasn't the time-series analysis done also for the HSC lineage differentiation study? That would be a nice addition to understand the number of concordant/progressive changes during lineage differentiation vs. changes that emerge sporadically.

7) I believe these are typos/oversights, but in the background section the descriptions of PCA and eigenvector decomposition are imprecise:

- at line 65 the sentence seem to indicate that eigenvectors and principal components are the same thing, but they aren't,

- at line 73 the authors write "magnitude and sign of eigenvalues derived from PCA have been the major determinants of compartment type", this is just wrong, eigenvalues are only used to rank eigenvectors. It is the sign and magnitude of the values of the first (or second) principal component that are used to determine compartment type.

All of this is correctly reported in the Methods so I believe these were simple oversights, but they should be corrected.

Reviewer #1 (Remarks to the Author):

Reviewer comments are in red, our responses are in black and copy-pasted text from the manuscript are colored in blue

The authors present *dcHiC* for differential compartment analysis. There is a need to innovate compartment analysis in Hi-C as the bioinformatic methodology has not changed much over the years. The authors mostly benchmark against Homer and use *dcHiC* to characterize differential compartments in published Hi-C datasets of differentiation and between individuals. However, there are a few major concerns to address in characterizing the tool that dampen my enthusiasm. As presented, it is unclear if *dcHiC* is an improvement of current methods. Additionally, it is difficult to see the significance of the tool, as there are little to no large departures from that of published findings.

We thank the reviewer for their comments. We would like to first clarify that *dcHiC*'s main contribution and innovation is in enabling multi-way comparisons of compartments and in identifying significant changes that do not involve compartment flips/switches. In the process, we have also implemented an SVD-based compartment caller that is more efficient than existing tools such as HOMER and CscoreTool. However, our compartment calls are largely in agreement with these existing PCA-based approaches as expected by the general formulations of the matrix decomposition methods. Other than computational efficiency (mainly in runtime and scalability), we do not claim any superiority over existing tools for the task of calling compartments. We do, however, show multiple lines of evidence supporting *dcHiC*'s advantages in differential analysis (significantly extended in this version) and in providing novel analysis modalities that were not present in any other tool to date.

Below we would like to highlight the most significant changes in this revision for your reference:

1. As suggested by all reviewers, we have now extensively studied the impact of sequencing depth and resolution on the compartment calls and differential compartment calls from *dcHiC* including a false positive rate analysis using replicates and recovery analysis using downsampling. These analyses demonstrated that our results are quite robust for most relevant settings and allowed us to provide guidelines about the sequencing depth and other requirements for proper utilization of *dcHiC*.
2. We now also applied *dcHiC* on single-cell Hi-C data demonstrating its utility in comparing different clusters and/or time points to study dynamic changes in compartmentalization from pseudo-bulk Hi-C profiles from as low as 80 single cells. Our analysis of the Tan et al. (Cell, 2021) single-cell Hi-C data from two regions of post-natal developing mouse brain (each with 6 time points) allowed us to reveal important genes related to synapse assembly and adult brain development that have dynamic and tissue-specific compartmentalization changes, which were not highlighted by the original publication.
3. As requested by multiple reviewers, we have now substantially expanded our comparative analysis to HOMER, Gorkin et al and added Cscore into these comparisons. We have profiled runtimes and memory utilization of each approach for compartment detection in multiple settings

and resolution, highlighting the clear advantage of *dcHiC* for high-resolution/finer-scale compartment analysis.

4. We also extensively compared the overlap of differential compartments identified by different approaches with changes/variation in orthogonal measurements from matched samples including gene expression, histone modifications and lamin B1 signal. These results confirmed and strengthened our previous findings that *dcHiC* differential compartments are consistently more enriched in variation of other biological signals.

5. To address confusions from multiple reviewers, we have added a detailed discussion of HOMER and *dcHiC* in terms of their specific implementation of compartment calling and the source of *dcHiC*'s performance improvement including the use of more efficient data structure.

6. We have expanded our documentation in Wiki page of our Github repository (<https://ay-lab.github.io/dcHiC/>) to include single-cell analysis of Tan et. al and provided two demos with accompanying test data to show how *dcHiC* can be used for bulk and single-cell Hi-C data analysis: <https://github.com/ay-lab/dcHiC/demo>

7. We have also added additional analyses showing: i) the overlap of *dcHiC* results with subcompartment calls and their differences, ii) consistency of *dcHiC* calls across different resolutions including high-resolution such as 10kb and 20kb maps when possible, iii) the dynamic changes through time series analysis of specific lineages in the mouse hematopoietic lineage.

1. First, the authors compare only to Homer. However, both methods rely on eigenvalue decomposition. They should compare to alternate methods, like CscoreTool: <https://pubmed.ncbi.nlm.nih.gov/29244056/>.

We have now done this and incorporated the results in Figure 2. In the revised Figure 2, we show correlation plots of all three methods against each other and Lamin B1 data. As expected, CscoreTool performs comparably with *dcHiC* and HOMER in compartment detection. We reflect these changes in our two different sections of the manuscript as follows:

“To establish the validity of the *dcHiC* results, we first compared compartment calls to two other common compartment-finding approaches: a canonical PCA-based approach (HOMER [33]), and the CscoreTool [17], a method that uses a likelihood function over a sliding window to infer compartment scores. The resulting compartment scores were highly similar among the three methods at 100Kb resolution using mouse ESC Hi-C data, with Pearson's $r=0.96$ between *dcHiC* and HOMER, 0.97 between HOMER and CscoreTool, and 0.98 between CscoreTool and *dcHiC* (**Figures 2A-C**). Similar to A/B compartment decomposition from Hi-C data, association with the nuclear lamina (or radial position) is another strong indicator of a broad-level chromatin state with heterochromatin localizing at the periphery and euchromatin at the nucleus center. All three methods also showed strong negative correlation with Lamin B1 data, confirming the previous findings [27, 36], with R-values of -0.91, -0.89, and -0.90 for *dcHiC*, HOMER, and CscoreTool, respectively (**Figures 2D-F**). We further plotted the compartment scores for chromosome

2 and chromosome 6 for ESCs and NPCs from *dcHiC*, HOMER and CscoreTool alongside Lamin B1 association signal confirming the high concordance (**Figure 2G-H**). These results established that *dcHiC*, similar to existing approaches, accurately captures compartment patterns. ”

Figure 2: Comparison of *dcHiC* compartment scores with HOMER compartment scores and Lamin B1 association data. (A-C) Genome-wide comparison of *dcHiC*, HOMER, and Cscore compartment scores against each other for mouse ESCs. **(D-F)** Genome-wide comparison of *dcHiC*, HOMER, and Cscore compartment scores against Lamin B1 profiles for mouse ESCs. **(G-H)** Browser views of the compartment scores from three different methods and Lamin B1 signal in chromosomes 6 and 2 in mouse ESC. Arrows highlight a subset of regions where the compartment assignment was not consistent among the three methods. **(I)** Genome-wide runtimes for compartment calling by each of the three methods at 10Kb, 25Kb, 40Kb, 50Kb, and 100Kb resolution for mESC Hi-C data. The runtimes include a sequential run of compartment calling for each chromosome and this is repeated for two pseudo-replicates of mESC data with runtimes summed up. **(H)** Genome-wide runtimes for 50kb resolution mESC Hi-C maps at 10%, 20%, 40%, 60%, 80%, and 100% down-sampling rate (100% = 500 million reads) for two pseudo-replicates (similar to Fig. 2I).

2. In these comparisons, the authors should compare and report standard benchmarks, i.e. time and memory usage at various resolutions and sequencing depths.

We have now done this and incorporated the results in Figure 2 and in Supplementary Tables S1-S4. We added a new results section that outline our findings:

“Performance evaluation of compartment calling by *dcHiC* and other approaches

Next, we assessed the resource utilization of *dcHiC* against HOMER and CscoreTool for compartment calling, a prerequisite to differential compartment analysis as well as the major bottleneck for high-resolution analysis in general. We evaluated the time and memory utilization of these three methods using two mouse ESC pseudo-replicates (~500M reads each), from which we generated contact maps at 5 different resolutions and 6 different sequencing depths (30 combinations; **Supplemental Information, Table S1-4**). In **Figure 2I-J**, we plotted genome-wide runtimes at 100% sampling rate for 5 different resolutions and for 50kb resolution at 6 different down-sampling rates showing that *dcHiC* runs 4-13x faster than CscoreTool and 22-33x faster than HOMER across these conditions. Across all read depths and all resolutions we tested, *dcHiC* ran 1.3-15x faster than CscoreTool and 10-52x faster than HOMER genome-wide (**Supplementary Tables S1-2**). **Figure 2J** also demonstrated that *dcHiC* scales better with increasing sequencing depth. With respect to memory use, at full read depth and 100kb resolution, CscoreTool had a lower peak memory (~0.24Gb) usage than *dcHiC* (~0.34Gb) and HOMER (~1.2Gb). For resolutions of 50Kb, 40Kb and 25Kb Hi-C data at 100% sequencing depth, all the three tools were within 30% of each other (~1.13Gb, ~1.25Gb and ~1.3Gb for CscoreTool, *dcHiC* and HOMER, respectively) with CscoreTool utilizing the least amount memory for computing the compartment score at every resolution (**Supplementary Tables S3-4**). For these time and memory profiling, we ran all tests genome-wide, and used one CPU per chromosome (Intel Xeon Gold 6252 CPU @ 2.10GHz). Running HOMER genome-wide at 10Kb resolution did not finish after 100 hours of compute time for ESC data.”

3. The authors should evaluate the effects of sequencing depth on compartment calls and differential compartment calls by downsampling maps. They should then report the false-positive / false-negative rates compared to the high-read depth map. This should be done for a range of resolutions and sequencing depths.

We agree with the reviewer that testing robustness of our method with respect to read depth and resolution is very important. To this end, as suggested, we performed a series of new analyses using a range of resolutions and sequencing depths, which led to the addition of a new section and Supplementary Figures S4 and S5 as well as Supplementary Tables S5-S9. These results can be summarized as:

- We observe that the compartment calls (before differential analysis) are highly concordant within 100% to 40% (500M-200M reads) down-sampling rate ($R > 0.9$; **Supplementary Table S6**).
- We see that *dcHiC* is robust to type-1 error when comparing replicates at different resolutions and read depths (**Supplementary Table S5**).
- We see no false positive differential compartment calls among samples with 100% to 60% down-sampling (500M-300M reads), whereas false positive calls appear when 20% or 10% down-sampled replicates are introduced providing us guidelines for proper use of *dcHiC* (**Supplementary Table S8**).
- We found that there is a high recall of differential compartments ($>80\%$) called from full sequencing depth (“ground truth”) for down-sampling rates of 40% or more (**Supplementary Figure S4**) with similar trends for other resolutions including 50kb, 40kb and 25kb (**Supplementary Figure S5**).
- These analyses allowed us to provide guidelines about the sequencing depth and other requirements for proper utilization of *dcHiC*.
- Overall, *dcHiC* results are quite robust for most relevant settings.

“Robustness of *dcHiC* differential compartment calls

Next, we sought to see how well the pairwise differential compartment calls between different Hi-C profiles are preserved through down-sampling and at different resolutions. We used the 4 ESC biological replicates (230M-1.2B reads) and the 3 NPC biological replicates (720M-1.5B reads) at their full sequencing depth (100%) and then down-sampled each replicate separately to 5 different read depths: 80%, 60%, 40%, 20%, and 10%. To profile the effects of down-sampling, we first compared ESC and NPC replicates at each read depth using 100Kb resolution contact maps. We found that there is a high recall of differential compartments ($>80\%$) called from full sequencing depth (“ground truth”) for down-sampling rates of 40% or more (**Supplementary Figure S4**). In order to also assess the role of resolution in recall of differential compartments, we repeated the same down-sampling experiments for 4 other resolutions: 50Kb, 40Kb, 25Kb, and 10Kb. We observed that except from 10Kb resolution, all other cases were similar to 100Kb where 40% down-sampling still led to a high recall ($>75\%$), whereas for 10Kb resolution, the results at 60% down-sampling had a recall of 80% that dropped to 61% for 40% down-sampling (**Supplementary Figure S5A-D**). Given that the sequencing depth for each replicate varied between 200M to 1.5B reads, we believe that with replicates of at least 80-100M reads, differential compartment analysis at 25Kb or lower resolution can be carried out with high/acceptable recall of all compartmentalization changes that can be detected with deeper sequencing. Later in this section we analyze the role of sequencing depth and resolution on precision of differential compartment detection.

...

To further assess the type-1 error rate, we carried out a series of differential compartment analysis between mouse ESC pseudo-replicates (2 replicates) at different resolutions (100Kb, 50Kb, 40Kb, 25Kb and 10Kb) and down-sampling rates (100%, 80%, 60%, 40%, 20% and 10% of 500 million sequencing depth). We measured the number of differential compartments when running two down-sampled replicates against each other at different resolutions and our results indicate that *dcHiC* is robust to type-1 error when comparing replicates at different resolutions and read depths (**Supplementary Table S5**). We also evaluated the type-1 error rate, when two mouse ESC pseudo-replicate Hi-C maps of different sequencing depth are compared by *dcHiC*. Across the 21 comparisons, we first see that the compartment calls are highly correlated within 100% to 40% (500M-200M reads) of read depth (**Supplementary Table S6**). The correlations with high read depth samples drop substantially for 20% (100M reads) and further for 10% (50M reads) sample. We noticed this occurred because compartment scores for some chromosomes started to not fully reflect the compartmentalization pattern at lower read depths. Removing the 5 chromosomes (chr 4, 5, 14, 17, X) with such issues, we see correlations at lower read depths improve, however not to the point that we highly concordant (correlation >0.9) compartment calls between two pseudo-replicates (**Supplementary Table S7**). While evaluating the false-positive calls, we first observed that correlations between compartment scores are closely related to the number of differential calls. When we utilized *dcHiC* to find differential compartments (i.e., false positive calls) between two replicates of different sequencing depth by down-sampling Hi-C maps at 100Kb resolution, we see no false positives up among samples with 100% to 60% down-sampling (500M-300M reads) (**Supplementary Table S8**). We also do not obtain any false positives even for lower depth samples when they are compared against the sample with the same rate of down-sampling. However, a substantial number of false positive differential calls appear when 20% or 10% down-sampled samples are compared to higher depth samples (**Supplementary Table S8**). Like compartmental correlations, here also when we filter out the 5 chromosomes with issues in compartment calls at low read depths, we see that the false positive rates dramatically improve for 40% and for 20% down-sampled samples (**Supplementary Table S9**). Based on these results, we believe compartment scores and differential compartment calls are robust when comparing Hi-C maps that are sufficiently sequenced (100M or more reads) and are within 2-3-fold read depth of each other. “

Supplementary Figure S4: Effect of sequencing depth on pairwise differential compartment calls (mouse ESC vs NPC) by *dcHiC*. The panel shows the percentages of differential compartments detected at 100% sequencing depth that are recovered after down-sampling the biological replicates (4 for ESC and 3 for NPC) at different levels (80% to 10%) for 100kb resolution analysis.

Supplementary Table S5: Effect of resolution and sequencing depth on type-1 error rate. The table shows differential compartments identified by comparing two pseudo-replicates against each other using *dcHiC* on mouse ESC. The comparison is repeated at different sequencing depths (rows) and resolutions (columns).

Down sampling rates (100% = 500 million)	ESC (Pseudo-replicate 1 vs 2)				
	100Kb	50Kb	40Kb	25Kb	10Kb
100%	0	0	0	0	0
80%	0	0	0	0	0
60%	0	0	0	0	0
40%	0	0	0	0	0
20%	0	0	0	0	0
10%	0	0	0	2	6

Supplementary Table S8: Effect of differential down-sampling (different down-sampling rates for each replicate) on the differential compartment detection by *dcHiC* between two mouse ESC pseudo-replicates.

Differential compartments	Down sampling rates (100% = 500 million)	ESC Pseudo-replicate 2					
		100%	80%	60%	40%	20%	10%
ESC Pseudo-replicate 1	100%	0	0	0	0	247	827
	80%	0	0	0	84	353	907
	60%	0	0	0	124	348	1215
	40%	0	84	124	0	335	1027
	20%	247	353	348	335	0	737
	10%	827	907	1215	1027	737	0

4. The manuscript relies heavily on correlation with other features, such as histone marks, replication timing, etc to validate the compartment calls and differences. However, they validate that the differences between Homer and dHiC (and CScores) reflect differences in the Hi-C. They should show Hi-C examples of the loci where Homer and dHiC differ. Currently, it is difficult to tell if Homer or dHiC is more accurate. Homer detects more differences, which appear to also correspond to differences in correlated features (Fig 3F&H). This dramatically dampens my enthusiasm.

We thank the review for raising this point. Figure 3F in the previous (and new) version of the manuscript showed the overlap between *dcHiC* and HOMER pairwise differential calls. We have now expanded characterization of the differences between two methods by:

a. Specific analysis of inconsistent compartment calls (A vs B) between *dcHiC* and HOMER:

“Next, we further analyzed the 4-7% of the genome that is labeled in opposite compartments by *dcHiC* in comparison to HOMER for ESC and NPC (**Supplementary Figure S1A-B**). Overall, *dcHiC-B* but HOMER-A regions (~1% for ESC and NPC) showed positive lamin B1 signal and lower gene expression levels compared to *dcHiC-A* but HOMER-B regions (**Supplementary Figure S1C-D**). The latter set (3% for ESC and 6% for NPC) had a mix of regions with positive and negative lamin B1 association as well as gene expression values that are lower than constitutive A but higher than constitutive B compartment regions (compare to **Figure 3**) suggesting a weak compartmentalization for these regions into either A or B compartment.”

heterochromatin (or B) assignment. (**D**) Shows the expression distribution of genes overlapping with the inconsistently labelled regions by *dcHiC* and HOMER.

b. Further analysis of differences in differential compartment calls from *dcHiC* and HOMER:

“To compare the calls made by the two different methods, we plotted the absolute differences in lamin B1 signal, replication timing and log2 gene expression values of all the reported differential compartments (**Figure 3G**) or method-specific differential compartments (**Figure 3H**) for each method. These results show that *dcHiC* differential compartments are significantly (unpaired t-test p-values < 0.05) enriched for regions with higher ESC and NPC differentials for lamin B1 association and replication timing signals although both methods captured regions with signal differences in all three measures. We also performed differential expression analysis between ESCs and NPCs to map the differentially expressed (DE) genes (DEseq2 [41], FDR<0.05, fold change>4) on the differential compartments. We observed that *dcHiC* differential compartment bins were enriched in the number of DE genes (**Figure 3I**) as well as the fold change (log2) and significance (DESeq2) of the difference for those DE genes (**Supplementary Figure S2A-B**). Further, we also looked at the average number of histone modification peak (MACS2 p-value < 1e-5) differences between ESC and NPC per 100Kb for the regions from *dcHiC* and HOMER’s differential calls (**Supplementary Figure S3**). For all three different histone marks (H3K4me1, H3K4me3, H3K27ac), we observed a higher number of peak differences per 100Kb for *dcHiC* compared to HOMER. ”

Supplementary Figure S3: Average difference in absolute number of histone peaks (MACS2 p-value < 1e-5) per 100Kb of differential bins within *dcHiC* and HOMER differential compartments. **(A)** Shows the absolute difference of average number H3K27ac peaks between ESC and NPC per 100Kb of *dcHiC* and HOMER differential compartments. The result shows that there is more difference in H3K27ac peaks per 100Kb of differential compartments identified by *dcHiC*. **(B-C)** The result shows that are more differences in both H3K4me3 and H3K4me1 peaks per 100Kb of differential compartments called by *dcHiC* compared to HOMER.

c. Example regions where *dcHiC* and HOMER calls differ:

For ESC vs NPC comparison, we have now scanned the differential compartments specifically called by either *dcHiC* or HOMER and extracted examples of such cases (see below). We have also added the LaminB1 signal (First two tracks) and gene expression values (TPM, last two tracks) along with compartment signal track to better annotate these differences. Following are the few example regions where *dcHiC* and HOMER calls differ. Consistent with the higher number of differential calls by HOMER, these examples also suggested that HOMER is more sensitive to detect changes, including some that are not visually appreciable (e.g., chr6 and chr10). This is likely related to the specific t-test (moderated t-test) that is used by limma for microarray analysis, which HOMER employs for differential compartment calling. We also found *dcHiC*-specific differences that involve substantial changes in compartmentalization that are missed by HOMER (e.g., chr 13). This being said, there are examples of changes that are HOMER-specific and involve visible differences, which we believe are missed by *dcHiC* (e.g., chr18). We hope these examples highlight the advantages and limitations of each method sufficiently.

5. They should also examine the compartment scores independently by computing the intensity scores (AA)/(AB) of the actual Hi-C signal for these regions.

We appreciate the reviewer's suggestion and for pointing out an interesting analysis. We now have plotted the distribution of distance normalized intensity scores (Observed/Expected interaction) of AA divided by AB (as suggested) for all differential A compartments from ESC (panel A below) and from NPC (panel B below). We observed that for both cell types of the differential A-compartments have overall weaker AA/AB intensity scores compared to random non-differential A-compartments. When we compared this trend between *dcHiC* and HOMER differential calls, we did not see any substantial differences with differential calls from both cases leading to significantly lower AA/AB scores compared to random non-differential A compartment regions.

6. Similar to above, Gorkin et al identified far more differences than *dcHiC*, and it is not clear whether or not these are false positives. The authors should show the Hi-C signal at Gorkin exclusive vs *dcHiC* exclusive differences. Additionally, they should test whether Gorkin specific differences correlate with differences in the histone marks that they published.

We acknowledge that the concern raised by the reviewer is valid and we have performed additional analyses in this revision to address this issue. This led to addition of two new supplementary figures (Supplementary Figure S10 and S11) and their discussion in the main text. We also provide below example regions where Gorkin et al. specific differences are illustrated.

- a. **Analysis of the overlap between *dcHiC* and Gorkin et al differential compartment calls and variable histone modification regions previously identified across different LCLs:**

“We then asked whether the identified differential compartment regions were enriched in regions with variability in histone marks (H3K27ac, H3K4me3, H3K4me1 and H3K27me3) across different individuals. The variable histone modification regions/peaks identified for human LCLs by Kasowski et. al. 2013 [65] were mapped on differential compartments identified from *dcHiC* and by Gorkin et. al. [66]. Using the non-differential compartment regions as background for each method, we observed nearly no enrichment for regions called differential only by Gorkin et. al. (**Supplementary Figure S10**) while calls from *dcHiC* showed 26-45% of enrichment. The proportion of differential calls that overlapped with at least one variable region for each histone mark was also substantially higher for regions from *dcHiC* in comparison to Gorkin et. al. specific regions (**Supplementary Figure S11A-D**).”

Supplementary Figure S10:

Enrichment of differential compartments from either *dcHiC* or Gorkin et. al. paper that overlap at least one variable histone mark region/peak identified by Kasowski et. al (2013). The enrichment is computed with respect to all non-differential compartment regions for each method. We assess the enrichment for all calls from *dcHiC* in comparison to calls that are specific to Gorkin et. al.

Supplementary Figure S11:

The number of differential compartments from each method that overlap with a variable histone mark region (similar to Figure S10) for (A) H3K27ac, (B) H3K4me3, (C) H3K4me1, and (D) H3K27me3.

b. Example regions where Gorkin et al. calls cover a large fraction of the region without accompanying visual differences in compartment scores

- ~20Mb region of chromosome 16

- ~25Mb region of chromosome 18

Although the above examples were common and representative of the overall trend, one exception to this was a ~10Mb region on chromosome 4 (the chromosome with the highest fraction of differential calls by *dcHiC*) where *dcHiC* calls covered a larger fraction of the region compared to Gorkin et al. These tracks can be interactively browsed genome-wide using the link:

https://ay-lab.github.io/dcHiC/Gorking_hg19_40Kb.RefineY_Rconf90_FDR10.pcOri.html

7. Authors should show what the Gorkin compartment calls look like at the NR2F2 and THEMIS regions to see how dChIC compares.

The regions in question are plotted below. We plotted the 15Mb region around *NR2F2* and 25Mb region around *THEMIS*. There was a better consistency between *dChIC* and Gorkin et al. calls for the *THEMIS* locus whereas *NR2F2* locus showed a similar pattern to the other examples above where Gorkin et al calls covered most of the region.

As mentioned above, the same tracks went into this visualization can be interactively browsed from the link:

https://ay-lab.github.io/dcHiC/Gorking_hg19_40Kb.RefineY_Rconf90_FDR10.pcOri.html

Reviewer #2 (Remarks to the Author):

Reviewer comments are in red, our responses are in black and copy-pasted text from the manuscript are colored in blue

In this manuscript, Chakraborty and colleagues proposed *dcHiC* to perform differential analysis of A/B compartments, from Hi-C data. Overall, there are many merits of the manuscript. The method is simple yet well motivated. Differential compartment testing is useful for Hi-C data analysis. The manuscript is well organized and easy to follow. The authors applied their methods to three datasets, leading biologically findings that are expected and supported from auxiliary transcriptomics or epigenomic data (although I didn't find anything particularly note-worthy or novel in terms of biological insights). The GitHub site is well constructed and the examples from analyses performed in the manuscript are valuable for users.

We thank the reviewer for their comments and positive assessment. Below we would like to highlight the most significant changes in this revision for your reference:

1. As suggested by all reviewers, we have now extensively studied the impact of sequencing depth and resolution on the compartment calls and differential compartment calls from *dcHiC* including a false positive rate analysis using replicates and recovery analysis using downsampling. These analyses demonstrated that our results are quite robust for most relevant settings and allowed us to provide guidelines about the sequencing depth and other requirements for proper utilization of *dcHiC*.
2. We now also applied *dcHiC* on single-cell Hi-C data demonstrating its utility in comparing different clusters and/or time points to study dynamic changes in compartmentalization from pseudo-bulk Hi-C profiles from as low as 80 single cells. Our analysis of the Tan et al. (Cell, 2021) single-cell Hi-C data from two regions of post-natal developing mouse brain (each with 6 time points) allowed us to reveal important genes related to synapse assembly and adult brain development that have dynamic and tissue-specific compartmentalization changes, which were not highlighted by the original publication.
3. As requested by multiple reviewers, we have now substantially expanded our comparative analysis to HOMER, Gorkin et al and added Cscore into these comparisons. We have profiled runtimes and memory utilization of each approach for compartment detection in multiple settings and resolution, highlighting the clear advantage of *dcHiC* for high-resolution/finer-scale compartment analysis.
4. We also extensively compared the overlap of differential compartments identified by different approaches with changes/variation in orthogonal measurements from matched samples including gene expression, histone modifications and lamin B1 signal. These results confirmed and strengthened our previous findings that *dcHiC* differential compartments are consistently more enriched in variation of other biological signals.
5. To address confusions from multiple reviewers, we have added a detailed discussion of HOMER and *dcHiC* in terms of their specific implementation of compartment calling and the source of *dcHiC*'s performance improvement including the use of more efficient data structure.

6. We have expanded our documentation in Wiki page of our Github repository (<https://ay-lab.github.io/dcHiC/>) to include single-cell analysis of Tan et. al and provided two demos with accompanying test data to show how *dcHiC* can be used for bulk and single-cell Hi-C data analysis: <https://github.com/ay-lab/dcHiC/demo>

7. We have also added additional analyses showing: i) the overlap of *dcHiC* results with subcompartment calls and their differences, ii) consistency of *dcHiC* calls across different resolutions including high-resolution such as 10kb and 20kb maps when possible, iii) the dynamic changes through time series analysis of specific lineages in the mouse hematopoietic lineage.

However, the biggest concern is that the method needs to be compared with the state-of-the-art alternatives. The authors started by showing that the method generates results consistent with the standard PCA analysis. How about applying PCA analyses to the datasets they analyzed and what would the results look like? Would the standard PCA approach lead to similar findings? What does *dcHiC* offer that would have been missed by PCA?

We would like to address this concern, one shared by multiple reviewers, by clarifying *dcHiC*'s precise compartment calling method. In R, the standard way to compute eigenvalues is the `eigen()` or `svd()` function. However, when the matrix becomes large, these functions can be very time-consuming: the complexity to calculate all eigenvalues of an 'n' by 'n' matrix is $O(n^3)$. While in real applications including compartment analysis, we usually only need to compute a few eigenvectors, for example to visualize high dimensional data using PCA. The same thing happens in Singular Value Decomposition (SVD). It is often the case that only a Partial SVD or Truncated SVD is needed, and moreover the matrix is usually stored in sparse format. In the background, both our method and other popular programs like HOMER uses the eigendecomposition of the correlation matrices to find the principal components. HOMER in particular uses the default 'eigen' function of R to perform the PCA. Due to its inefficient handling of large matrices, however, the principal component calculation of Hi-C maps at higher resolution becomes impractical to run with eigen function. *dcHiC* implements the 'bigstatsr' package in R and uses Filebacked Big Matrices (FBM) via memory-mapping. This allows, for instance, matrix operations that includes memory and time-efficient low-rank approximation of standard PCA calculation based on the algorithm in C++ spectra library. Given an 'n' by 'n' large matrix A, the 'big_randomSVD' from 'bigstatr' package can calculate a specified number of eigenvectors of A. Users can specify the selection criterion by an argument which allows computing only the k largest eigenvalues and their corresponding eigenvectors.

One other popular method for compartment calling is CscoreTool, an approach that completely avoids the canonical PCA and instead uses a sliding genomic window that predicts whether a given region is A or B as a log-likelihood function of the number of interaction counts in the region. As such, *dcHiC* and HOMER are based on the same method (with different implementations), while CscoreTool is a different approach for compartment detection. We have now done the comparison and incorporated the results in Figure 2. In the revised Figure 2, we show correlation plots of all three methods against each other and Lamin B1 data. As expected, CscoreTool performs comparably with *dcHiC* and HOMER in compartment detection. We reflect these changes in our two different sections of the manuscript as follows:

“To establish the validity of the *dcHiC* results, we first compared compartment calls to two other common compartment-finding approaches: a canonical PCA-based approach (HOMER [33]), and the CscoreTool [17], a method that uses a likelihood function over a sliding window to infer compartment scores. The resulting compartment scores were highly similar among the three methods at 100Kb resolution using mouse ESC Hi-C data, with Pearson’s $r=0.96$ between *dcHiC* and HOMER, 0.97 between HOMER and CscoreTool, and 0.98 between CscoreTool and *dcHiC* (Figures 2A-C). Similar to A/B compartment decomposition from Hi-C data, association with the nuclear lamina (or radial position) is another strong indicator of a broad-level chromatin state with heterochromatin localizing at the periphery and euchromatin at the nucleus center. All three methods also showed strong negative correlation with Lamin B1 data, confirming the previous findings [27, 36], with R-values of -0.91, -0.89, and -0.90 for *dcHiC*, HOMER, and CscoreTool, respectively (Figures 2D-F). We further plotted the compartment scores for chromosome 2 and chromosome 6 for ESCs and NPCs from *dcHiC*, HOMER and CscoreTool alongside Lamin B1 association signal confirming the high concordance (Figure 2G-H). These results established that *dcHiC*, similar to existing approaches, accurately captures compartment patterns.”

Figure 2: Comparison of *dcHiC* compartment scores with HOMER compartment scores and Lamin B1 association data. (A-C) Genome-wide comparison of *dcHiC*, HOMER, and Cscore compartment scores against each other for mouse ESCs. **(D-F)** Genome-wide comparison of *dcHiC*, HOMER, and Cscore compartment scores against Lamin B1 profiles for mouse ESCs. **(G-H)** Browser views of the compartment scores from three different methods and Lamin B1 signal in chromosomes 6 and 2 in mouse ESC. Arrows highlight a subset of regions where the compartment assignment was not

consistent among the three methods. **(I)** Genome-wide runtimes for compartment calling by each of the three methods at 10Kb, 25Kb, 40Kb, 50Kb, and 100Kb resolution for mESC Hi-C data. The runtimes include a sequential run of compartment calling for each chromosome and this is repeated for two pseudo-replicates of mESC data with runtimes summed up. **(H)** Genome-wide runtimes for 50kb resolution mESC Hi-C maps at 10%, 20%, 40%, 60%, 80%, and 100% down-sampling rate (100% = 500 million reads) for two pseudo-replicates (similar to Fig. 2I).

We also show in the first section that *dcHiC*'s compartment calls are closely aligned with HOMER's (PCA) and whenever there is a mismatch of calls between the two methods the independent evidence (gene expression and lamin association) mostly supports *dcHiC* calls, although we note that such regions with method-specific compartment labels tend to have weak compartmentalization in general.

“Next, we further analyzed the 4-7% of the genome that is labeled in opposite compartments by *dcHiC* in comparison to HOMER for ESC and NPC (**Supplementary Figure S1A-B**). Overall, *dcHiC*-B but HOMER-A regions (~1% for ESC and NPC) showed positive lamin B1 signal and lower gene expression levels compared to *dcHiC*-A but HOMER-B regions (**Supplementary Figure S1C-D**). The latter set (3% for ESC and 6% for NPC) had a mix of regions with positive and negative lamin B1 association as well as gene expression values that are lower than constitutive A but higher than constitutive B compartment regions (compare to **Figure 3**) suggesting a weak compartmentalization for these regions into either A or B compartment.”

Supplementary Figure S1: Inconsistent A/B compartment assignments between *dcHiC* and HOMER. **(A-B)** Shows the number (top) and fraction (bottom) of consistent and inconsistent A/B assignments by *dcHiC* and HOMER in mouse ESC and NPC Hi-C maps. **(C)** Shows the Lamin B1 signal of inconsistently labelled regions with *dcHiC*-B but HOMER-A regions showing lamin B1 association supporting their

heterochromatin (or B) assignment. **(D)** Shows the expression distribution of genes overlapping with the inconsistently labelled regions by *dcHiC* and HOMER.

Further, as suggested by the reviewer, to inspect the effect of the standard PCA analysis on the differential compartment calls we compared the dHiC results with and without quantile normalized PCA. As, expected about more than 90% of the differential calls overlapped with each other and almost all the major regions along with others that we discuss in the paper remained as significant. As a support to this claim, we are showing the differential calls between the two approaches in the following four chromosomes. The first two rows show the principal component of ESC and NPC Hi-C maps followed by the $-\log_{10}(\text{Padj})$ values from quantile normalized and raw PCA approach.

Since our compartment calls largely match those of other established methods included PCA-based, we would also like to reiterate the point to the reviewer that dcHiC does not reinvent the PCA analysis approach from scratch but instead taking compartment scores (possible to start from PCA-based scores previously computed such as Gorkin et. al. data here) and performing a statistical analysis on their differences with or without replicates. The framework we developed here provides a systematic way to identify differential compartments by performing the standard or using any existing principal component analysis of Hi-C data and visualize these differences in different scenarios, including multiway, hierarchical, time-series and single-cell settings. The novelty of dcHiC lies in implementing an integrative framework and an easy-to-use tool for comparative analysis of Hi-C maps using standard PCA data that identifies biologically relevant differences in compartmentalization across multiple cell types.

Other major comments are listed below.

(1) As a statistical testing method, the authors need to demonstrate the validity of the methods: that is, the authors need to first show protection of type-I error. The authors can do this by applying dcHiC to Hi-C datasets with replicates or randomly splitting high-depth Hi-C data into two sets. Under such scenarios, no differential compartments are expected, which would allow an evaluation of the method validity. Without establishing the validity, a statistical hypothesis testing is meaningless.

We agree with the reviewer that testing robustness and validity of our method is very important. To this end, as suggested, we performed a series of new analyses using replicates from the same cell type or different cell types with a range of resolutions and sequencing depths, which led to the addition of a new section and Supplementary Figures S4 and S5 as well as Supplementary Tables S5-S9. These results can be summarized as:

- We observe that the compartment calls (before differential analysis) are highly concordant within 100% to 40% (500M-200M reads) down-sampling rate ($R > 0.9$; **Supplementary Table S6**).
- We see that *dcHiC* is robust to type-1 error when comparing replicates at different resolutions and read depths (**Supplementary Table S5**).
- We see no false positive differential compartment calls among samples with 100% to 60% down-sampling (500M-300M reads), whereas false positive calls appear when 20% or 10% down-sampled replicates are introduced providing us guidelines for proper use of *dcHiC* (**Supplementary Table S8**).
- We found that there is a high recall of differential compartments ($>80\%$) called from full sequencing depth (“ground truth”) for down-sampling rates of 40% or more (**Supplementary Figure S4**) with similar trends for other resolutions including 50kb, 40kb and 25kb (**Supplementary Figure S5**).
- These analyses allowed us to provide guidelines about the sequencing depth and other requirements for proper utilization of *dcHiC*.
- Overall, *dcHiC* results are quite robust for most relevant settings.

We added the following paragraph in the main manuscript to demonstrate the validity of the method in terms of its type-1 error rate -

“Robustness of *dcHiC* differential compartment calls

...

To further assess the type-1 error rate, we carried out a series of differential compartment analysis between mouse ESC pseudo-replicates (2 replicates) at different resolutions (100Kb, 50Kb, 40Kb, 25Kb and 10Kb) and down-sampling rates (100%, 80%, 60%, 40%, 20% and 10% of 500 million sequencing depth). We measured the number of differential compartments when running two down-sampled replicates against each other at different resolutions and our results indicate that *dcHiC* is robust to type-1 error when comparing replicates at different resolutions and read depths (**Supplementary Table S5**). We also evaluated the type-1 error rate, when two mouse ESC pseudo-replicate Hi-C maps of different sequencing depth are compared by *dcHiC*. Across the 21 comparisons, we first see that the compartment calls are highly correlated within 100% to 40% (500M-200M reads) of read depth (**Supplementary Table S6**). The correlations with high read depth samples drop substantially for 20% (100M reads) and further for 10% (50M reads) sample. We noticed this occurred because compartment scores for some chromosomes started to not fully reflect the compartmentalization pattern at lower read depths. Removing the 5 chromosomes (chr 4, 5, 14, 17, X) with such issues, we see correlations at lower read depths improve, however not to the point that we highly concordant (correlation >0.9) compartment calls between two pseudo-replicates (**Supplementary Table S7**). While evaluating the false-positive calls, we first observed that correlations between compartment scores are closely related to the number of differential calls. When we utilized *dcHiC* to find differential compartments (i.e., false positive calls) between two replicates of different sequencing depth by down-sampling Hi-C maps at 100Kb resolution, we see no false positives up among samples with 100% to 60% down-sampling (500M-300M reads) (**Supplementary Table S8**). We also do not obtain any false positives even for lower depth samples when they are compared against the sample with the same rate of down-sampling. However, a substantial number of false positive differential calls appear when 20% or 10% down-sampled samples are compared to higher depth samples (**Supplementary Table S8**). Like compartmental correlations, here also when we filter out the 5 chromosomes with issues in compartment calls at low read depths, we see that the false positive rates dramatically improve for 40% and for 20% down-sampled samples (**Supplementary Table S9**). Based on these results, we believe compartment scores and differential compartment calls are robust when comparing Hi-C maps that are sufficiently sequenced (100M or more reads) and are within 2-3-fold read depth of each other. ”

Supplementary Table S5: Effect of resolution and sequencing depth on type-1 error rate. The table shows differential compartments identified by comparing two pseudo-replicates against each other using *dcHiC* on mouse ESC. The comparison is repeated at different sequencing depths (rows) and resolutions (columns).

Down sampling rates (100% = 500 million)	ESC (Pseudo-replicate 1 vs 2)				
	100Kb	50Kb	40Kb	25Kb	10Kb
100%	0	0	0	0	0
80%	0	0	0	0	0
60%	0	0	0	0	0
40%	0	0	0	0	0
20%	0	0	0	0	0
10%	0	0	0	2	6

Supplementary Table S8: Effect of differential down-sampling (different down-sampling rates for each replicate) on the differential compartment detection by *dcHiC* between two mouse ESC pseudo-replicates.

Differential compartments	Down sampling rates (100% = 500 million)	ESC Pseudo-replicate 2					
		100%	80%	60%	40%	20%	10%
ESC Pseudo-replicate 1	100%	0	0	0	0	247	827
	80%	0	0	0	84	353	907
	60%	0	0	0	124	348	1215
	40%	0	84	124	0	335	1027
	20%	247	353	348	335	0	737
	10%	827	907	1215	1027	737	0

(2) The authors applied an outlier detection approach when performing differential compartment analysis, which appears somewhat ad hoc. A standard approach is to use parametric or non-parametric ANOVA analysis, which is computationally fast, as well as statistically straightforward and extensively used.

We thank the reviewer for raising this question and allowing us to clarify our reasoning. In the field of multivariate statistics, Mahalanobis distance is one of the most straightforward and common measures to detect outliers [1] and has been extensively used in other fields with success [2-13]. Outlier detection in multidimensional space requires scaling the contribution of individual variables to the distance value according to the variability of each observation [8, 14]. Mahalanobis distance uses a covariance matrix of variables to find the distance between data points and the center which works robustly on multivariate data. Indeed, we utilized an R function that implements Minimum Covariance Determinant (MCD) procedure that has been shown to improve multivariate outlier detection compared to basic Mahalanobis distance [7]. We now clarify this in the methods by adding the below text.

“For calculation of the covariance matrix, we utilize *covrob* function of the R package *robust*, which implements Minimum Covariance Determinant (MCD) procedure that has been shown to improve multivariate outlier detection.”

References –

1. Brereton, G., R. The Mahalanobis distance and its relationship to principal component scores. *Journal of Chemometrics* 29(3), (2015), 143-145.
2. J. Majewska: Identification of multivariate outliers problems and challenges of visualization methods. *Informatyka i Ekonometria* 4 (2015), 69–83.
3. W. Dai and M. G. Genton: Multivariate functional data visualization and outlier detection. *Journal of Computational and Graphical Statistics* 27(4) (2018), 923–934.
4. D. M. Hawkins: Identification of Outliers. Chapman and Hall, London, 1980.
5. Leys, C., Delacre, M., Mora, Y. L., Lakens, D., & Ley, C. (2019). How to Classify, Detect, and Manage Univariate and Multivariate Outliers, With Emphasis on Pre-Registration. *International Review of Social Psychology*, 32(1), 5.
6. Dashdondov, K., Kim, MH. Mahalanobis Distance Based Multivariate Outlier Detection to Improve Performance of Hypertension Prediction. *Neural Process Lett* (2021).
7. C. Leys, O. Klein, Y. Dominicy and C. Ley: Detecting multivariate outliers: Use a robust variant of the Mahalanobis distance. *Journal of Experimental Social Psychology* 74 (2018), 150–156.
8. G. M. Mimmack, S. Mason and J. Galpin: Choice of distance matrices in cluster analysis: defining regions. *Journal of Climate* 14 (2001), 2790–2797.
9. P. Rousseeuw, M. Debruyne, S. Engelen and M. Hubert: Robustness and outlier detection in Chemometrics. *Critical Reviews in Analytical Chemistry* 36(3), (2006), 221–242.
10. P. J. Rousseeuw and B. C. van Zomeren: Robust distances: simulation and cutoff Values. In: *Directions in Robust Statistics and Diagnostics, Part II.* (W. Stahel, S. Weisberg, eds.), Springer-Verlag, New York, 1991.
11. N. Haldar K. Farrukh A. Aftab and H. Abbas: Arrhythmia classification using Mahalanobis distance based improved fuzzy C-Means clustering for mobile health monitoring systems. *Neurocomputing*, 220 (2016), 221–235.
12. N. G. Sharma, et. al. Reconstruction of hit time and hit position of annihilation quanta in the J-PET detector using the Mahalanobis distance. *Nukleonika* 4 (2015), 765–769.
13. S. Stckl and M. Hanke: Financial applications of the Mahalanobis distance, *SSRN Electronic Journal* 1(2) (2014), 78–84.
14. Ghorbani, H. (2019). Mahalanobis distance and its application for detecting multivariate outliers. *Facta Universitatis Series Mathematics and Informatics* 34(3):583.

(3) The authors should be applauded for attempting finer-resolution in at least one dataset. Have the authors applied their methods at higher-resolution to the other two datasets. How would the depth of Hi-C data affect the finest resolution recommended?

We thank the reviewer for his/her comment. To be clear, *dcHiC* implements ‘bigstatsr’ package in R and uses Filebacked Big Matrices (FBM) via memory-mapping for instance matrix operations that includes time-efficient algorithm for the low-rank approximation of standard PCA calculation. This allows *dcHiC* to perform eigendecomposition and find compartment scores at finer resolution.

We have indeed applied *dcHiC* to the other datasets at high resolution. For instance, apart from the ESC vs NPC pairwise differential compartment comparison, we carried out the 100Kb to 20Kb differential compartment call comparison within 10-way mouse hematopoietic Hi-C maps. At 20Kb

resolution and with the relatively low read depth for this data, it became challenging to get compartment signals correctly for a few chromosomes such as chromosome 1 in MEP, chromosome 2 in MPP and chromosome X in general. After excluding these chromosomes, we were left with 25,035 differential compartment bins (~25% of the total genome). Our 100Kb analysis of the same set of Hi-C maps led to a similar percentage ~24% (no chromosomes excluded) of the genome as part of differential compartments. To check if both analyses detected similar regions, we calculated the fraction of common differential regions between the 100Kb and 20Kb dataset. We found around ~98% of the 100Kb differential compartments (4932 out of 5042, excluding chr1, 2 and X) and ~70% of the 20Kb differential compartments (17,635 out of 25,035) overlapped with each other. Since we start Gorkin et al. data analysis directly from 40kb compartment scores computed in the original paper (in order to have a fair comparison to the variable compartment regions reported by in the same paper), we did not perform a higher resolution analysis on this dataset. The sequencing depth of that dataset would have likely limited such analysis even if we reanalyzed the data from scratch.

We have also performed an extensive analysis of compartment and differential compartment calls and added a new section and supplementary material as we have elaborated in our response to this reviewer's Major comment #1 above. We hope that these results sufficiently demonstrate dcHiC's robustness across different resolutions and ability to perform high-resolution analysis when the contact maps have sufficient depth.

(4) Have the authors considered extending the approach to single cell Hi-C data? A recent study (PMID: 33484631) has suggested that compartment score can reflect 3D chromatin structure in single cells. Can dcHiC handle the sparse single Hi-C data?

This was an excellent suggestion! We now added a whole new section with an accompanying main figure (Figure 8) and three supplementary figures (S12-14) that focuses on analysis of compartments from single-cell Hi-C data during post-natal mouse brain development (Tan et. al. 2021). This analysis revealed dcHiC is able to call compartments and perform differential analysis from pseudo-bulk profiles of as low as 80 cells per condition as well as identify important known and novel gene regions with dynamic patterns of compartmentalization change during three developmental stages (6 timepoints) from two different brain regions.

From DISCUSSION

“Genomic compartment identification from single-cell Hi-C maps is still a major issue in this field due to sparsity of single-cell contact maps even at coarse resolution. A recent paper (Zhang et. al. 2021) demonstrated the challenges introduced by different technical and biological factors in reliably calling and comparing A/B compartments across single cells. Here, by studying a recent single-cell Hi-C (Dip-C) data characterizing post-natal dynamics of mouse brain development in two brain regions, we showed that *dcHiC* is able to call compartments and perform differential analysis from pseudo-bulk profiles of as low as 80 cells per condition. We observed a higher number of differential compartments in cortex region compared to hippocampus. The differential compartments were overlapping with cell-type specific marker genes and previously known variable genes identified

through single-cell transcriptomic analysis. The time series clustering of the differential compartments across three stages ('Early', 'Mid' and 'Late') of development helped us identify other important genes in differential compartments that were not captured in the original paper. Although we highlighted the use of dcHiC for time course analysis of single-cell data, the same can be applied to any pre-defined set of clusters of cells either with respect to their functional annotations (cell type or subset) or sample conditions”

From RESULTS

Differential compartment analysis of pseudo-bulk single cell Hi-C data from post-natal mouse brain development

The post-natal dynamics of mammalian brain development is still a fundamental question in developmental biology [27]. Although, gene expression dynamics has been studied in developing adult and embryonic brains [67-70], the dynamics of 3D genome organization in conjunction with transcriptional changes remain largely uncharacterized. Tan et. al. 2021 [27] attempted to address the issue by integrating single-cell gene expression and single-cell Hi-C data from two mouse brain regions (Cortex and Hippocampus). They employed diploid chromatin conformation capture (Dip-C) method and generated over 3k single-cell Hi-C maps from cortex and hippocampus encompassing 6 different time points that comprehensively describe the dynamic 3D genome organization. This high-quality, time-course and single-cell resolution Hi-C data provided us with an opportunity to showcase the expansion of *dcHiC*'s utility in performing differential analysis on the pseudo-bulk single-cell Hi-C data. Tan et. al. 2021 comprehensively described neuronal sub-types from the single-cell 3D genome data and most of their analysis focused on comparisons among the sub-types. Here we performed differential compartmental analysis among the time points to demonstrate the utility of dcHiC in identifying dynamic changes in compartmentalization for both brain regions. To perform the differential analysis, we first categorized the 6 time points from each brain region into three groups, namely – Early (includes Day 1, 7), Mid (Day 28, 56) and Late (Day 309, 347) as shown in the figure **Figure 8A**. The two time points in each group were treated as replicates and pseudo-bulk Hi-C maps were analyzed at 250kb resolution. We then used dcHiC with default parameters to perform differential compartment analysis and reported compartmental changes below an FDR threshold of 10% (**Figure 8B-C**). Comparing the three groups (Early, Mid and Late), *dcHiC* found a total of ~140Mb of the genome (562 Hi-C bins at 250Kb resolution) in cortex and ~53Mb of the genome (212 Hi-C bins) in hippocampus to be differential in their compartmentalization (**Figure 8B-C**). The higher number of differential compartments in cortex may be reflective of its sudden change in compositional structure in the “Mid” group (i.e., higher fraction of oligodendrocytes) previously identified by Tan et. al. 2021. Intersecting the differential compartments between the tissues revealed 89 Hi-C bins that are common as well as 473 Hi-C bins uniquely differential in cortex and 31 Hi-C bins uniquely differential in hippocampal tissue. The 562/473 bins in cortex overlapped with 2,973/2,566 genes while 212/31 bins in hippocampus encompassed 873/466 genes. A subset of the top cell-type specific marker

and variable genes ranked by PC1 loading identified in the previous study (Tan et. al., 2021) such as *Tshz2*, *Vip*, *Sst*, *Sox11*, *Tubb2b*, *Nrep*, and *Syt1* were also found to be part of *dcHiC*-identified dynamic changes (**Supplementary Figure S12A-F**). We also observed other important genes in differential compartments that are either specific to one region or are common (**Supplementary Figure S12G-L**). For example, we identified *Grin2a* and *Nrg3* (**Supplementary Figure S12**) as part of differential compartments in both brain regions. *Grin2a* provides instructions for making a protein called GluN2A, which is a component of NMDA receptors. This protein is found in nerve cells of the brain region involved in speech and language processing [71]. Neuregulin 3 (encoded by *Nrg3*) is structurally related to neuregulin 1 (NRG1) [71], which plays a critical role in controlling the growth and differentiation of glial, epithelial and muscle cells [72]. The expression of *Nrg3* is known to be highly restricted within developing and adult nervous system. Tissue-specific differential compartments encompassing important genes included the region containing *Chrm5*, which encodes a muscarinic cholinergic receptor that binds acetylcholine and was found within a differential compartment in hippocampus but not in cortex (**Supplementary Figure S12**). On the other hand, *Clstn2*, which is predicted to bind calcium ion and help positive regulation of synapse assembly and synaptic transmission was observed in a differential compartment that is specific to cortex (**Supplementary Figure S12**).

The time-course analysis of three time points (early, mid and late) also helped us identify significantly differential compartments falling into similar dynamic patterns (e.g., descending or ascending) for each brain region. The **Figure 8D-E** shows the descending (cluster 1) and ascending (cluster 2) differential compartments across early, mid and late group in cortex. The genes overlapping with cluster 1 (descending pattern) of **Figure 8D** showed enrichment in development and morphogenesis related GO terms, while differential compartments of cluster 2 (ascending pattern) in **Figure 8E** were enriched in terms like membrane potential and synaptic signaling related biological functions. We also observed two other cluster patterns for cortex that are worth mentioning (**Supplementary Figure S13A-B**). The first one corresponded to a peak in 'Mid' stage (cluster 3, **Supplementary Figure S13A**) and the second one corresponded to a dip (cluster 4, **Supplementary Figure S13B**). Cluster 3 genes showed an enrichment of non-specific functional terminologies while genes belonging to cluster 4 showed specific enrichment of terms related to nervous system (**Supplementary Figure S13C-D**). Tan et.al. 2021 previously described a sudden change in compositional structure types where they observed a higher fraction of glial cells in the 'Mid' stage (Day 28, 56) within cortex. Interestingly, we found *CD33*, a gene that is known to be expressed in microglia [73], as part of cluster 3 (peak in 'Mid'; **Supplementary Figure S14A**). The region containing *Trex1*, a gene with enriched expression in glial cells in human brain [74] also belonged to cluster 3 (**Supplementary Figure S14A**). Two example genes overlapping cluster 4 regions were *Gabra5* and *Anks1b*, both of which were specifically enriched for high expression in excitatory and inhibitory neurons [74] (**Supplementary Figure S14B-D**). **Figure 8F-G** shows the significantly differential compartments with descending or ascending patterns in hippocampus. Unlike cluster 1 of cortex, the genes within

differential compartments following a descending pattern in hippocampus are marginally enriched in Wnt signaling, cell surface and transmembrane receptor signaling pathways. The genes overlapping with cluster 2 (**Figure 8G**) in hippocampus, like those in cortex, are also enriched for biological functions such as membrane potentials and synaptic signaling terms.

Example genes from differential compartments among different stages of post-natal mouse brain development:

Figure 8H-I shows a pair of differential compartments from each region that follows the descending and ascending pattern of score transition among three time points and are overlapping with interesting genes. We observed *Sox11* (**Figure 8H**, left panel), a gene identified by Tan et. al. 2021 [27] as one of the top variable genes, as part of the cluster 1 (**Figure 8D**) in cortex. In the early stages of cortex, *Sox11* resides within an active compartment but with more differentiation the region overlapping with the gene undergoes a gradual A→B transition. The right panel of **Figure 8H** shows another gene *Csmd1* from cluster 2 in cortex. CUB and SUSHI multiple domains 1 (*Csmd1*) is known to be expressed in developing neurons [75] and plays critical role in learning and memory formation [76]. We found this gene as part of a specific differential compartment in cortex that gradually changes from B in early to A in late stages of development. The left panel in **Figure 8I** shows the differential compartment encompassing the *Cntnap4* gene in hippocampus following cluster 1's pattern shown in **Figure 8F**. *Cntnap4* is part of the common set of regions that are differential in both brain regions. The right panel in **Figure 8I** shows a hippocampus-specific differential compartment that overlaps *Dennd1a*. DENN domain containing 1A or *Dennd1a* is a protein coding gene known to be involved in vesicle-mediated transport pathways [77] and Rab regulation of trafficking [78]. Although highly expressed in neuronal as well as glial cells of the brain, the specific role of this gene in hippocampus remains to be investigated.

Overall, these results showed that *dcHiC* addresses a need in the differential analysis of single-cell Hi-C data by first utilizing pseudo-bulk profiles from a low number of cells (80 to 251) to characterize compartmentalization of each condition. *dcHiC* then systematically compares multiple conditions such as timepoints or clusters with the same approach we use for the bulk cell Hi-C data. Our analysis here from post-natal developing mouse brain offers an example scenario where the comparison of multiple conditions with replicates (three developmental stages) was essential to identify important known and novel genes and to characterize dynamic patterns shared across different regions.

Figure 8: Differential compartment analysis of pseudo-bulk single cell Hi-C data from post-natal mouse brain development. (A) Single-cell Hi-C summary and our categorization of 6 time points from each mouse brain region into three groups, namely – Early (includes Day 1, 7), Mid (Day 28, 56) and Late (Day 309, 347). (B-C) The transition of differential compartments in Early, Mid and Late groups within cortex and hippocampal region. (D-E) Time-series clustering of differential compartments in cortex and the functional enrichment of genes overlapping with these compartments. (F-G) Time-series clustering of differential compartments in hippocampus and the functional enrichment of genes overlapping with the compartments. (H-I) Example genes overlapping differential compartments from cortex and hippocampus.

Minor comments:

(1) Under the “Computation and quantile normalization of compartment scores for comparison” sub-section of the Methods section, the authors said input to SVD is “distance-normalized” matrices: how were the matrices distance-normalized?

We have now added text to clarify this. It is the commonly used method of dividing the observed contact count for each locus pair (pixel or entry) by the average contact count of all pairs (including pairs with zero count) with the same genomic distance across that chromosome.

“observed/expected contact count for each genomic distance bin”

(2) What is the computational costs of dcHiC? How does it scale with respect to input data depth, number of cell types compared etc?

We have now done an extensive analysis on this for dcHiC and in comparison, to other methods and incorporated the results in Figure 2 and in Supplementary Tables S1-S4. We added a new results section that outline our findings:

“*Performance evaluation of compartment calling by dcHiC and other approaches*”

Next, we assessed the resource utilization of *dcHiC* against HOMER and CscoreTool for compartment calling, a prerequisite to differential compartment analysis as well as the major bottleneck for high-resolution analysis in general. We evaluated the time and memory utilization of these three methods using two mouse ESC pseudo-replicates (~500M reads each), from which we generated contact maps at 5 different resolutions and 6 different sequencing depths (30 combinations; **Supplemental Information, Table S1-4**). In **Figure 2I-J**, we plotted genome-wide runtimes at 100% sampling rate for 5 different resolutions and for 50kb resolution at 6 different down-sampling rates showing that *dcHiC* runs 4-13x faster than CscoreTool and 22-33x faster than HOMER across these conditions. Across all read depths and all resolutions we tested, *dcHiC* ran 1.3-15x faster than CscoreTool and 10-52x faster than HOMER genome-wide (**Supplementary Tables S1-2**). **Figure 2J** also demonstrated that *dcHiC* scales better with increasing sequencing depth. With respect to memory use, at full read depth and 100kb resolution, CscoreTool had a lower peak memory (~0.24Gb) usage than *dcHiC* (~0.34Gb) and HOMER (~1.2Gb). For resolutions of 50Kb, 40Kb and 25Kb Hi-C data at 100% sequencing depth, all the three tools were within 30% of each other (~1.13Gb, ~1.25Gb and ~1.3Gb for CscoreTool, *dcHiC* and HOMER, respectively) with CscoreTool utilizing the least amount memory for computing the compartment score at every resolution (**Supplementary Tables S3-4**). For these time and memory profiling, we ran all tests genome-wide, and used one CPU per chromosome (Intel Xeon Gold 6252 CPU @ 2.10GHz). Running HOMER genome-wide at 10Kb resolution did not finish after 100 hours of compute time for ESC data.”

(3) How does the method perform when the data depth differs substantially? Conceptually, quantile normalization handles the issue but can the authors show some results with differential depths across the tested cell types?

We have now extensively characterized the robustness of our method with respect to read depth and resolution. We performed a series of new analyses using a range of resolutions and

sequencing depths, which led to the addition of a new section and Supplementary Figures S4 and S5 as well as Supplementary Tables S5-S9. These results can be summarized as:

- We observe that the compartment calls (before differential analysis) are highly concordant within 100% to 40% (500M-200M reads) down-sampling rate ($R > 0.9$; **Supplementary Table S6**).
- We see that *dcHiC* is robust to type-1 error when comparing replicates at different resolutions and read depths (**Supplementary Table S5**).
- We see no false positive differential compartment calls among samples with 100% to 60% down-sampling (500M-300M reads), whereas false positive calls appear when 20% or 10% down-sampled replicates are introduced providing us guidelines for proper use of *dcHiC* (**Supplementary Table S8**).
- We found that there is a high recall of differential compartments ($>80\%$) called from full sequencing depth (“ground truth”) for down-sampling rates of 40% or more (**Supplementary Figure S4**) with similar trends for other resolutions including 50kb, 40kb and 25kb (**Supplementary Figure S5**).
- These analyses allowed us to provide guidelines about the sequencing depth and other requirements for proper utilization of *dcHiC*.
- Overall, *dcHiC* results are quite robust for most relevant settings.

“Robustness of *dcHiC* differential compartment calls

Next, we sought to see how well the pairwise differential compartment calls between different Hi-C profiles are preserved through down-sampling and at different resolutions. We used the 4 ESC biological replicates (230M-1.2B reads) and the 3 NPC biological replicates (720M-1.5B reads) at their full sequencing depth (100%) and then down-sampled each replicate separately to 5 different read depths: 80%, 60%, 40%, 20%, and 10%. To profile the effects of down-sampling, we first compared ESC and NPC replicates at each read depth using 100Kb resolution contact maps. We found that there is a high recall of differential compartments ($>80\%$) called from full sequencing depth (“ground truth”) for down-sampling rates of 40% or more (**Supplementary Figure S4**). In order to also assess the role of resolution in recall of differential compartments, we repeated the same down-sampling experiments for 4 other resolutions: 50Kb, 40Kb, 25Kb, and 10Kb. We observed that except from 10Kb resolution, all other cases were similar to 100Kb where 40% down-sampling still led to a high recall ($>75\%$), whereas for 10Kb resolution, the results at 60% down-sampling had a recall of 80% that dropped to 61% for 40% down-sampling (**Supplementary Figure S5A-D**). Given that the sequencing depth for each replicate varied between 200M to 1.5B reads, we believe that with replicates of at least 80-100M reads, differential compartment analysis at 25Kb or lower resolution can be carried out with high/acceptable recall of all compartmentalization changes that can be detected with deeper sequencing. Later in this section we analyze the role of sequencing depth and resolution on precision of differential compartment detection. ”

Supplementary Figure S4: Effect of sequencing depth on pairwise differential compartment calls (mouse ESC vs NPC) by *dcHiC*. The panel shows the percentages of differential compartments detected at 100% sequencing depth that are recovered after down-sampling the biological replicates (4 for ESC and 3 for NPC) at different levels (80% to 10%) for 100kb resolution analysis.

Supplementary Table S5: Effect of resolution and sequencing depth on type-1 error rate. The table shows differential compartments identified by comparing two pseudo-replicates against each other using *dcHiC* on mouse ESC. The comparison is repeated at different sequencing depths (rows) and resolutions (columns).

Down sampling rates (100% = 500 million)	ESC (Pseudo-replicate 1 vs 2)				
	100Kb	50Kb	40Kb	25Kb	10Kb
100%	0	0	0	0	0
80%	0	0	0	0	0
60%	0	0	0	0	0
40%	0	0	0	0	0
20%	0	0	0	0	0
10%	0	0	0	2	6

Supplementary Table S8: Effect of differential down-sampling (different down-sampling rates for each replicate) on the differential compartment detection by *dcHiC* between two mouse ESC pseudo-replicates.

Differential compartments	Down sampling rates (100% = 500 million)	ESC Pseudo-replicate 2					
		100%	80%	60%	40%	20%	10%
ESC Pseudo-replicate 1	100%	0	0	0	0	247	827
	80%	0	0	0	84	353	907
	60%	0	0	0	124	348	1215
	40%	0	84	124	0	335	1027
	20%	247	353	348	335	0	737
	10%	827	907	1215	1027	737	0

Reviewer #3 (Remarks to the Author):

In this study, Chakraborty and colleagues introduce a new computational method to determine differential chromatin compartmentalization across 2 or more samples. The method is briefly introduced in the main manuscript, and described in more detail in the Methods section, and then ample space is dedicated to demonstrate the power of this approach in distinct biological contexts comparing Hi-C datasets ranging from 2 to 20 in number. In addition, the authors implement and perform multiple downstream analyses to gain information from the differential compartment regions detected by *dcHiC*. Overall, this is an interesting approach addressing an under-appreciated and under-studied problem (it's amazing how much has been done to detect and compare TADs and how little to do so for chromatin compartments).

We thank the reviewer for their constructive evaluation of our work and agree about the importance of developing tools and benchmarks focused on the analysis of chromatin compartments. Below we would like to highlight the most significant changes in this revision for your reference:

1. As suggested by all reviewers, we have now extensively studied the impact of sequencing depth and resolution on the compartment calls and differential compartment calls from *dcHiC* including a false positive rate analysis using replicates and recovery analysis using downsampling. These analyses demonstrated that our results are quite robust for most relevant settings and allowed us to provide guidelines about the sequencing depth and other requirements for proper utilization of *dcHiC*.
2. We now also applied *dcHiC* on single-cell Hi-C data demonstrating its utility in comparing different clusters and/or time points to study dynamic changes in compartmentalization from pseudo-bulk Hi-C profiles from as low as 80 single cells. Our analysis of the Tan et al. (Cell, 2021) single-cell Hi-C data from two regions of post-natal developing mouse brain (each with 6 time points) allowed us to reveal important genes related to synapse assembly and adult brain development that have dynamic and tissue-specific compartmentalization changes, which were not highlighted by the original publication.
3. As requested by multiple reviewers, we have now substantially expanded our comparative analysis to HOMER, Gorkin et al and added Cscore into these comparisons. We have profiled runtimes and memory utilization of each approach for compartment detection in multiple settings and resolution, highlighting the clear advantage of *dcHiC* for high-resolution/finer-scale compartment analysis.
4. We also extensively compared the overlap of differential compartments identified by different approaches with changes/variation in orthogonal measurements from matched samples including gene expression, histone modifications and lamin B1 signal. These results confirmed and strengthened our previous findings that *dcHiC* differential compartments are consistently more enriched in variation of other biological signals.
5. To address confusions from multiple reviewers, we have added a detailed discussion of HOMER and *dcHiC* in terms of their specific implementation of compartment calling and the source of *dcHiC*'s performance improvement including the use of more efficient data structure.

6. We have expanded our documentation in Wiki page of our Github repository (<https://ay-lab.github.io/dcHiC/>) to include single-cell analysis of Tan et. al and provided two demos with accompanying test data to show how *dcHiC* can be used for bulk and single-cell Hi-C data analysis: <https://github.com/ay-lab/dcHiC/demo>

7. We have also added additional analyses showing: i) the overlap of *dcHiC* results with subcompartment calls and their differences, ii) consistency of *dcHiC* calls across different resolutions including high-resolution such as 10kb and 20kb maps when possible, iii) the dynamic changes through time series analysis of specific lineages in the mouse hematopoietic lineage.

Given the main contribution of this work is the development of a tool, I would have appreciated more insight and testing on the method itself, while the detailed description of the results from the various comparisons could sometime be reduced, especially when simply confirming previous findings.

We appreciate the comment and completely agree on the need for cutting down detailed descriptions. However, as you will see, we had to actually extensively expand the results section in order to address the detailed comments from each reviewer. We believe it may be critical for the reviewers to have access to all this information at this stage for making their final decision. However, once/if accepted, we will move the detailed results descriptions into the supplementary information in accordance with editorial office's guidelines and requests.

Below I highlight more detailed suggestions:

1) A major concern of using PC values as compartment scores is data resolution (total number of Hi-C contacts). The authors employ a quantile normalization to make these values comparable across different experiments (which is great), but they should still test to what extent differential compartment regions can be detected by simply changing the resolution of the same experiment. To this extent, the authors should test the robustness of their method to data resolution in different ways such as:

We thank the reviewer for this suggestion. In our previous submission, we already had a comparison between 100kb and 10kb resolution calls for the ESC-NPC comparison. We have now performed a series of new analyses using a range of resolutions and sequencing depths, which led do the addition of a new section and Supplementary Figures S4 and S5 as well as Supplementary Tables S5-S9.

These results can be summarized as:

- We observe that the compartment calls (before differential analysis) are highly concordant within 100% to 40% (500M-200M reads) down-sampling rate ($R > 0.9$; **Supplementary Table S6**).
- We see that *dcHiC* is robust to type-1 error when comparing replicates at different resolutions and read depths (**Supplementary Table S5**).

- We see no false positive differential compartment calls among samples with 100% to 60% down-sampling (500M-300M reads), whereas false positive calls appear when 20% or 10% down-sampled replicates are introduced providing us guidelines for proper use of *dcHiC* (**Supplementary Table S8**).
- We found that there is a high recall of differential compartments (>80%) called from full sequencing depth (“ground truth”) for down-sampling rates of 40% or more (**Supplementary Figure S4**) with similar trends for other resolutions including 50kb, 40kb and 25kb (**Supplementary Figure S5**).
- These analyses allowed us to provide guidelines about the sequencing depth and other requirements for proper utilization of *dcHiC*.
- Overall, *dcHiC* results are quite robust for most relevant settings.

“Robustness of *dcHiC* differential compartment calls

Next, we sought to see how well the pairwise differential compartment calls between different Hi-C profiles are preserved through down-sampling and at different resolutions. We used the 4 ESC biological replicates (230M-1.2B reads) and the 3 NPC biological replicates (720M-1.5B reads) at their full sequencing depth (100%) and then down-sampled each replicate separately to 5 different read depths: 80%, 60%, 40%, 20%, and 10%. To profile the effects of down-sampling, we first compared ESC and NPC replicates at each read depth using 100Kb resolution contact maps. We found that there is a high recall of differential compartments (>80%) called from full sequencing depth (“ground truth”) for down-sampling rates of 40% or more (**Supplementary Figure S4**). In order to also assess the role of resolution in recall of differential compartments, we repeated the same down-sampling experiments for 4 other resolutions: 50Kb, 40Kb, 25Kb, and 10Kb. We observed that except from 10Kb resolution, all other cases were similar to 100Kb where 40% down-sampling still led to a high recall (>75%), whereas for 10Kb resolution, the results at 60% down-sampling had a recall of 80% that dropped to 61% for 40% down-sampling (**Supplementary Figure S5A-D**). Given that the sequencing depth for each replicate varied between 200M to 1.5B reads, we believe that with replicates of at least 80-100M reads, differential compartment analysis at 25Kb or lower resolution can be carried out with high/acceptable recall of all compartmentalization changes that can be detected with deeper sequencing. Later in this section we analyze the role of sequencing depth and resolution on precision of differential compartment detection.

...

To further assess the type-1 error rate, we carried out a series of differential compartment analysis between mouse ESC pseudo-replicates (2 replicates) at different resolutions (100Kb, 50Kb, 40Kb, 25Kb and 10Kb) and down-sampling rates (100%, 80%, 60%, 40%, 20% and 10% of 500 million sequencing depth). We measured the number of differential compartments when running two down-sampled replicates against each other at different resolutions and our results indicate that *dcHiC* is robust to type-1 error when comparing

replicates at different resolutions and read depths (**Supplementary Table S5**). We also evaluated the type-1 error rate, when two mouse ESC pseudo-replicate Hi-C maps of different sequencing depth are compared by *dcHiC*. Across the 21 comparisons, we first see that the compartment calls are highly correlated within 100% to 40% (500M-200M reads) of read depth (**Supplementary Table S6**). The correlations with high read depth samples drop substantially for 20% (100M reads) and further for 10% (50M reads) sample. We noticed this occurred because compartment scores for some chromosomes started to not fully reflect the compartmentalization pattern at lower read depths. Removing the 5 chromosomes (chr 4, 5, 14, 17, X) with such issues, we see correlations at lower read depths improve, however not to the point that we highly concordant (correlation >0.9) compartment calls between two pseudo-replicates (**Supplementary Table S7**). While evaluating the false-positive calls, we first observed that correlations between compartment scores are closely related to the number of differential calls. When we utilized *dcHiC* to find differential compartments (i.e., false positive calls) between two replicates of different sequencing depth by down-sampling Hi-C maps at 100Kb resolution, we see no false positives up among samples with 100% to 60% down-sampling (500M-300M reads) (**Supplementary Table S8**). We also do not obtain any false positives even for lower depth samples when they are compared against the sample with the same rate of down-sampling. However, a substantial number of false positive differential calls appear when 20% or 10% down-sampled samples are compared to higher depth samples (**Supplementary Table S8**). Like compartmental correlations, here also when we filter out the 5 chromosomes with issues in compartment calls at low read depths, we see that the false positive rates dramatically improve for 40% and for 20% down-sampled samples (**Supplementary Table S9**). Based on these results, we believe compartment scores and differential compartment calls are robust when comparing Hi-C maps that are sufficiently sequenced (100M or more reads) and are within 2-3-fold read depth of each other. “

Supplementary Table S5: Effect of resolution and sequencing depth on type-1 error rate. The table shows differential compartments identified by comparing two pseudo-replicates against each other using *dcHiC* on mouse ESC. The comparison is repeated at different sequencing depths (rows) and resolutions (columns).

Down sampling rates (100% = 500 million)	ESC (Pseudo-replicate 1 vs 2)				
	100Kb	50Kb	40Kb	25Kb	10Kb
100%	0	0	0	0	0
80%	0	0	0	0	0
60%	0	0	0	0	0
40%	0	0	0	0	0
20%	0	0	0	0	0
10%	0	0	0	2	6

2A) In the first part of their algorithm, the authors present a new/fast approach to detect compartments (A and B) and compare their strategy to the more standard approach based on PCA as implemented in HOMER. Here I have a couple of questions:

- if I understand correctly the main difference between the two is that *dcHiC* employs SVD, while HOMER standard PCA, is that the only difference? Can the author provide a few more details on the two approaches?

(Also, how does HOMER perform differential compartment analysis? Why does it return such a larger number of hits?

We would like to address this concern, one shared by multiple reviewers, by clarifying *dcHiC*'s precise compartment calling method. In R, the standard way to compute eigenvalues is the `eigen()` or `svd()` function. However, when the matrix becomes large, these functions can be very time-consuming: the complexity to calculate all eigenvalues of an 'n' by 'n' matrix is $O(n^3)$. While in real applications including compartment analysis, we usually only need to compute a few eigenvectors, for example to visualize high dimensional data using PCA. The same thing happens in Singular Value Decomposition (SVD). It is often the case that only a Partial SVD or Truncated SVD is needed, and moreover the matrix is usually stored in sparse format. In the background, both our method and other popular programs like HOMER uses the eigendecomposition of the correlation matrices to find the principal components. HOMER in particular uses the default 'eigen' function of R to perform the PCA. Due to its inefficient handling of large matrices, however, the principal component calculation of Hi-C maps at higher resolution becomes impractical to run with eigen function. *dcHiC* implements the 'bigstatsr' package in R and uses Filebacked Big Matrices (FBM) via memory-mapping. This allows, for instance, matrix operations that includes memory and time-efficient low-rank approximation of standard PCA calculation based on the algorithm in C++ spectra library. Given an 'n' by 'n' large matrix A, the 'big_randomSVD' from 'bigstatr' package can calculate a specified number of eigenvectors of A. Users can specify the selection criterion by an argument which allows computing only the k largest eigenvalues and their corresponding eigenvectors.

For HOMER, after calling the principal compartments on the Hi-C maps using the 'runPCA.pl' function, the pairwise differential compartment analysis involves two default steps - the first step is to annotate the compartment bedGraph file using 'annotatePeaks.pl' function followed by quantifying the differential features using 'getDiffExpression.pl' function as described under the 'Quick reference for PCA analysis' page. By default, annotatePeaks.pl uses the genomics positions to determine the closest transcription start sites (TSS) of that genome. To annotate the location of a given peak (or compartment score) in terms of important genomic features, annotatePeaks.pl calls a separate program (assignGenomeAnnotation) to efficiently assign peaks to one of millions of possible annotations genome wide. Once the assignment is complete, the analysis of differential regulation is handled by the 'getDiffExpression.pl' function. The getDiffExpression.pl program is essentially a wrapper for R/Bioconductor/limma/EdgeR/DESeq2 to make running those programs easy using data generated by other HOMER programs. For

differential principal component analysis HOMER suggests to use 'limma' package. In the background, HOMER first implements 'lmFit' i.e. fits multiple linear models on the design matrix by generalized least squares. The coefficients of the fitted models describe the differences between testing conditions. Given a linear model fit from 'lmFit' function, HOMER then implements 'eBayes' function from the same package to compute the moderated t-statistics and to rank annotated regions in order of evidence for differentiability. By default, limma uses moderated t-statistics to estimate significance in microarray experiments. The moderated t-statistic (t) is the ratio of the M-value (Difference in log₂ signal intensity between two channels) to its standard error. This has the same interpretation as an ordinary t-statistic except that the standard errors have been moderated across genes, effectively borrowing information from the ensemble of genes to aid with inference about each individual gene. Limma uses this Empirical Bayes method to moderate the sample variances, which are mean squared deviations across the same region among the replicates. In summary, the moderated t-test is a t-test using the square root of the moderated variance as the standard deviation instead of the sample variance which allows limma to detect a more differential genes compared to others available methods. We believe the approach that has shown to work well for microarray probes leads to a higher number of differential compartment calls of Hi-C maps especially at lower-resolution like 100Kb resolution.

One other popular method for compartment calling is CscoreTool, an approach that completely avoids the canonical PCA and instead uses a sliding genomic window that predicts whether a given region is A or B as a log-likelihood function of the number of interaction counts in the region. As such, *dcHiC* and HOMER are based on the same method (with different implementations), while CscoreTool is a different approach for compartment detection. We have now done the comparison and incorporated the results in Figure 2. In the revised Figure 2, we show correlation plots of all three methods against each other and Lamin B1 data. As expected, CscoreTool performs comparably with *dcHiC* and HOMER in compartment detection

2B) while the correlations shown in Figure 1A and 1D are strong, since compartments are called based on the sign of the PC, an additional (more proper) comparison would have been to show the fraction of bins that have different A/B assignments with the two approaches.

Indeed in both comparisons in Figure 1A and 1D it appears that there is a subset of bins in the top-left quadrant that would be called A by *dcHiC* and B by HOMER. Of course I expect differences between the tools, 100% consistency would be unrealistic, but it is curious that while some A compartment regions in *dcHiC* are called B by HOMER (top-left quadrant), the vice versa almost never occur (bottom-right quadrant) suggesting a systematic shift of scores. Can the authors quantify the fraction of bins where discordant calls occur? Do they have an intuition on why discordant calls are almost exclusively in one direction?

We thank the review for raising this point. Figure 3F in the previous (and new) version of the manuscript showed the overlap between *dcHiC* and HOMER pairwise differential calls. We have now expanded characterization of the differences between two methods by performing an **analysis of inconsistent compartment calls (A vs B) between *dcHiC* and HOMER.**

“Next, we further analyzed the 4-7% of the genome that is labeled in opposite compartments by *dcHiC* in comparison to HOMER for ESC and NPC (**Supplementary Figure S1A-B**). Overall, *dcHiC-B* but HOMER-A regions (~1% for ESC and NPC) showed positive lamin B1 signal and lower gene expression levels compared to *dcHiC-A* but HOMER-B regions (**Supplementary Figure S1C-D**). The latter set (3% for ESC and 6% for NPC) had a mix of regions with positive and negative lamin B1 association as well as gene expression values that are lower than constitutive A but higher than constitutive B compartment regions (compare to **Figure 3**) suggesting a weak compartmentalization for these regions into either A or B compartment.”

Supplementary Figure S1: Inconsistent A/B compartment assignments between *dcHiC* and HOMER. **(A-B)** Shows the number (top) and fraction (bottom) of consistent and inconsistent A/B assignments by *dcHiC* and HOMER in mouse ESC and NPC Hi-C maps. **(C)** Shows the Lamin B1 signal of inconsistently labelled regions with *dcHiC-B* but HOMER-A regions showing lamin B1 association supporting their

heterochromatin (or B) assignment. **(D)** Shows the expression distribution of genes overlapping with the inconsistently labelled regions by *dcHiC* and HOMER.

3) Along the same lines of the previous comment, the authors did not compare their compartment calls with those of other approaches, especially when these allow to call subcompartments (see PMID: 25497547, PMID: 31699985, PMID: 33972523). Besides the comparisons of A and B compartment calls made by these approaches, it would be interesting to combine subcompartments inferred by them with *dcHiC* results to have a more granular analysis of significant compartment differences: do they at least involve subcompartment flips? are regions in different subcompartment equally likely to change or certain subcompartments are more "flexible"? (E.g. in PMID: 33972523 the authors talk about subcompartments enriched for differentiation genes, the regions of frequently change compartments across cell lines)

We have now done an extensive comparison of *dcHiC* results and subcompartment flips using Calder subcompartment calls. This led to addition of a new section and a supplementary figure

(Figure S7). Our conclusion was that although subcompartment analysis is needed to capture the multistate genomic activity across cell lines, and algorithms such as Calder provide a useful approach to decipher the underlying epigenetic and transcriptional heterogeneity they are likely not suitable for the task of *de novo* detection of compartmentalization changes (i.e., as any change that involves a subcompartment label flip) across samples due to a large number of transitions (70-80%) involving sub-compartment types that are very similar (distance of 1 or 2), at least in the case of ESC vs NPC.

“Differential compartments are associated with sub-compartment transitions during ESC to NPC lineage differentiation

Recent studies have shown that beyond open and closed chromatin, genome activity encompasses multiple states of compartmentalization which can be captured via a more refined sub-compartment analysis [14, 15, 47]. Therefore, we hypothesized that differential compartments identified by dcHiC, whether they involve compartment flips or not, should also be associated with changes in sub-compartments between conditions. To compare the changes in sub-compartments with differential compartments, we mapped the *dcHiC* differential calls on the ‘Calder’ [47] derived sub-compartments within mouse ESC and NPC cell lines. The Calder algorithm infers a complete hierarchy of compartment domains using intrachromosomal interactions and classifies each A/B compartment into 4 sub-compartments each (8 in total; A/B.1.1, A/B.1.2, A/B.2.1, A/B.2.2) adopting a more nuanced representation of the two primary compartment classes. We applied Calder on ESC and NPC Hi-C maps separately and retrieved a total of 7,967 100Kb bins (~800Mb) with sub-compartment assignments for both ESC and NPC. For these bins, we then assessed the overlap of differential calls from dcHiC with the differences in sub-compartment labels. Out of 1,981 dcHiC bins, for 1,820 we had Calder labels on both cell types and among those 97.6% (1777 bins) overlapped with differential sub-compartment labels. For the remaining 6,147 bins with Calder labels that do not overlap with dcHiC calls, still a high but smaller percentage (74.8%) corresponded to differences in sub-compartment labels. **Supplementary Figure S7A-B** shows the total number of differential compartment transitions, grouped based on their sub-compartment classes within ESC and NPC lineages. These results highlight that nearly all dcHiC differential compartments have underlying changes in sub-compartment assignments consistent with our initial hypothesis. In terms of being able to do a differential analysis directly from sub-compartments, however, the large percentage (~80% or 6,377 out of 7,967 100kb bins) of sub-compartment transitions/flips suggest that this approach may lead to low specificity in detecting important differences and would need to be coupled with additional filters and/or supplemented by further statistical assessments.

To better understand the type of sub-compartment flips that are overrepresented in dcHiC calls, we compared the transition probabilities among sub-compartment labels (ESC vs NPC) obtained from dcHiC differential calls versus non-differential regions (**Supplementary Figure S7C-D**). The fold-change values show that dcHiC differential

calls are significantly enriched for sub-compartment transitions with a distance of 3 or more in the sub-compartment hierarchy (e.g., A.1.1 to A.2.2 (distance of 3) or A.1.1 to B.1.1 (distance of 4)) supporting the strong compartmentalization change of these bins (**Supplementary Figure S7C-D**). We observed highly enriched transitions from ESC-A subcompartments to strong NPC-B subcompartments (B.2.1 and B.2.2) that corresponded to substantial reduction in the transcriptional activity of overlapping genes going from ESC to NPC (**Supplementary Figure S7E**). An example of such sub-compartment transition was the 145-148Mb region in chromosome 4 encompassing 71 unique genes (**Supplementary Figure S7F**). This locus harbored genes with known functions including pluripotency (*Rex2*) and migration and invasion inhibition (*Miip*) [48].

Although the broad classification of A and B compartments is likely insufficient to capture the multistate genomic activity across cell lines, our sub-compartment analysis suggested that differential analysis using compartment scores is able to effectively capture changes involving sub-compartments with biological significance. Sub-compartment inferring algorithms such as Calder [47] provide a useful approach to decipher the underlying epigenetic and transcriptional heterogeneity within tissue types, differentiation stages and other conditions but are not directly applicable for the task of *de novo* detection of compartmentalization changes across samples due to a large number of transitions involving sub-compartment types that are very similar (distance of 1 or 2). ”

Supplementary Figure S7: Differential compartments are associated with sub-compartment transitions during ESC to NPC lineage differentiation. **(A-B)** shows the total number of differential compartment transitions, grouped based on their sub-compartment classes within ESC and NPC lineages. **(C-D)** Shows the background normalized transitions, or the fold-change values obtained from differential sub-compartment frequencies divided by non-differential sub-compartment changes. **(E)** The sub-compartment flipping correspond to changes in genomic activity. The panel shows a significant alteration in the gene expression pattern when ESC-A.2.1 flips to NPC-B.2.2 sub-compartment. **(F)** Shows one of such flips in chromosome 4 (145-148Mb region) encompassing 71 unique genes. The average expression of these genes in ESC is around 20 TPM and ~0.4 TPM in NPC.

4A) To evaluate the results obtained by dcHiC, the authors performed several enrichment analyses (epigenetic features, gene expression, gene ontology etc.) While these are welcome, it would be great to have a better feeling of how frequently a significant compartment change is supported by orthogonal evidence. How many compartment changes are indeed associated with differentially expressed genes (and vice versa)? How many are associated with epigenetic changes (the authors could check if ChIP-seq data for histone modifications is available)?

We acknowledge that the concern raised by the reviewer is valid and we have performed additional analyses in this revision to address this issue. In addition to Supplementary Figure 1 that was discussed above for ESC – NPC comparison, we also added Supplementary Figures S2 and S3 with gene expression and histone modification changes in comparison to dcHiC and HOMER calls as well as Supplementary Figures S10 and S11 for the discussion of variable histone modification regions for the Gorkin et al. data. Both analyses suggest higher enrichment of differences in biological signals for dcHiC compared to other methods.

a. Further analysis of differences in differential compartment calls from dcHiC and HOMER:

“To compare the calls made by the two different methods, we plotted the absolute differences in lamin B1 signal, replication timing and log₂ gene expression values of all the reported differential compartments (**Figure 3G**) or method-specific differential compartments (**Figure 3H**) for each method. These results show that dcHiC differential compartments are significantly (unpaired t-test p-values < 0.05) enriched for regions with higher ESC and NPC differentials for lamin B1 association and replication timing signals although both methods captured regions with signal differences in all three measures. We also performed differential expression analysis between ESCs and NPCs to map the differentially expressed (DE) genes (DESeq2 [41], FDR<0.05, fold change>4) on the differential compartments. We observed that *dcHiC* differential compartment bins were enriched in the number of DE genes (**Figure 3I**) as well as the fold change (log₂) and significance (DESeq2) of the difference for those DE genes (**Supplementary Figure S2A-B**). Further, we also looked at the average number of histone modification peak (MACS2 p-value < 1e-5) differences between ESC and NPC per 100Kb for the regions from dcHiC and HOMER’s differential calls (**Supplementary Figure S3**). For all three different histone marks (H3K4me1, H3K4me3, H3K27ac), we observed a higher number of peak differences per 100Kb for dcHiC compared to HOMER. ”

Supplementary Figure S3: Average difference in absolute number of histone peaks (MACS2 p-value < 1e-5) per 100Kb of differential bins within *dChIC* and HOMER differential compartments. **(A)** Shows the absolute difference of average number H3K27ac peaks between ESC and NPC per 100Kb of *dChIC* and HOMER differential compartments. The result shows that there is more difference in H3K27ac peaks per 100Kb of differential compartments identified by *dChIC*. **(B-C)** The result shows that are more differences in both H3K4me3 and H3K4me1 peaks per 100Kb of differential compartments called by *dChIC* compared to HOMER.

b. Analysis of the overlap between *dChIC* and Gorkin et al differential compartment calls and variable histone modification regions previously identified across different LCLs:

“We then asked whether the identified differential compartment regions were enriched in regions with variability in histone marks (H3K27ac, H3K4me3, H3K4me1 and H3K27me3) across different individuals. The variable histone modification regions/peaks identified for human LCLs by Kasowski et. al. 2013 [65] were mapped on differential compartments identified from *dChIC* and by Gorkin et. al. [66]. Using the non-differential compartment regions as background for each method, we observed nearly no enrichment for regions called differential only by Gorkin et. al. (**Supplementary Figure S10**) while calls from *dChIC* showed 26-45% of enrichment. The proportion of differential calls that overlapped with at least one variable region for each histone mark was also substantially higher for regions from *dChIC* in comparison to Gorkin et. al. specific regions (**Supplementary Figure S11A-D**).”

Supplementary Figure S10:

Enrichment of differential compartments from either dcHiC or Gorkin et. al. paper that overlap at least one variable histone mark region/peak identified by Kasowski et. al (2013). The enrichment is computed with respect to all non-differential compartment regions for each method. We assess the enrichment for all calls from dcHiC in comparison to calls that are specific to Gorkin et. al.

Supplementary Figure S11:

The number of differential compartments from each method that overlap with a variable histone mark region (similar to Figure S10) for (A) H3K27ac, (B) H3K4me3, (C) H3K4me1, and (D) H3K27me3.

4B) Also the authors always report compartment changes in terms of number of bins, but I suspect in many cases multiple bins are contiguous (a compartment change of only 1 bin is more likely to be due to noise). If they account for contiguity how many changes do they get and what is their size distribution?

Possibly using orthogonal evidence and a size threshold could help determine the true differences and further filtering false positives which could emerge for technical reasons such as data resolution.

We thank the reviewer for their suggestion about further studying differential compartment sizes with the possibility of using this information for filtering. In the current setting, dcHiC does not filter out singleton bins of differential compartments or segregate them from longer stretches of differences. Upon this comment, we decided to further study such singleton regions first. In the pairwise comparison between ESC and NPC, dcHiC predicted 1,981 differential compartments at

100Kb resolution. When we merged all the contiguous differential compartments using ‘bedtools merge’, we were left with 272 singleton genomic bins (~14% of all calls). To check if these singleton bins overlap important differences such as in gene expression, we overlapped the ESC-NPC differentially expressed genes (4,958 DEGs, FDR < 0.05 and $|\log_2FC| > 2$) and found that 98 out of 272 differential compartments (~36%) overlapping with DE genes (a total of 138 genes). Upon manual inspection, we observed that a few of these singleton bins overlapped with important and relevant genes like *Ctndd2* and *Pcdh7* with neuronal functions, the pluripotency marker *Dppa5a* as well as several other genes plotted below. We, therefore, made the decision to not filter (or treat differently) the singleton bins from longer stretches of differential compartment calls. Such filtering, especially with large bin sizes such as 100kb, may lead to a significant loss in sensitivity and equally importantly, the regions that are missed by this filter may be enriched with genes that have critical functions within the context of the compared Hi-C maps. We agree, however, that for high-resolution analysis such filtering can be useful to eliminate regions with sporadic changes. Although not incorporated into the tool, we provided a script that can perform such filtering in our GitHub repository under: <https://github.com/ay-lab/dcHiC/tree/master/utility>

A few minor comments:

5) How much do the results of dcHiC depend on the weighting scheme adopted (Eq. 6 in the Methods)? It would be important to understand the contribution of this parameter to the results. How would the result change without weighting? What if a different weighting strategy was adopted (e.g. 75% quantile instead of max Z)?

We carried out the differential compartment analysis between ESC vs NPC samples using different weighing strategy and here is the table showing its effect on number of differential compartment calls -

Weight	Differential Compartment
Prob{max Z}	1,981
0	2,412
0.1	1,841
0.2	1,357
0.3	876
0.4	524
0.5	262
0.6	114
0.7	13
0.8	-

Essentially, selecting max Z-score provides more weight to the points that are distant from others among the samples (further from the diagonal) than to points that are closer together in the multidimensional space (close to the diagonal). Equation (4) in the methods section of the manuscript is the standard MD formulation, which we modify using the weighted centers as computed through Equations (6) to (8). We have also looked at the successive overlap of differential calls with increasing weight and found that the smaller set is always a subset of the larger one.

6) Why wasn't the time-series analysis done also for the HSC lineage differentiation study? That would be a nice addition to understand the number of concordant/progressive changes during lineage differentiation vs. changes that emerge sporadically.

We thank the reviewer for asking this question. We have now performed the time-series analysis of the differential compartments on the Long-Term Hematopoietic stem cells (LT-HSC) to Granulocytes (GR) (6 time-points) and LT-HSC to Megakaryocytes (MK) (7 time-points) lineage differentiation separately. The time-series analysis pointed out some interesting patterns in both the cell differentiation types but the functional enrichment result for time-series clusters were somewhat non-specific. We now discuss these results in a new paragraph and two supplementary figures (S8 and S9).

“Further, we have performed time-series analysis of the differential compartments on the Long-Term Hematopoietic stem cells (LT-HSC) to Granulocytes (GR) (6 time-points) and LT-HSC to Megakaryocytes (MK) (7 time-points) lineage differentiation separately. For LT-HSC to GR differentiation, the first 3 clusters show a general pattern of differential compartments with decreasing genomic activity while the last 3 shows an increase (**Supplementary Figure S8A**). The functional enrichments for genes within each cluster involved general terms such as ‘morphogenesis’, ‘development’ and ‘organization’ (**Supplementary Figure S8B**). When we repeat the same analysis for LT-HSC to MK lineage differentiation, we observed more nuanced patterns involving four clusters with distinct signatures in MEPs (**Supplementary Figure S9A**). For these clusters (Cluster 1, 2, 4, and 5), the change in compartment score is most prominent at the MEP stage and is generally prominent after this stage. We believe this is due to the unique condensed chromosomal organization observed in MEP stage along with MKs [33]. This previous study proposed that in these cell types, there is a reduction in long-range chromatin interactions, which resembles the condensed chromosome structures found in mitotic metaphase cells [33]. We believe the time-series analysis of differential compartments from *dcHiC* thus captured this feature of MEPs while also capturing two clusters (cluster 3 and 6) with gradual increase or decrease in their compartment scores (**Supplementary Figure S9A**). The functional enrichment of genes in each cluster again involve general terms such as ‘morphogenesis’, ‘development’ and ‘differentiation’ (**Supplementary Figure S9B**).”

7) I believe these are typos/oversights, but in the background section the descriptions of PCA and eigenvector decomposition are imprecise:

- at line 65 the sentence seem to indicate that eigenvectors and principal components are the same thing, but they aren't,

- at line 73 the authors write “magnitude and sign of eigenvalues derived from PCA have been the major determinants of compartment type”, this is just wrong, eigenvalues are only used to rank eigenvectors. It is the sign and magnitude of the values of the first (or second) principal component that are used to determine compartment type.

All of this is correctly reported in the Methods so I believe these were simple oversights, but they should be corrected.

We thank the reviewer for pointing these issues. We have now fixed all of these and did another detailed check on grammar and such oversights.

Reviewers' Comments:

Reviewer #1:

Remarks to the Author:

The authors addressed my concerns and have improved the manuscript. I do have one remaining concern in that the sequencing depth does appear to impact the recall rate quite dramatically (i.e. 500 vs 200 million reads makes it drop to 80%). Because 200 million contacts is fairly commonly found in published Hi-C data, this would indicate a loss of 1/5th of the recall in addition to the unmeasurable recall rate at 500 million reads. That's somewhat concerning, especially since the recall curves do not plateau, possibly indicating that 500 million is insufficient to have good recall rates. However, I don't believe this should detract from the importance of the work and would like to acknowledge that this is already a dramatic improvement over current methods of differential compartment analysis which have traditionally lacked robust statistical basis. I greatly appreciate the authors' revisions.

Reviewer #2:

Remarks to the Author:

The authors have made tremendous efforts addressing reviewers' comments. I have no further comments.

Reviewer #3:

Remarks to the Author:

In this revised version of their study, the authors did an impressive job including several new analyses to address my previous concerns as well as those of the other reviewers.

I just have a few minor notes / requests of clarification on their responses:

1) I believe my comment 2B had been misinterpreted. What I meant originally was to investigate the differences in compartment calling between the two methods (dHiC and HOMER), NOT the differences in differential compartment calling.

In the current Figure 2A,B,C the authors compare compartment scores, which are used to call compartments, among each pair of tools (dHiC, HOMER, and Cscore).

These values are correlated but are not centered with respect to the diagonal of the cartesian plane. For example in 2A one can see several bins that have positive scores for dHiC and negative scores for HOMER, but the opposite is not true.

The same is seen and even stronger in Fig. 2B and C. In Fig. 2C many bins have positive scores for dHiC but negative for Cscore, while the opposite almost never happens. Since compartments are typically called based on the sign of these scores, how should one interpret such phenomena?

Do the authors know what drives this systematic difference?

Also, in Fig. 2I, it would be helpful to maintain the colors as in the other panels.

2) In the comparison with calder, I'm not sure I got the numbers right... from what I read in the text, it seems that in total ~8000 bins have been analyzed, each of size 100KB. That corresponds to ~800M base pairs, which is approximately one third of the mouse genome (~2.5B). Why is that?

In addition, we have recently analyzed that dataset with the same tool and got very different results, we obtained ~22,500 bins of size 100kb, out of which ~59% changed sub-compartment label, not 80% as stated in the rebuttal document.. Am I misinterpreting something?

There might be version differences of the tool, not sure, but the differences seem quite big..

REVIEWER COMMENTS

Reviewer #1 (Remarks to the Author):

The authors addressed my concerns and have improved the manuscript. I do have one remaining concern in that the sequencing depth does appear to impact the recall rate quite dramatically (i.e. 500 vs 200 million reads makes it drop to 80%). Because 200 million contacts is fairly commonly found in published Hi-C data, this would indicate a loss of 1/5th of the recall in addition to the unmeasurable recall rate at 500 million reads. That's somewhat concerning, especially since the recall curves do not plateau, possibly indicating that 500 million is insufficient to have good recall rates. However, I don't believe this should detract from the importance of the work and would like to acknowledge that this is already a dramatic improvement over current methods of differential compartment analysis which have traditionally lacked robust statistical basis. I greatly appreciate the authors' revisions.

We thank the reviewer for their constructive comments and positive views of our work. On this specific issue, we would first like to clarify that the downsampling analysis with respect to recall that the reviewer refers to is actually done on the real replicates (4 for ESC and 3 for NPC) rather than pseudo-replicates, which we used for time, memory usage and false positive rate estimation. Therefore, the read depths are not 500M for 100% and 200M for 40% for this analysis. They change between 230M-1.2B reads for ESC 720M-1.5B reads for NPC replicates at 100% depth (4v3 analysis). Therefore, 40% downsampling would put one ESC replicate below 100M reads. And that one replicate is the likely culprit of the decline in recall. To further assess this, we have now repeated our recall analysis for 100kb resolution by removing that one replicate of 230M reads leaving us with reads ranging from 600M to 1.5B reads (3v3 analysis).

First of all, this new 3v3 analysis captured 1906 out of 1981 differential bins from 4v3 analysis suggesting that removing the low depth replicate had minimal effect on recall. Importantly, when we repeated the downsampling analysis to see what fraction of those 1906 differential bins are captured at different sequencing depths, we saw that 40% downsampling (putting the least sequenced replicate at around 240M reads) kept the recall rate at around 90% and 20% downsampling was at around 80% recall (see figure below). These results suggest that replicates with substantially lower sequencing depths may not contribute much to overall discovery power and they may adversely affect recall rate if they are sequenced below 100M reads. We have now revised the text accordingly with some of this above discussion. We sincerely thank the reviewer for a chance to revisit and clarify this important point.

	100	80	60	40	20	10
3 v 3						
100 Kb	1	0.95	0.930303	0.8924242	0.8070707	0.5989899

Reviewer #3 (Remarks to the Author):

In this revised version of their study, the authors did an impressive job including several new analyses to address my previous concerns as well as those of the other reviewers. I just have a few minor notes / requests of clarification on their responses:

1) I believe my comment 2B had been misinterpreted. What I meant originally was to investigate the differences in compartment calling between the two methods (dcHiC and HOMER), NOT the differences in differential compartment calling. In the current Figure 2A,B,C the authors compare compartment scores, which are used to call compartments, among each pair of tools (dcHiC, HOMER, and Cscore). These values are correlated but are not centered with respect to the diagonal of the cartesian plane. For example in 2A one can see several bins that have positive scores for dcHiC and negative scores for HOMER, but the opposite is not true. The same is seen and even stronger in Fig. 2B and C. In Fig. 2C many bins have positive scores for dcHiC but negative for Cscore, while the opposite almost never happens. Since compartments are typically called based on the sign of these scores, how should one interpret such phenomena? Do the authors know what drives this systematic difference?

We thank the reviewer for clarification of their previous comment. Although we did not address it in our response to this specific comment 2B, we have done a detailed analysis of compartment call differences between HOMER and dcHiC in our previous submission which is mentioned in the text below and shown in detail in Supp. Fig 1 (previous and current).

“Next, we further analyzed the 4-7% of the genome that is labeled in opposite compartments by dcHiC in comparison to HOMER for ESC and NPC (Supplementary Figure S1A-B). Overall, dcHiC-B but HOMER-A regions (~1% for ESC and NPC) showed positive lamin B1 signal and lower gene expression levels compared to dcHiC-A but HOMER-B regions (Supplementary Figure S1C-D). The latter set (3% for ESC and 6% for NPC) had a mix of regions with positive and negative lamin association as well as gene expression values that are lower than constitutive A but higher than constitutive B compartment regions (compare to Figure 3) suggesting a weak compartmentalization for these regions into either A or B compartment.”

In short, we found that regions with method-specific compartment labels (A or B) tend to have weak compartmentalization in general. This is expected since “zero” is rather an arbitrary threshold to determine compartment labels from a continuous score and values close to zero are likely to change sign even with small technical differences in between methods.

Regardless, we have done some further analysis, this time including Cscore to better characterize these discordantly labeled regions. As shown below and in Supp. Fig 1A-B, and as also noticed by the reviewer, the discordant calls are mainly in the direction of dcHiC A compartment being called B by HOMER and by Cscore for both ESC and NPC.

ESC bins		HOMER	
		A	B
dcHiC	A	11131	821
	B	307	13093

NPC bins		HOMER	
		A	B
dcHiC	A	11565	1415
	B	144	12229

ESC bins		Cscore	
		A	B
dcHiC	A	12362	1977
	B	16	9237

NPC bins		Cscore	
		A	B
dcHiC	A	11542	1390
	B	567	11635

To further evaluate the dcHiC, HOMER and Cscore calls, we also mapped the ESC and NPC Lamin B1 signal regions over the compartments. In general, the Lamin B1 detached regions (-ve signal) should correspond to Hi-C A compartments while Lamin B1 attached regions (+ve signal) represents Hi-C B compartments. In the following two tables, we observed that HOMER and Cscore have higher number of B compartment calls with regions detached

from the lamina suggesting they might be overcalling B compartments. On the other hand, dChIC and HOMER compared to cScore for ESCs and dChIC and cScore compared to HOMER for NPCs called higher number of A compartments with lamin attachment suggesting they may be overcalling A compartments. As mentioned above and highlighted also in our down-sampling analysis for compartment correlations, it is difficult to assign a confident compartment label for regions with a compartment score that is close to zero. Therefore, we avoid making conclusive statements about this issue at this point.

Lineage	Method	ESC LaminB1		Lineage	Method	NPC LaminB1	
		Detached (-ve)	Attached (+ve)			Detached (-ve)	Attached (+ve)
ESC.A	dcHiC	10,726	1,193	NPC.A	dcHiC	11,366	1,587
	HOMER	10,351	1,048		HOMER	10,540	1,153
	Cscore	8,857	396		Cscore	10,610	1,499
ESC.B	dcHiC	1,161	12,180	NPC.B	dcHiC	2,063	10,244
	HOMER	1,536	12,325		HOMER	2,889	10,678
	Cscore	2,408	11,931		Cscore	2,808	10,217

2) Also, in Fig. 2l, it would be helpful to maintain the colors as in the other panels.

We thank the reviewer for this suggestion. We have now revisited the plots to match colors across each panel.

3) In the comparison with calder, I'm not sure I got the numbers right... from what I read in the text, it seems that in total ~8000 bins have been analyzed, each of size 100KB. That corresponds to ~800M base pairs, which is approximately one third of the mouse genome (~2.5B). Why is that? In addition, we have recently analyzed that dataset with the same tool and got very different results, we obtained ~22,500 bins of size 100kb, out of which ~59% changed sub-compartment label, not 80% as stated in the rebuttal document.. Am I misinterpreting something? There might be version differences of the tool, not sure, but the differences seem quite big..

We thank the reviewer for catching the issue. We are deeply sorry with this mishap of numbers. Due to a wrong file usage, our sub-compartment overlapping statistics left out a large chunk of the genome and that resulted in the mismatch mentioned by the reviewer. The results did not change the overall conclusions about subcompartment analysis being unsuitable for a direct differential compartment analysis. We also saw only a minimal change in the number of dChIC differential bins overlapping with Calder labels (1862 instead of 1820).

We have now corrected this and the results we updated in the manuscript are in line with the numbers reported by the reviewer's analysis. The new numbers are highlighted in blue in the main text. Briefly:

- Calder retrieved a total of 24,546 100kb bins (~2.4 GB), **instead of ~800Mb reported before**, with sub-compartment assignments for both ESC and NPC.
- Out of 1,981 dChIC bins, for 1,862, **instead of 1,820**, we had Calder labels on both cell types and among those 97.5% (1,816 bins), **instead of 97.6% (1777 bins)**, overlapped with differential sub-compartments.
- For the remaining 22,684 bins, **instead of 7,967**, with Calder labels that do not overlap with dChIC differential calls, still a high but smaller percentage (57.5%), **instead of 74.8%**, corresponded to differences in sub-compartment labels.
- In terms of being able to do a differential analysis directly from sub-compartments, however, a large percentage (~60.5% or 14,866 out of 24,546 100kb bins), **instead of (~80% or 6,377 out of 7,967 100kb bins)**, of sub-compartment transitions/flips suggest that this approach may lead to low specificity in detecting important differences and would need to be coupled with additional filters and/or supplemented by further statistical assessments.

Reviewers' Comments:

Reviewer #1:

Remarks to the Author:

My comments have been addressed.

Reviewer #3:

Remarks to the Author:

The reviewers addressed my last few requests, happy to recommend this manuscript for publication.